# Triboiontronics with temporal control of electrical double layer formation

Xiang Li [1,2], Roujuan Li[1,2], Shaoxin Li [1], Zhong Lin Wang [1,3,4,5] & Di Wei [1,6]

The nanoscale electrical double layer plays a crucial role in macroscopic ion adsorption and reaction kinetics. In this study, we achieve controllable ion migration by dynamically regulating asymmetric electrical double layer formation. This tailors the ionic-electronic coupling interface, leading to the development of triboiontronics. Controlling the charge-collecting layer coverage on dielectric substrates allows for charge collection and adjustment of the substrate-liquid contact electrification property. By dynamically managing the asymmetric electrical double layer formation between the dielectric substrate and liquids, we develop a direct-current triboiontronic nanogenerator. This nanogenerator produces a transferred charge density of 412.54 mC/m$^2$, significantly exceeding that of current hydrovoltaic technology and conventional triboelectric nanogenerators. Additionally, incorporating redox reactions to the process enhances the peak power and transferred charge density to 38.64 W/m$^2$ and 540.70 mC/m$^2$, respectively.

The nanoscale electrical double layer (EDL) has significant implications in various chemical[1,2], biological[3], and environmental[4] processes. It has a pivotal role in determining the electrical properties of interfaces and impacts the adsorption[5], reaction rates[6], and selective transport of substances[7]. Understanding the EDL formed at solid-liquid or liquid-liquid interfaces is crucial in fields such as energy harvesting[8–10], storage[11], catalysis[12], and colloid formation[13]. For this reason, there has been continuous exploration of the structure and composition of the EDL in the past two centuries. The initial flat model, proposed by Helmholtz in 1853[14], described the EDL as two opposite-charged layers tightly formed at the electrode-electrolyte interface. In the 1910s, Gouy and Chapman introduced the concept of the diffuse layer within the EDL[15,16]. Later in the mid-20th century, Stern and Graham considered the influence of solvation and ion size, leading to the widely used Stern EDL model[17,18]. This model consists of a compact Stern layer and a diffuse layer composed of free ions in solution. The Stern layer contains charges adsorbed tightly on the electrode surface, divided into the inner Helmholtz plane (IHP) and the outer Helmholtz plane (OHP),

depending on their distance to the electrode. It should be noted that these EDL models focused primarily on the conductor-liquid interface and did not consider the interaction between insulating solid dielectrics and liquids. Recently, a two-step model of the EDL was proposed[19–21], which could be applied to the insulating dielectric-liquid interfaces. In this model, electron transfer and ionization reactions occurred almost simultaneously, leading to the simultaneous transfer of electrons and adsorption of ions on the dielectric surface to form the IHP. Then, under the electrostatic forces, ions of opposite polarity were adsorbed onto the charged surface, forming the OHP. According to the two-step model, the EDL represents an excellent ionic-electronic coupling interface, at which the dynamics of ions and electrons could be adjusted.

Compared with electronics that solely relied on electrons as charge carriers, iontronics harnessed a diverse array of multivalent ions for charge transport[22], which enhanced the capability to transport multiple charge carriers and information at the ionic-electronic coupling interface, presenting significant potential for enhancing energy

[1]Beijing Institute of Nanoenergy and Nanosystems, Chinese Academy of Sciences, Beijing, P. R. China. [2]School of Nanoscience and Engineering, University of Chinese Academy of Sciences, Beijing, P. R. China. [3]Beijing Key Laboratory of Micro-Nano Energy and Sensor, Center for High-Entropy Energy and Systems, Beijing Institute of Nanoenergy and Nanosystems, Chinese Academy of Sciences, Beijing, P. R. China. [4]Guangzhou Institute of Blue Energy, Knowledge City, Huangpu District, Guangzhou, P. R. China. [5]Georgia Institute of Technology, Atlanta, GA, USA. [6]Centre for Photonic Devices and Sensors, University of Cambridge, 9 JJ Thomson Avenue, Cambridge, UK. e-mail: zhong.wang@mse.gatech.edu; dw344@cam.ac.uk

conversion and information flow[23]. In particular, recent advancements in the dynamic regulation of the EDL have transformed this century-old concept into a novel energy harvesting paradigm[8,9,23]. Firstly, through moving the EDL boundary on the nanostructured carbon on the dielectric substrate driven by liquid evaporation, an ionic current at the level of hundreds of microamperes per square meter could be generated in the hydrovoltaic technology[8], demonstrating the energy harvesting potential. Secondly, in the solid-liquid triboelectric nanogenerator (TENG)[9], by moving the boundary of the diffuse layer on the dielectric material, the electronics displacement current at a milli-ampere level per square meter could be induced in the external circuit connecting the two back charge-collecting layers. Thirdly, in a direct current triboiontronic nanogenerator (DC-TING)[23], the charge density of the diffuse layer on the dielectric surface was supplemented to create an ion concentration gradient to drive ion migration, adjusting the electronic displacement current to generate a DC ionic-electronic coupling current at levels reaching hundreds of milliamperes per square meter. Dynamic regulation of EDLs has created exciting prospects for the advancement of energy harvesting[24–26], capable of converting mechanical motion or liquid evaporation into usable electrical power. Nevertheless, recent advancements primarily concentrated on modulating the characteristics of the diffuse layer, which carried a relatively small triboelectric charge content, resulting in constrained output performance. If the migration of charge carriers could be regulated by controlling the formation of the EDL, the transport and storage of charge carriers could be effectively optimized, thereby achieving higher ion flux and power output.

In this paper, controllable ion migration behavior was achieved by dynamically controlling asymmetric electrical double layer formation between dielectric substrates and liquids, forming triboiontronics. By regulating the coverage of the charge-collecting layer on the dielectric substrate, not only could charge be collected, but also the contact electrification (CE) property between the substrate and the liquid could be adjusted to form distinct EDLs. Through dynamically controlling the CE between the liquid and dielectric substrates with identical charge-collecting layers, asymmetric EDL formation was achieved. It created an ion concentration gradient, generating efficient ionic current in the physically adsorptive direct-current triboiontronic nanogenerator (PDC-TING). The resulting peak power density and transferred charge density reached 8.45 W/m² and 412.54 mC/m², respectively, representing a significant improvement over hydro-voltaic technology and conventional TENGs. Introducing redox reactions by altering metal charge collectors enhanced performance further, leading to a more efficient synergistic DC-TING (SDC-TING) with a peak power density of 38.64 W/m² and transferred charge density of 540.70 mC/m², respectively. This demonstrated that dynamically controlling the EDL formation can regulate ion flux, enhancing energy transfer in integrated energy harvesting and storage devices. It could also mimic the tactile sensing mechanism of the human body to construct the bionic neurologic circuit, which has potential application value in direct human-computer interaction and neuromorphic computing in the future.

## Results

### Triboiontronics via dynamically controlling electrical double layer formation

The EDL functioned as an exceptional interface for ionic-electronic coupling between the solid dielectric and liquid within the two-step model. By dynamically controlling the EDL, iontronics enabled precise control over ion migration and electron coupling transfer at the interface. Here, through dynamically controlling EDL formation at solid-liquid interfaces, efficient triboiontronics was established. Firstly, a metal layer (Au) as a charge-collecting layer was sputtered onto a dielectric substrate (polyethylene terephthalate, PET). Upon contact of deionized (DI) water with the bottom Au/PET layer, it might engage in

direct interaction with the PET substrate via microscopic cracks in the sputtered Au layer, leading to the solid-liquid CE and the establishment of a stable EDL (Fig. 1a). Secondly, as the top Au/PET layer moved downwards to contact with DI water, the initial CE led to the formation of a new EDL, thus establishing two EDLs with significantly different symmetries (Fig. 1b). This created an ion concentration gradient, facilitating electron transfer in the external circuit and generating an ionic current $I_{i1}$. Thirdly, the detachment of the top Au/PET layer halted ion migration (Fig. 1c). Through repeated contact and separation cycles, the constructed ion concentration gradient gradually weakened, yet the ion migration persisted until equilibrium was achieved between the two EDLs (Fig. 1d). Based on this regulation mechanism, the PDC-TING was invented, as illustrated in Fig. 1e. The verification experiment of the regulation mechanism was carried out (Fig. 1f). When 200 μL water was dropped on the pristine bottom Au/PET layer sputtered with 1-minute Au (Supplementary Fig. 1a), the PDC-TING that was driven by the ion concentration gradient could generate a short-circuit current ($I_{SC}$) of 0.97 μA in DC form at an operating frequency of 1 Hz. When the water was replaced by polytetrafluoroethylene (PTFE) film attached to the bottom Au/PET layer (Supplementary Fig. 1b), the PDC-TING was converted to a conventional solid-solid TENG based on the coupling principle of the CE and electrostatic induction (Supplementary Fig. 2), generating alternating current (AC) electronics signals with $I_{SC}$ of 0.80 μA. When DI water was dispensed onto the PTFE film adhered to the bottom Au/PET surface (Supplementary Fig. 1c), the ion migration was hindered, and the conventional solid-liquid TENG was constructed. Weaker AC electronics signals with $I_{SC}$ of 3.53 nA were generated only by electrostatic induction. The corresponding transferred charge ($Q_{SC}$) and open-circuit voltage ($V_{OC}$) are shown in Supplementary Fig. 3. Notably, upon applying 200 μL of oil or liquid paraffin onto the pristine bottom Au/PET layer, the absence of ions in these solutions could impede the formation of an ion concentration gradient. The AC displacement electrical signals were generated by the CE and electrostatic induction, with $I_{SC}$ of 3.17 nA for oil and $I_{SC}$ of 3.53 nA for liquid paraffin (Fig. 1g). In contrast, under the 99% glycerol containing a small number of ions, the DC output was generated ($I_{SC}$ of 59.17 nA). The corresponding $Q_{SC}$ and $V_{OC}$ are exhibited in Supplementary Fig. 4. Experiments demonstrated that ion migration propelled by the ion concentration gradient might play a pivotal role in the operating mechanism of the PDC-TING.

The impact of the extent and direction of asymmetry in the EDL formation on the PDC-TING output was investigated by pre-moistening different Au/PET layers (Fig. 1h). Specific pre-moistening methods are shown in Supplementary Fig. 5. When applying 200 μL water in the PDC-TING with both pristine Au/PET layers sputtered with 1- m Au, the constructed ion concentration gradient is weaker, resulting in a lower positive $I_{SC}$ of 0.97 μA. Pre-moistening was exclusively applied to the bottom Au/PET layer with water to pre-form the EDL. It could effectively increase the extent of the ion concentration gradient in the PDC-TING, improving the positive $I_{SC}$ to 8.20 μA. On the contrary, the pre-moistening of the top Au/PET layer could pre-built the EDL, causing a higher reverse ion concentration gradient in the PDC-TING with negative electrical signals ($I_{SC}$ of − 2.75 μA). When both Au/PET layers were pre-moistened, EDLs formed on their respective surfaces, resulting in a diminished ion concentration gradient within the PDC-TING and weaker positive electrical signals with $I_{SC}$ of 0.45 μA. The corresponding $Q_{SC}$ and $V_{OC}$ are displayed in Supplementary Fig. 6. Experimental results confirmed that the extent and direction of asymmetry in the EDL formation played a crucial role in determining the efficacy and direction of ion migration, thereby influencing the PDC-TING output. Pre-moistening the bottom Au/PET layer could facilitate establishing a higher ion concentration gradient, significantly enhancing the ion flux within the PDC-TING. Furthermore, as indicated in Supplementary Fig. 7, the PDC-TING output remained relatively stable at $I_{SC}$ of about 8.20 μA, irrespective of whether the bottom

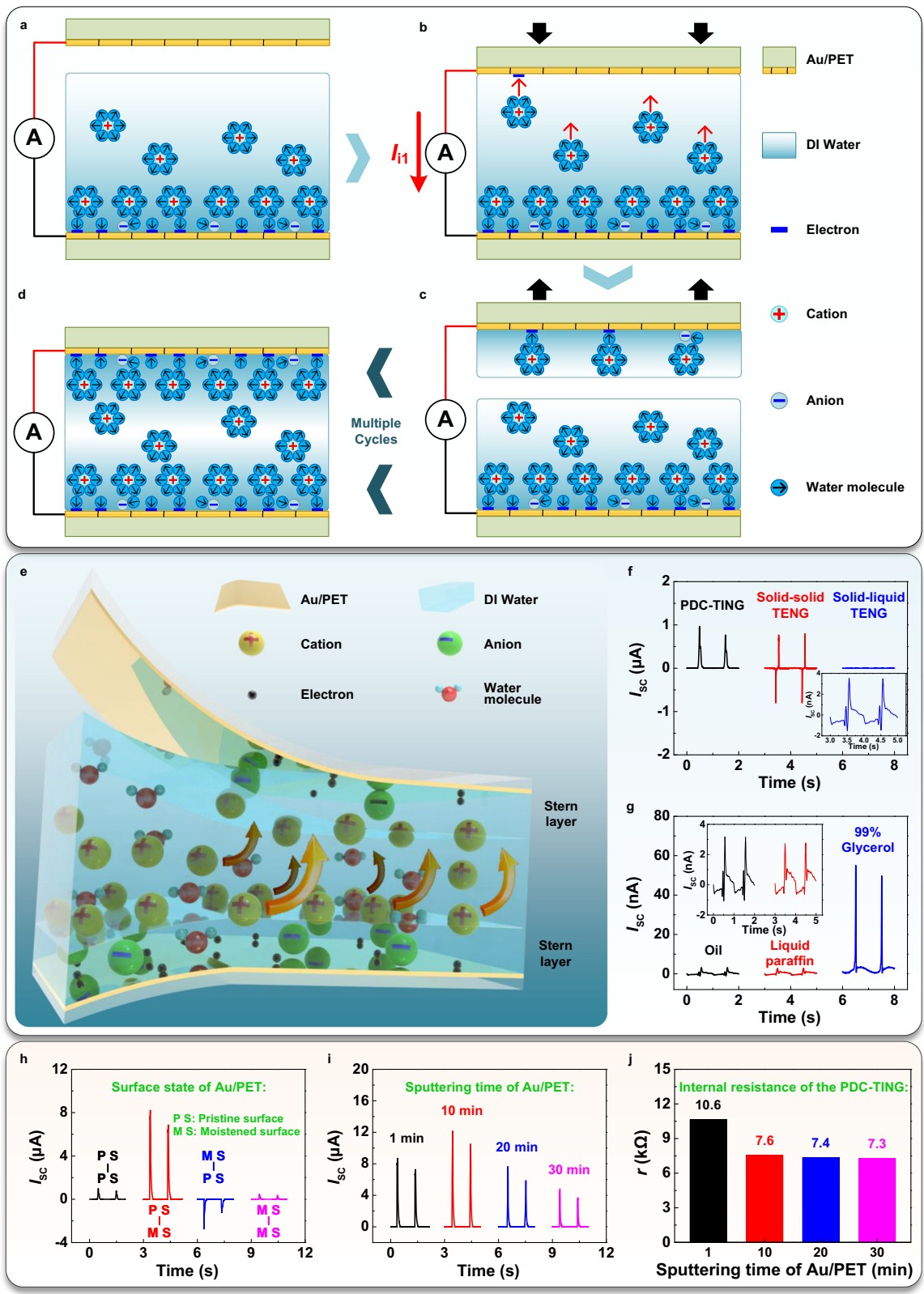

**Fig. 1 | The principle of the triboiontronics and the construction of the PDC-TING. a** Triboiontronics with temporal control of electrical double layer formation. Firstly, when DI water contacted the bottom Au/PET layer, it interacted with the PET substrate through microscopic cracks in the sputtered Au layer, forming a stable EDL. **b** Secondly, as the top Au/PET layer moved downward to contact with water, the initial CE led to form a new EDL, establishing two EDLs with significantly different symmetries. This created an ion concentration gradient, generating an ionic current $I_{i1}$. **c** Thirdly, the detachment of the top Au/PET layer halted ion migration. **d** Through repeated contact and separation cycles, the ion migration persisted until equilibrium was achieved between the two EDLs. **e** The model display of the PDC-TING. **f** Comparison of output performance of different generators. **g** The effect of different liquid types on the PDC-TING output. **h** The effect of the extent and direction of the ion concentration gradient on the PDC-TING output. **i** The effect of the Au-layer sputtering time of Au/PET layers on the PDC-TING output. **j** The effect of the Au-layer sputtering time of Au/PET layers on the internal resistance of the PDC-TING.

Au/PET layer was pre-moistened, immersed in DI water for 2 h, or subsequently underwent evaporation and drying at room temperature (~25 °C) followed by re-moistening. Thus, the PDC-TING output appears to be minimally influenced by the swelling property, water content, and dehydration status of the PET substrate. As an interface phenomenon, as long as the water was in full contact with the bottom Au/PET layer, the EDL density could be guaranteed to ensure the PDC-TING output with relatively stable $I_{SC}$ of about 8.20 µA, independent of the dielectric substrate thickness (Supplementary Fig. 8). It was different from the traditional TENG required a thinner dielectric material to ensure output (Supplementary Fig. 9), thereby the PDC-TING had wider applicability to materials. Subsequently, the effect of the Au sputtering time of the Au/PET layers on the PDC-TING output was investigated (Fig. 1i). Compared to the PDC-TING output ($I_{SC}$ of 8.20 µA) using the Au/PET layer sputtered with 1- m Au, when the Au sputtering time was increased to 10 min, the $I_{SC}$ was enhanced to 12.18 µA. However, as the Au sputtering time further gradually increased to 30 min, the $I_{SC}$ of the PDC-TING was decreased continuously to 4.79 µA. The corresponding $Q_{SC}$ and $V_{OC}$ are shown in Supplementary Fig. 10. In addition, increasing the sputtering time of Au on the PET substrate reduced the internal resistance of the PDC-TING from 10.6 kΩ to 7.3 kΩ (Fig. 1j). Scanning electron microscope (SEM) observations revealed microscopic cracks on the surfaces of PET film and Au/PET film sputtered for 1 minute (Supplementary Fig. 11a, b), indicating insufficient Au deposition within 1 minute, leading to higher internal resistance (10.6 kΩ). With 10 min of sputtering, the cracks notably diminished (Supplementary Fig. 11c), suggesting improved Au coverage and enhanced electrical conductivity, resulting in reduced internal resistance to 7.6 kΩ. Further extending sputtering time beyond 20 min nearly eliminated the cracks, maintaining internal resistance at around 7.4-7.3 kΩ (Supplementary Fig. 11d, e). With the increase in sputtering time, the decrease in transmittance of Au/PET films also proved the increase in Au coverage on the PET substrate (Supplementary Fig. 12). Therefore, adjusting the sputtering time of the metal enabled control over its coverage on the dielectric substrate, thereby affecting the conductivity of the charge-collecting layer. In addition, controlled microscopic cracks could enable precise adjustment of CE properties between the substrate and the liquid, forming diverse EDLs. 10 min of sputtering was set as the optimized time to enhance the performance of the PDC-TING.

To explore the effect of Au layer coverage on the CE characteristics of Au/PET films in more detail, a dedicated system was constructed (Fig. 2a). A 6 cm long tested film was mounted on an acrylic substrate at a 45° inclination angle, and a grounding syringe consistently released a single drop of approximately 25 µL DI water onto the film. Triboelectric charges carried by the droplet were measured when it slid through the tested film into the Faraday cylinder, and then the surface potential of the tested film was measured (Supplementary Fig. 13). According to the two-step EDL model (Fig. 2b), the Stern layer adhered tightly to the tested film, suggesting that the triboelectric charges might be associated with the diffuse layer retained within the droplet. Thus, the evaluation of the triboelectric charge and the surface potential might indicate the EDL density on the tested film. The surface potential of the pristine tested films was calibrated to 0 V. When the water droplet slid through the pristine PET, it generated a positive $I_{SC}$ of 3.50 nA and $Q_{SC}$ of 0.38 nC (Fig. 2c, d), and the surface potential was about −59 V (Fig. 2e). Conversely, sliding on pristine pure Au resulted in weaker reverse signals, measuring $I_{SC}$ of 0.16 nA and $Q_{SC}$ of 0.022 nC, with the surface potential around 3 V. Increasing sputtering time from 1 min 30 min for different pristine Au/PET films decreased positive electrical signals from $I_{SC}$ of 1.42 nA and $Q_{SC}$ of 0.15 nC to $I_{SC}$ of 0.26 nA and $Q_{SC}$ of 0.030 nC. Correspondingly, the surface potential gradually changed from −30 V to −6 V. These signals, reflecting the EDL charge density, revealed that adjusting metal sputtering time

controls coverage on the dielectric substrate. It affected the conductivity of the charge-collecting layer and CE characteristics of the dielectric substrate, enabling the construction of EDLs with varied densities at DI water-PET substrates.

Based on the above experiments, EDL models were developed for various solid-liquid contact interfaces. Firstly, the EDL for the dielectric surface followed the two-step model (Fig. 2f). Secondly, weaker reverse electrical signals from falling droplets indicated reversed charge distribution and lower density in the EDL for the pure metal surface (Fig. 2g). Due to high hydration-free energy, cations couldn't directly adsorb onto the solid surface[27], forming an IHP with the water molecule dipole layer while hydrated cations constituted the OHP. Thirdly, when DI water contacted the sputtered metal charge-collecting layer of the dielectric substrate, the EDL could form through microscopic cracks (Fig. 2h). Its charge distribution resembled that of the EDL from the dielectric surface, with intermediate charge density compared to the dielectric and pure metal surfaces. While pure dielectric could form the EDL, they cannot directly transfer charges due to insulation. Conversely, although pure Au exhibited excellent conductivity, the minimal triboelectric charge in the EDL limited the ion concentration gradient, resulting in a weak electrical signal. Only the dielectric substrate with a sputtered metal layer could ensure that PDC-TING produces effective ion flux, utilizing the distinct advantages conferred by the formation of cracks in the metal sputtering layer. These cracks serve as conduits for liquid infiltration, thereby leveraging the solid-liquid CE properties of the dielectric layer. This enhances charge storage and transfer efficiency. In addition, the cracks facilitate efficient current collection, reducing resistance and improving charge transport within the material. This combined effect enhances the performance of the hybrid material, enabling applications with high power density and rapid charge/discharge rates while offering versatile and innovative solutions across diverse domains. The function of the charge-collecting layer in the combined effect is similar to the role of graphene in hydrovoltaic technology[8,28,29]. However, the chemical vapor deposition (CVD) method demands stringent environmental conditions and intricate processes for the preparation of graphene, etc. In contrast, the magnetron sputtering of metal charge collectors does not require high temperatures or precise atmosphere control as the physical vapor deposition (PVD) method, facilitating large-area deposition. It provides a cost-effective and versatile means to adjust material properties by altering metal coverage on dielectric substrates. This technique enables the creation of a tunable ionic-electronic coupling interface, facilitating in-depth studies of iontronics.

After determining the operation principle of the PDC-TING, the influence of operating frequency on its output was studied (Fig. 3a and Supplementary Fig. 14). As it decreased from 4 Hz to 1 Hz, the $I_{SC}$ of 12.18 µA and $V_{OC}$ of 0.19 V of the PDC-TING remained at a stable amplitude. This phenomenon may be attributed to the ion concentration gradient during the initial contact between the top Au/PET and water, which is influenced by the asymmetry extent of the EDL formation, regardless of the operating frequency. However, the decrease in frequency prolonged the contact time between the top Au/PET layer and water, thereby increasing the migration time of ions and augmenting the generated $Q_{SC}$ from 0.35 µC to 0.60 µC. It differed from conventional TENGs, where the output typically decreases with decreasing operating frequency (Supplementary Fig. 15). To effectively improve the PDC-TING output, three strategies to enhance the ion concentration gradient were explored. The first strategy focused on the regulation effect of electrostatic fields. Based on the solid-solid CE, positive and negative electrostatic fields on the polyamide (PA) and PTFE films were created, respectively. Increasing the rubbing times from 1 to 15 times resulted in saturation of positive and negative charges on the PA and PTFE surface at the 10th time, respectively. The positive electrostatic field on the PA film could promote the

PDC-TING output (Fig. 3b), increasing $I_{SC}$ from 12.42 µA to 14.70 µA (corresponding $Q_{SC}$ and $V_{OC}$ are shown in Supplementary Fig. 16). It might be attributed to the increase in positive electrostatic field and enhanced bottom EDL density (Supplementary Fig. 17), strengthening the ion concentration gradient. Conversely, the negative electrostatic field on the PTFE film could reduce the PDC-TING output (Fig. 3c),

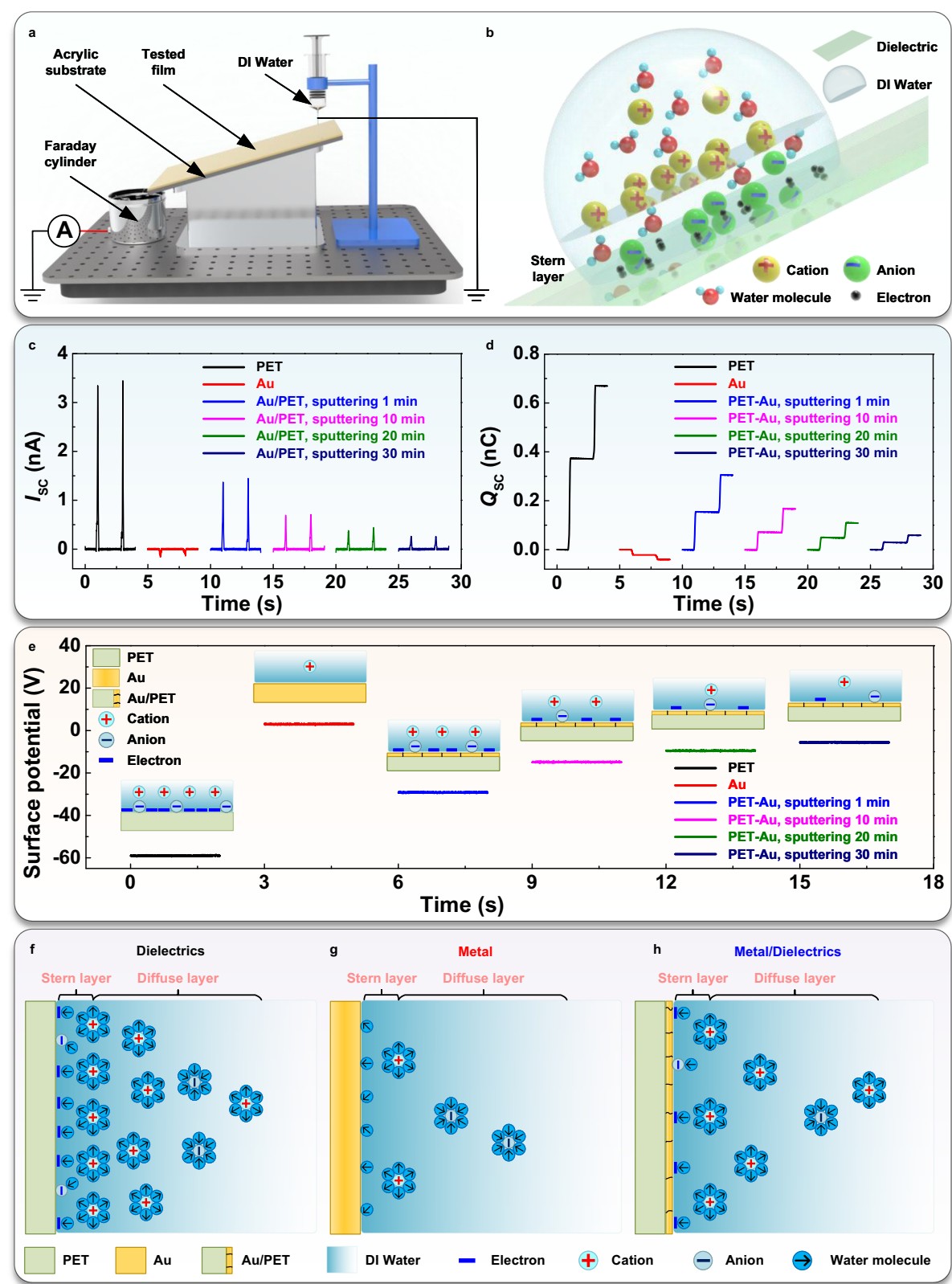

**Fig. 2 | The effect of the Au layer sputtering time of the Au/PET layer on the substrate-DI water CE was investigated. a** The testing system of triboelectric charges carried by the droplet sliding through the tested film surface. **b** The two-step EDL model for the dielectric surface. **c** and **d** the $I_{SC}$ and $Q_{SC}$ generated by water droplets sliding through the different film surfaces. **e** The surface potential of different films after droplet sliding. **f**–**h** EDLs on different surfaces of the dielectric, metal, and dielectric substrate sputtered with the metal layer, respectively.

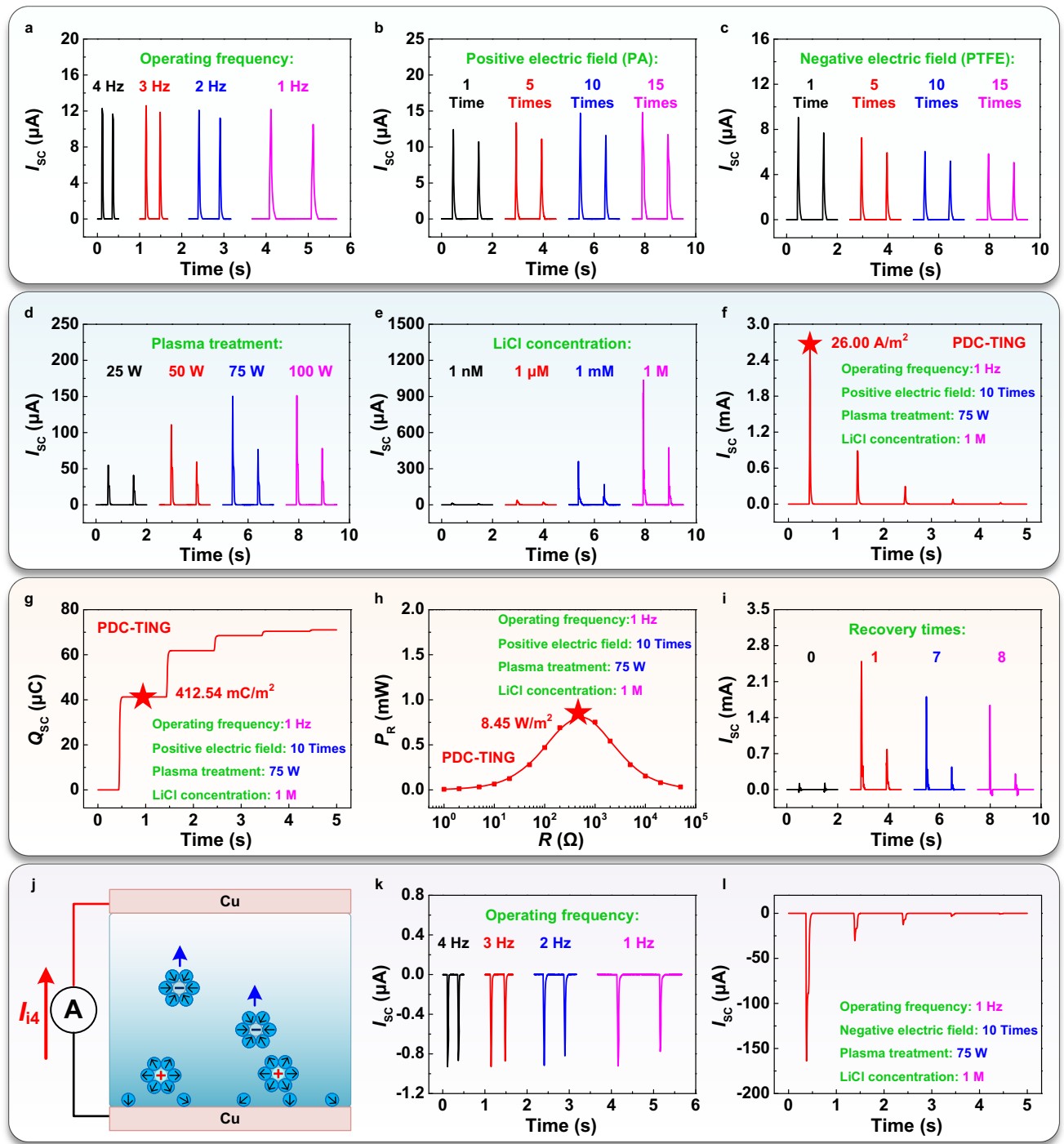

**Fig. 3 | The specific affecting factors on the PDC-TING output were explored.** **a** The $I_{SC}$ of the PDC-TING at different operating frequencies. **b**, **c** The positive and negative electrostatic fields were used to adjust the constructed ion concentration gradient. **d** The influence of hydrophilicity of the bottom Au/PET layer on the constructed ion concentration gradient. **e** The effect of ion concentrations in the liquid on the constructed ion concentration gradient. **f**, **g**, **h** The $I_{SC}$ density, $Q_{SC}$ density, and $P_R$ density of the PDC-TING could reach 26.00 A/m², 412.54 mC/m², and 8.45 W/m², respectively. **i** The PDC-TING output could be restored by the electrochemical recovery. **j** Through dynamically controlling EDL formation between the liquid and the pure metal, an ion concentration gradient could also be constructed to drive the directional migration of ions in the metal-based PDC-TING. **k** The electrical signal direction of metal-based PDC-TING was opposite to the PDC-TING based on the metal/dielectric substrate. **l** Due to the sparse EDL on the pure metal surface, the optimal negative output of the metal-based PDC-TING was weaker than the optimal positive output of the PDC-TING based on the metal/dielectric substrate.

decreasing $I_{SC}$ from 9.06 μA to 6.05 μA (corresponding $Q_{SC}$ and $V_{OC}$ are shown in Supplementary Fig. 18). It might be attributed to the negative electrostatic field, which reduces the bottom EDL density (Supplementary Fig. 19), thereby weakening the concentration gradient. Secondly, Plasma treatment of the bottom Au/PET layer was conducted to examine the impact of surface hydrophilicity on the ion concentration gradient (Fig. 3d). Initially, untreated Au/PET layers generated $I_{SC}$ of 12.18 μA, with a contact angle of approximately 100° (Supplementary Fig. 20), indicating hydrophobic properties. Increasing the power of Plasma treatment from 25 W to 75 W reduced the contact angle from about 49° to 19° by introducing oxygen-containing functional groups, enhancing hydrophilicity. It effectively increased

the bottom EDL density, boosting the ion concentration gradient and elevating $I_{SC}$ from 54.79 μA to 150.08 μA. Beyond 75 W, further increases in power did not significantly affect hydrophilicity, thereby maintaining output stability. The corresponding $Q_{SC}$ and $V_{OC}$ are shown in Supplementary Fig. 21. Thirdly, the effect of ion concentration in the liquid on the PDC-TING output was investigated by replacing DI water with different LiCl solutions (Fig. 3e). When the concentration of LiCl solution was increased from 1 nM to 1 M, improvements were observed in the PDC-TING output ($I_{SC}$ from 13.10 μA to 1035.74 μA), and the internal resistance was effectively reduced from about 7.6 MΩ to 500 Ω. The corresponding $Q_{SC}$ and $V_{OC}$ are shown in Supplementary Fig. 22. Experiments demonstrated that increasing the ion concentration in the liquid could effectively improve the ion concentration gradient and reduce the internal resistance, thereby significantly improving the PDC-TING output. In addition, as the ion concentration increased from 1 μM to 1 M (Supplementary Fig. 23), the output performance remained consistent for neutral (LiCl) and alkaline (NaOH) solutions, with $I_{SC}$ increasing from about 40 μA to approximately 1000 μA. However, under acidic conditions (HCl), the PDC-TING exhibited weaker $I_{SC}$ compared to the other two solutions. Particularly at a higher concentration of 1 M, a lower reverse electrical signal was generated (about 450 μA). This behavior might be attributed to the higher adsorption of H$^+$ ions from the HCl solution onto the dielectric surface, leading to a reversal of the charge distribution pattern within the EDL[21,28,30].

The optimal output of PDC-TING could reach $I_{SC}$ of 2.60 mA (Fig. 3f), $Q_{SC}$ of 41.25 μC (Fig. 3g), $P_R$ (peak power) of 0.85 mW (Fig. 3h), and $V_{OC}$ of 0.25 V (Supplementary Fig. 24). The $Q_{SC}$ density and $P_R$ density could reach 412.54 mC/m$^2$ and 8.45 W/m$^2$, respectively. During its operation, the EDL on the top Au/PET surface became denser, resulting in a gradual decrease in the ion concentration gradient. To maintain the PDC-TING output, the electrochemical recovery could be utilized to restore the initial surface of the top Au/PET layer (Supplementary Fig. 25). After undergoing 7 recovery times, the PDC-TING could still generate ideal DC signals with $I_{SC}$ of 1.81 mA, $Q_{SC}$ of 32.93 μC, and $V_{OC}$ of 0.20 V (Fig. 3i and Supplementary Fig. 26). By dynamically controlling EDL formation between the pure metal (such as Cu) and liquids, a lower ion concentration gradient was achieved, enabling the development of metal-based PDC-TING (Fig. 3j and Supplementary Fig. 27). As the operating frequency decreased from 4 Hz to 1 Hz, It exhibited a relatively stable negative $I_{SC}$ of − 0.93 μA and $V_{OC}$ of − 0.14 V, and the $Q_{SC}$ changed from − 23.82 nC to − 36.18 nC (Fig. 3k and Supplementary Fig. 28). The optimal negative output of the metal-based PDC-TING could reach $I_{SC}$ of − 163.85 μA, $Q_{SC}$ of − 7.45 μC, $V_{OC}$ of − 0.21 V, and $P_R$ of 3.03 μW (Fig. 3l and Supplementary Fig. 29), which was weaker than the optimal positive output of the PDC-TENG with Au/PET. This phenomenon may be attributed to the opposite and weaker charge distribution and density in the EDL for the pure metal surface (Fig. 2g) compared to the metal/dielectric surface (Fig. 2h).

### More efficient triboiontronics via the synergy of controlling EDL formation and redox reactions

Modifying the charge-collecting layer on the PET dielectric substrate by changing the metal type, enhanced the efficiency of SDC-TING (Fig. 4a and Supplementary Fig. 30). Upon contact between the top Au/PET layer and water, alongside the fully contacted bottom Cu/PET layer, an ion concentration gradient was established. The contrasting electrochemical behaviors of Au and Cu in water induced redox reactions on their surfaces. Cu, being more electrochemically active, underwent oxidation, releasing electrons into the water to form Cu$^{2+}$ ions, as illustrated by:

$$Cu - 2e^- = Cu^{2+} \tag{1}$$

Simultaneously, the Au surface accepted electrons and potentially reduced O$_2$ in the water to form OH$^-$. Its equation might be expressed as:

$$O_2 + 2H_2O + 4e^- = 4OH^- \tag{2}$$

The cyclic voltammetry curve depicted in Supplementary Fig. 31 served as verification that the redox reaction indeed occurs. Introducing this redox reaction enabled the production of more Cu$^{2+}$ ions. Their directional migration, driven by the electroelectrochemical potential difference, effectively boosts the ion flux. The synergistic effect of dynamically controlling EDL formation and redox reactions was systematically investigated (Fig. 4b). Initially, pre-moistening the bottom Cu/PET layer individually with water established the EDL. Upon initial contact of the top Au/PET layer with water, the combined ion concentration gradient and electrochemical potential difference drove upward ion migration, significantly increasing the $I_{SC}$ to 16.82 μA. After about 50 sec, the $I_{SC}$ stabilized at 4.80 μA under the electrochemical potential difference alone. Subsequently, pre-moistening the top Au/PET layer individually led to a reverse ion concentration gradient upon contact with water. This counteracted the electrochemical potential difference, leading to a decrease in $I_{SC}$ to 1.33 μA, with the gradient disappearing after approximately 20 sec. Finally, pre-moistening both layers constructed a weaker positive ion concentration gradient. This, combined with the electrochemical potential difference, contributed to an $I_{SC}$ of 5.77 μA for PDC-TING, with the synergistic output lasting 15 sec. These findings highlighted the effective promotion of coupling migration of ions and enhancement of ion flux through the synergistic effect of dynamically controlling EDL formation and redox reactions. In particular, under the synergistic effect, the pure energy harvesting device of PDC-TING was converted to an integrated energy harvesting and storage device SDC-TING. Utilizing the substantial Cu$^{2+}$ ion flux generated by the redox reaction, the SDC-TING could enhance its output performance and operational stability simultaneously. In an environment with an ion concentration gradient, the variation in electrochemical potential or Gibbs free energy from the ion concentration gradient is analogous to the free energy change in electron transfer reactions. The relationship between electron transfer rate ($k$) and Gibbs free energy change ($\Delta G$) can be expressed by the following equation:

$$k = k_0 e^{\left(-\frac{\Delta G}{4k_B T}\right)} \tag{3}$$

where $k_0$ is the pre-exponential factor, $k_B$ is the Boltzmann constant, and $T$ is the temperature. $\Delta G$ is the Gibbs free energy change, which can be represented by the electrochemical potential difference $\Delta u$:

$$\Delta G = - zF\Delta u \tag{4}$$

where $z$ is the number of charges in the reaction, and $F$ is the Faraday constant. In an ion concentration gradient environment, the relationship between the $\Delta \mu$ and the ion concentration gradient can be expressed by the Nernst equation:

$$\Delta u = RT \ln\left(\frac{a_2}{a_1}\right) \tag{5}$$

where $R$ is the ideal gas constant, and $a_1$ and $a_2$ represent the activity of ions in the solution. The relationship between the $\Delta u$ and the concentration of ions $c$ can be derived from the activity of ions:

$$\Delta u = RT \ln\left(\frac{c_2}{c_1}\right) \tag{6}$$

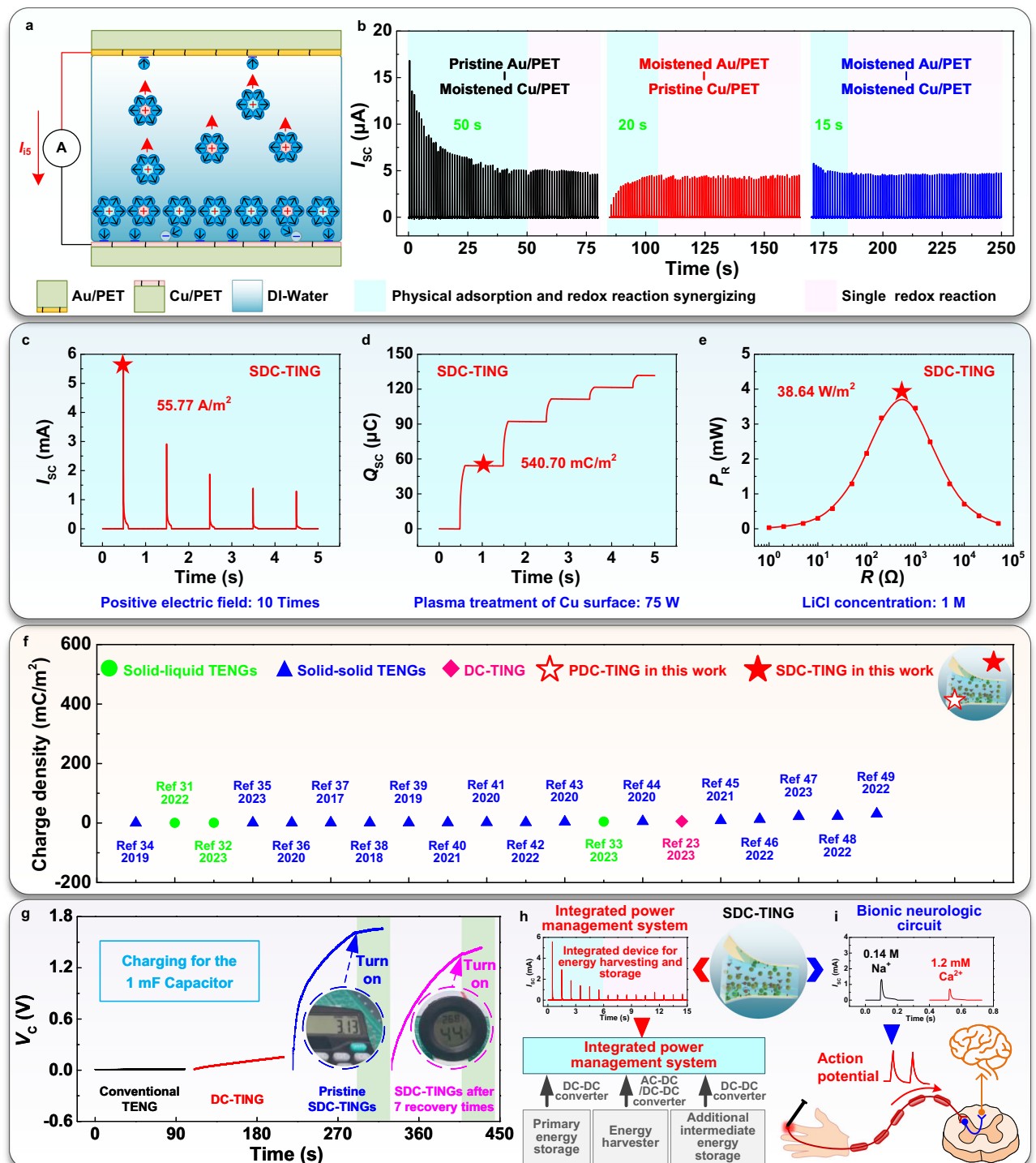

**Fig. 4 | More efficient triboiontronics via the synergistic effect of dynamically controlling EDL formation and redox reaction. a** The synergistic principle of dynamically controlling EDL formation and redox reaction in the SDC-TING. **b** The verification experiment of their synergistic effect. **c, d, e** The $I_{SC}$ density, $Q_{SC}$ density, and $P_R$ density of the SDC-TING. **f** The comparison of the $Q_{SC}$ density of the conventional solid-liquid TENGs, solid-solid TENGs, DC-TING, PDC-TING, and SDC-TING. **g** The comparison of the charging performance of different types of generators for 1mF capacitors. **h** The SDC-TING could be used as an excellent integrated power management system with brilliant application prospects in the field of energy harvesting and storage. **i** Mimicking human tactile neural circuits, the SDC-TING could construct a bionic neurologic circuit by controlling ion migration behavior.

In summary, an increase in ion concentration gradient will lead to an increase in electrochemical potential difference, thereby affecting the change in Gibbs free energy, and consequently impacting the electron transfer rate. With an internal resistance of 500 Ω, the SDC-TING with 1 M LiCl could achieve the $I_{SC}$ of 5.58 mA (Fig. 4c), $Q_{SC}$ of 54.07 μC (Fig. 4d), $P_R$ of 3.86 mW (Fig. 4e), and $V_{OC}$ of 0.30 V (Supplementary Fig. 32). The $Q_{SC}$ density and $P_R$ density of the SDC-TING could reach up to 540.70 mC/m$^2$ and 38.64 W/m$^2$, which was increased by 435.35 and 2869.23% respectively compared to that completely depending on the redox reactions (Supplementary Fig. 33). After comparison, the contribution of dynamically controlling EDL formation to the $Q_{SC}$ of the SDC-TING during the initial contact was

81.32%, while the contribution of redox reactions was 18.68%. In addition, through the 10-node connecting in series (Supplementary Fig. 34), the 0.30 V $V_{OC}$ of a single SDC-TING was effectively increased to 2.88 V (Supplementary Fig. 35).

Compared with the $Q_{SC}$ densities of hydrovoltaic technology by moving the EDL boundary[8,28,30], conventional solid-liquid TENGs[9,31–33] and solid-solid TENGs[34–49], or DC-TING[23], that of PDC-TING or SDC-TING was increased by several orders of magnitude (Fig. 4f). Hydrovoltaic technology[8,28,30] typically manipulated the EDL boundary on the charge-collecting layer of the dielectric substrate, often resulting in limited ion concentration gradients and increased migration distances, which restricted overall output. Conventional solid-liquid TENG[9,31–33] and solid-solid TENG[34–49] utilized triboelectric charges on dielectric materials to induce electronic displacement currents. The insulation properties of dielectrics usually led to higher internal resistance[50] (in the ten mega-Ohm range), enabling higher $V_{OC}$ but limiting both $I_{SC}$ and $Q_{SC}$. DC-TING[23] supplemented charge density within the diffuse layer on the dielectric surface to create an ion concentration gradient, driving ion migration and adjusting electronic displacement current for efficient ionic-electronic coupling. It had higher power density characteristics, but its higher internal resistance (in the ten mega-Ohm range) restricted $Q_{SC}$. In contrast, the PDC-TING and SDC-TING, built upon efficient ion migration propelled by the ion concentration gradient, exhibited higher current and lower internal resistance (in the hundred Ohm level), leading to a conventional $P_R$ density but a higher $Q_{SC}$ density compared to their counterparts. They could be better suited for applications demanding long-term, continuous supply of stable energy output, such as calculators or portable environmental monitoring devices. As shown in Fig. 4g, under the condition of ensuring the same total working area and at the same operating frequency of 1 Hz, the conventional solid-solid TENG could hardly supply energy for the 1 mF capacitor, and the DC-TING could only charge it to 0.15 V within 100 s. Excitingly, the pristine SDC-TINGs in the 10-node series could charge a 1 mF capacitor to 1.5 V within 60 sec without the rectifier bridge to power a calculator (Supplementary Fig. 36 and Supplementary Movie 1). After 7 recovery times, it could still ensure the stable operation of a hygrothermograph (Supplementary Fig. 37 and Supplementary Movie 2). Furthermore, the SDC-TING, as an integrated device for energy harvesting and storage, unveiled profound application prospects in the realms of energy and information flow. Specifically, in the field of energy, the SDC-TING could operate in two interrelated stages: the integrated energy harvesting and storage stage, followed by the separate energy storage stage. During the integrated stage, the SDC-TING synergistically generated more power, delivering an accelerated energy supply to electrical apparatuses such as capacitors (Supplementary Fig. 38a, b). The subsequent separate energy storage stage substantially enhanced the stability of the SDC-TING (Supplementary Fig. 38c), significantly surpassing that of the PDC-TING. It could serve as an integrated power management system without complex conversion circuits to connect primary energy storage, energy harvester, and additional intermediate energy storage units, potentially opening new avenues for research in the efficient harvesting and storage of energy (Fig. 4h). In the field of information flow, the SDC-TING demonstrated the capability to regulate various ion fluxes, such as $Na^+$ and $Ca^{2+}$, by establishing a mechano-driven ion concentration gradient (Fig. 4i), thereby generating information flow. Ions carry essential information and perform specific functions in biological systems. For instance, $Ca^{2+}$ signals activate neurotransmitter release, modulate neuronal function, and facilitate cardiac muscle relaxation efficiently[51,52]. $Na^+$ ions regulate blood pressure, volume, and osmotic equilibrium and enable excitation propagation by controlling action potential rate and duration[53,54]. The property of SDC-TING allows it to mimic human tactile neural circuits, facilitating the development of bionic neurologic circuits. These circuits can perform threshold-sensing control functions, paving the way for future human-computer interaction and neuromorphic computing. This highlights the significant potential of SDC-TING in energy and information flow.

## Discussion

The controllable ion migration was achieved by dynamically controlling EDL formation between dielectric substrates and liquids, forming efficient triboiontronics. A cost-effective and versatile way to adjust material properties by altering metal coverage on dielectric substrates was developed. Through dynamically controlling the CE between liquids and dielectric substrates with the same type of charge-collecting layer, an ion concentration gradient was constructed in the PDC-TING, generating a peak power density of 8.45 W/m² and a transferred charge density of 412.54 mC/m². Designed redox reactions were further introduced to synergize with dynamically controlling EDL formation, enabling a more efficient SDC-TING with a peak power density of 38.64 W/m² and a transferred charge density of 540.70 mC/m². Both the PDC-TING and SDC-TING demonstrated several orders of magnitude increases in transferred charge density compared to hydrovoltaic technology and conventional TENGs by manipulating the EDL boundary. The synergistic regulation of EDLs could balance the charge density in the vicinity of the dielectric substrate and create a tunable ionic-electronic coupling interface, facilitating in-depth studies of iontronics. Such triboiontronics could not only integrate energy harvesting and storage into one device but also offer a versatile platform for probing the ion dynamics for human-computer interaction and neuromorphic computing.

## Methods

### Materials

The magnetron sputtering deposition system (Discovery 635, Denton Vacuum, America) sputtered the metal charge-collecting layer on the dielectric substrate surface. The sputtering power used for this process was 50 W. The sputtering method was DC sputtering. The rotational speed during sputtering was kept at 50 r/min.

### Fabrication of the PDC-TING and SDC-TING

The PDC-TING and SDC-TING had an area of 1 cm². The PDC-TING and SDC-TING mainly consisted of five main parts: the bottom dielectric substrate, the bottom charge-collecting layer, the liquid, the upper charge-collecting layer, and the upper dielectric substrate. The thickness of the PET film substrate constituting the PDC-TING and SDC-TING was 100 μm. The thickness of the pure Cu constituting the metal-based PDC-TING and SDC-TING was 1 mm. The thickness of the PTFE film constituting the conventional TENG and DC-TING was 100 μm. The plasma treatment of the charge-collecting layer on the bottom dielectric substrate was carried on by a plasma cleaner (CPC-A, CIF, China) with a treatment time of 30 sec.

### Electrical measurement

The PDC-TING and SDC-TING operated normally with driving forces supplied by a linear motor (PL0119x600/520, LinMot, Switzerland). Their output electrical signals were collected by a test system consisting of an electrometer (6514, Keithley, USA) and a data acquisition card (BNC-2120, National Instruments, USA). The Faraday cylinder was used to test the charge carried by the droplets sliding through the tested film. The high-speed surface potentiometer (347, TREK, USA) was used to test the surface potential of the film. Morphologies of devices were observed by the SEM (SU8020, Hitachi) with the 5.0 kV accelerating voltage and 10 μA emission current.

### Calculation method

The short-circuit current ($I_{SC}$) density, transferred charge ($Q_{SC}$) density, and peak power ($P_R$) density designed in this manuscript are calculated by using the short-circuit current, transferred charge, and peak power generated by the device divided by the device's generating area (the area of the charge-collecting layer), respectively.

**Reporting summary**

Further information on research design is available in the Nature Portfolio Reporting Summary linked to this article.

## Data availability

The authors declare that all the data that support the findings of this study are available within the article and its supplementary information files. Source data are provided in this paper. Source data are provided with this paper.

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

## Acknowledgements

This work was supported by the Beijing Natural Science Foundation (Grant No. IS23040).

## Author contributions

D.W. and Z.L.W. proposed the idea and the project. D.W. designed all the experiments and supervised the whole project. X.L. carried out the experiments in this paper and analyzed the corresponding data. X.L., R.J.L., and S.X.L. analyzed the operation principle of the PDC-TING. All the authors discussed the results and commented on the manuscript. D.W. and X.L. wrote this paper.

## Competing interests

The authors declare no competing interests.
