## [Peer Review File · Nature Communications]

Triboiontronics with temporal control of electrical double layer formationEditorial Note: Parts of this Peer Review File have been redacted as indicated to remove third-party material where no permission to publish could be obtained.

REVIEWER COMMENTS

Reviewer #1 (Remarks to the Author):

The authors report the dynamic regulation of asymmetric electrical double layers to form efficient triboiontronics. The paper is solid and well-written, though, the authors could improve the manuscript with respect to the following comments:

- 1) The caption of Figure 2a should provide details regarding Figure 2a i, ii, iii and iv
- 2) Page 10, line 178 + page 13, line 208, water is playing a key-role in such triboiontronics, so quid of the swelling ratio or the water content of the pre-moistening substrate? In addition, quid of dehydration of the substrate over time and its impact on the performances?
- 3) Page 10, line 189. the purpose of using ion-free oil, liquid paraffin and glycerol should be introduce within the text for improved clarity

Reviewer #2 (Remarks to the Author):

Triboiontronic nanogenerators represents an interesting approach for energy harvesiting applications, especially in the low frequency range and for energy storage. I intepret this approach as a combination of traditional TENGs and energy harvested from ion concentration/salinity gradients. This study investigates parameters and properties that affects output characteristics of triboiontronic nanogenerators. Charge collecting layer, operating frequency, and salt concentration is a few parameters in focus of this study that highly influence the performance.

I find the term "asymmetric EDL" a bit unclear as it is very open for intepretation. My intepretation is that the assymetry is mostly based on a not yet fully formed/charged EDL on the upper substrate, originated from a difference in concentration gradient. But from the word assymetric, that could be material properties or device geometries for example. Therefor, I would consider interpreting and naming this approach as "temporal control of EDL formation" or similar, rather than a "regulation of assymetric EDL".

I find the comparison with TENG and previous TINGs is ambitious, yet a bit difficult to validate. From the perspective that the energy here can represent a combination of triboelectric energy and Gibbs free energy/chemical potential from the salinity gradient, I would really appreciate an analysis of this from the authors, and additional references from the chemical potential perspective. I not that the highest values are recorded at the highest concentration. The situation might be clearer with a more analytical description of the equation used. Can the numbers be divided into different sources of contribution?

The previous work of the authors, referenced in 21. Li, X. et al. Triboiontronics for efficient energy and information flow. Matter 6, 3912-3926 (2023), is closely related. Here, peak power density is highlighted as the advancement that is outstanding in relation to previous work. In the current manuscript, the peak power density is lower, and the transferred charge density is higher. For what applications would either property be most important?

I also lack a more thorough description of the addition of redox reactions, and method description of experiment 4a and b. What voltage was applied to drive the redox reactions? Apologies if this information is there but missed.

When discussing the collecting layer thickness, would the authors like to comment on the drop in internal resistance from 1 to 10 min? Could not just thickness but also coverage/pinholes play a role? Here, microscopy images would be interesting, but not necessary. I would however consider moving some images from SI to main text.

The text is well organized and easy to follow. The introduction and conclusion sections are especially well conducted, and well summarized. Plenty of data is provided, and the SI is very thorough and a great support for the study.

Again, I find this work and approach very interesting! The energy storage perspective (and information flow) could be highlighted a bit more. Some more analysis and decriptions/discussion is needed for me to properly validate the comparison presented in the graphical abstract and Fig 4f. Figure 4i is motivating, but I would appreciate more discussion/validation and comments on how this method would be realised.

Reviewer #3 (Remarks to the Author):

This manuscript reports a direct-current triboiontronic nanogenerator based on the physically adsorbed ions. The data format in the manuscript is very monotonous, and there is a lack of data or characterization to support the concept of an asymmetric double electric layer. The concept proposed by the authors is not innovative, the basic principle is not novel, the output performance is not impressive, and there are many confusing and unexplained descriptions. Considering the high standards of this journal, I believe that this manuscript is not suitable for publication in this journal. Here are some questions for reference:

1. Lines 114-116 states that the denser Stern layer on the PET surface hinders subsequent liquid-solid interface charge transfer. The authors should provide more evidence to support this claim. One way to do this would be to characterize the potential of the PET solid surface. Additionally, does a denser Stern layer on the PET surface mean that the solid surface charge is saturated?
2. Lines 122-123 states that the low ion concentration in the diffusion layer limits the output. However, in liquid-solid contact electrification, high ion concentrations in the liquid usually suppress the output. The authors are requested to explain this contradiction.
3. Figure 1 shows that the authors' data indicate that water carries almost no charge in the later stages of the test. The reason for this needs to be further explained.
4. The thickness of gold or copper should be characterized for different deposition times, and their surface morphology should also be characterized. Is the metal film uniform and completely covers the PET when the deposition time is short?
5. The authors did not provide a description of Fig. 1 g-i. The comparison of these three EDL is very important, and the authors are requested to supplement it.
6. The organization of the author's Figure 1 is problematic. Overall, the manuscript is more concerned with increasing the output performance. However, from the perspective of Figure 1, the introduction of the charge collection layer reduces the charge carried by the liquid. This is easy to cause misunderstanding for readers because they cannot obtain the advantages of this new strategy. I strongly suggest that the author should put the relevant data of Figure 2 into Figure 1 to further highlight the ability of this strategy to increase output performance. In addition, the significance of the data in Figure 1 is confusing, and the contact between the liquid and PET does not guide the subsequent data.
7. It is not sufficient to only discuss the change in liquid charge in Figure 1. Data on the change in solid surface charge should also be included.
8. In Line 161, the authors believe that the cations in the lower EDL will participate in the construction of the upper EDL. This seems impossible, as the size of the EDL is only a few to tens of nanometers. The signal may be generated by the dynamic change of the upper contact area with water, which causes the charge imbalance between the upper and lower devices.
9. The "pre-moistening of PET film substrates" statement is quite confusing. Why wet the PET film instead of the metal? How is this process achieved? A relevant schematic diagram and description are necessary.
10. Figure 3a and Fig S11 show a significant increase in transferred charge, while the current remains constant. This seems to contradict our expectations, as transferred charge and current should be positively correlated.
11. Many data plots show the same pattern, with the second peak being smaller than the first. Please explain the reason for this.
12. The change in the contact angle of the material after different plasma treatments needs to be supplemented.
13. The strategy of enhancing the output by redox reaction is not desirable, as it will continuously consume copper and greatly limit the durability of the device.
14. Different pH liquids should be measured, and the durability of the device output should also be measured.
15. The authors believe that the origin of the device output in this manuscript is from liquid-solid contact electrification. However, contact electrification is dominated by displacement current, and in this study, the liquid is directly contacted with the metal conductor, in which case the output may be dominated by conduction current.
16. The presence of PET is puzzling. What would happen if PET is removed? Does PET generate electrostatic induction? How will the thickness of PET affect the output? This raises a new question, the relationship between the existence of Figure 1 and Figures 2-4 is confusing.
17. In this manuscript, are the charge transfer carriers electrons and ions? Can the contributions

of these two be distinguished?

18. There are some details problems. It is suggested that the author carefully check the manuscript and make necessary modifications.

Line 198 is missing a period at the end of the sentence.

Line 227 should include the figure caption "Fig. 2g" for clarity.

Replies to Comments

Triboiontronics via temporally controlling electrical double layer formation

Ref. No.: NCOMMS-24-01792

Dear editor and reviewers

We have deeply appreciated all reviewers' helpful comments. Provided below is our detailed response to each comment/suggestion. The specific changes made to the manuscript to address each point are highlighted in yellow.

Responses to Reviewers' Comments:

Reviewer #1 (Remarks to the Author):

The authors report the dynamic regulation of asymmetric electrical double layers to form efficient triboiontronics. The paper is solid and well-written, though, the authors could improve the manuscript with respect to the following comments:

Response: We deeply appreciate to the reviewer's precious suggestions and valuable comments to improve the paper. In response to the reviewer's comments, we have made point-to-point modifications to the manuscript.

1. The caption of Figure 2a should provide details regarding Figure 2a i, ii, iii and iv.

Response: Thank you for your very constructive suggestions. In the latest manuscript, we shifted the order of the descriptions in Fig. 1 and Fig. 2. To further improve the overall readability and clarity of the manuscript, detailed information about the new Figure 1a i-iv (old Figure 2a-iv) was added to the caption.

Corresponding revisions please see the revised caption of Fig. 1a in the manuscript (Page 8, lines 132-138):

“Fig. 1 | The principle of the triboiontronics and the construction of the PDC-TING. a, Triboiontronics via temporally controlling EDL formation. (i) Firstly, when DI water contacted the bottom Au/PET layer, it interacted with the PET substrate through microscopic cracks in the sputtered Au layer, forming a stable EDL. (ii) Secondly, as the top Au/PET layer moved downward to contact with water, the initial CE led to form a new EDL, establishing two EDLs with significantly different symmetries. This created an ion concentration gradient, generating an ionic current I_{i1} . (iii) Thirdly, the detachment of the top Au/PET layer halted ion migration. (iv) Through repeated contact and separation cycles, the ion migration persisted until equilibrium was achieved between the two EDLs.”

Fig. 1 | The principle of the triboiontronics and the construction of the PDC-TING. **a**, Triboiontronics via temporally controlling EDL formation. (i) Firstly, when DI water contacted the bottom Au/PET layer, it interacted with the PET substrate through microscopic cracks in the sputtered Au layer, forming a stable EDL. (ii) Secondly, as the top Au/PET layer moved downward to contact with water, the initial CE led to form a new EDL, establishing two EDLs with significantly different symmetries. This created an ion concentration gradient, generating an ionic current I_{il} . (iii) Thirdly, the detachment of the top Au/PET layer halted ion migration. (iv) Through repeated contact and separation cycles, the ion migration persisted until equilibrium was achieved between the two EDLs. **b**, The model display of the PDC-TING. **c**, Comparison of output performance of different generators. **d**, The effect of different liquid types on the PDC-TING output. **e**, The effect of the extent and direction of the ion concentration gradient on the PDC-TING output. **f**, The effect of the Au-layer sputtering time of Au/PET layers on the PDC-TING output. **g**, The effect of the Au-layer sputtering time of Au/PET layers on the internal resistance of the PDC-TING.

2. Page 10, line 178+, page 13, line 208, water is playing a key-role in such triboiontronics, so quid of the swelling ratio or the water content of the pre-moistening substrate? In addition, quid of dehydration of the substrate over time and its impact on the performances?

Response: We are particularly grateful for your comments. The effect of the extent and direction of asymmetry in the EDL formation on the output of the physically adsorptive direct-current triboiontronic nanogenerator (PDC-TING) was investigated by pre-moistening different gold/polyethylene terephthalate (Au/PET) layers (Fig. 1e and Fig. S5). The specific pre-moistening methods were shown in Fig. S6. When applying 200 μL deionized (DI) water in the PDC-TING with both pristine Au/PET layers sputtered with 1-minute Au, the constructed ion concentration gradient is weaker, resulting in lower positive outputs with I_{SC} (short-circuit current) of 0.97 μA , Q_{SC} (transferred charge) of 59.00 nC, and V_{OC} (open-circuit voltage) of 0.073 V. Pre-moistening was exclusively applied to the bottom Au/PET layer with DI water to pre-form a dense EDL. Subsequently, it could effectively increase the extent of the ion concentration gradient in the PDC-TING, improving the positive output to I_{SC} of 8.20 μA , Q_{SC} of 523.15 nC, and V_{OC} of 0.23 V. On the contrary, the pre-moistening of the top Au/PET layer could pre-built a dense EDL, causing a higher reverse ion concentration gradient in the PDC-TING with negative electrical signals (I_{SC} of -2.75 μA , Q_{SC} of -100.69 nC, and V_{OC} of -0.11 V). When both Au/PET layers were pre-moistened, dense EDLs

formed on their respective surfaces, resulting in a diminished ion concentration gradient within the PDC-TING and weaker positive electrical signals (I_{SC} of 0.45 μA , Q_{SC} of 26.78 nC, and V_{OC} of 0.034 V). Experimental results confirmed that the extent and direction of asymmetry in the EDL formation played a crucial role in determining the efficacy and direction of ion migration, thereby influencing the PDC-TING output. Pre-moistening solely the bottom Au/PET layer could facilitate the establishment of a higher ion concentration gradient, thereby significantly enhancing the ion flux within the PDC-TING.

Furthermore, the influence of the swelling property, water content, and dehydration situation of the Au/PET layer on the PDC-TING output was further investigated through systematic experiments. As indicated by values I_{SC} of about 8.20 μA and Q_{SC} of about 523.15 nC in Fig. S7, the PDC-TING output remained relatively stable regardless of whether the bottom Au/PET layer was pre-moistened, immersed in DI water for 2 hours, or subsequently underwent evaporation and drying at room temperature (approximately 25°C) followed by re-moistening. Hence, the PDC-TING output might be minimally influenced by the swelling property, water content, and dehydration status of the PET substrate. This observation might be attributed to two primary factors. Firstly, the EDL formation at the interface occurs once the Au/PET layer comes into full contact with DI water, largely independent of the water content within the PET substrate. Secondly, the PET film demonstrates low water absorption, typically ranging from 0.06% to 0.129%, resulting in the relative stability of the Au/PET layer in

DI water with minimal swelling or water loss.

Corresponding revisions please see the revised manuscript (Pages 9-10, lines 143-164):

“The effect of the extent and direction of asymmetry in the EDL formation on the PDC-TING output was investigated by pre-moistening different Au/PET layers (Fig. 1e and Fig. S5). The specific pre-moistening methods were shown in Fig. S6. When applying 200 μL water in the PDC-TING with both pristine Au/PET layers sputtered with 1-minute Au, the constructed ion concentration gradient is weaker, resulting in lower positive outputs with I_{SC} of 0.97 μA , Q_{SC} of 59.00 nC, and V_{OC} of 0.073 V. Pre-moistening was exclusively applied to the bottom Au/PET layer with water to pre-form a dense EDL. Subsequently, it could effectively increase the extent of the ion concentration gradient in the PDC-TING, improving the positive output to I_{SC} of 8.20 μA , Q_{SC} of 523.15 nC, and V_{OC} of 0.23 V. On the contrary, the pre-moistening of the top Au/PET layer could pre-built a dense EDL, causing a higher reverse ion concentration gradient in the PDC-TING with negative electrical signals (I_{SC} of -2.75 μA , Q_{SC} of -100.69 nC, and V_{OC} of -0.11 V). When both Au/PET layers were pre-moistened, dense EDLs formed on their respective surfaces, resulting in a diminished ion concentration gradient within the PDC-TING and weaker positive electrical signals (I_{SC} of 0.45 μA , Q_{SC} of 26.78 nC, and V_{OC} of 0.034 V). Experimental results confirmed that the extent and direction of asymmetry in the EDL formation played a crucial role

in determining the efficacy and direction of ion migration, thereby influencing the PDC-TING output. Pre-moistening the bottom Au/PET layer could facilitate establishing a higher ion concentration gradient, significantly enhancing the ion flux within the PDC-TING. Furthermore, as indicated by values I_{sc} of about 8.20 μA and Q_{sc} of about 523.15 nC in Fig. S7, the PDC-TING output remained relatively stable regardless of whether the bottom Au/PET layer was pre-moistened, immersed in DI water for 2 hours, or subsequently underwent evaporation and drying at room temperature (approximately 25°C) followed by re-moistening. The PDC-TING output might be minimally influenced by the swelling property, water content, and dehydration status of the PET substrate.”

Fig. 1 | The principle of the triboiontronics and the construction of the PDC-TING. **a**, Triboiontronics via temporally controlling EDL formation. (i) Firstly, when DI water contacted the bottom Au/PET layer, it interacted with the PET substrate through microscopic cracks in the sputtered Au layer, forming a stable EDL. (ii) Secondly, as the top Au/PET layer moved downward to contact with water, the initial CE led to form a new EDL, establishing two EDLs with significantly different symmetries. This created an ion concentration gradient, generating an ionic current I_{il} . (iii) Thirdly, the detachment of the top Au/PET layer halted ion migration. (iv) Through repeated contact and separation cycles, the ion migration persisted until equilibrium was achieved between the two EDLs. **b**, The model display of the PDC-TING. **c**, Comparison of output performance of different generators. **d**, The effect of different liquid types on the PDC-TING output. **e**, The effect of the extent and direction of the ion concentration gradient on the PDC-TING output. **f**, The effect of the Au-layer sputtering time of Au/PET layers on the PDC-TING output. **g**, The effect of the Au-layer sputtering time of Au/PET layers on the internal resistance of the PDC-TING.

Fig. S5 | The effect of the extent and direction of asymmetry in the EDL formation on the PDC-TING output was investigated by pre-moistening different Au/PET layers. **a**, The comparison of the Q_{sc} . **b**, The comparison of the V_{oc} .

Fig. S6 | The specific pre-moistening methods of pre-moistening different Au/PET layers. a, The top and bottom Au/PET layers were pristine. **b**, Pre-moistening was exclusively applied to the bottom Au/PET layer with DI water to pre-form a dense EDL. **c**, Pre-moistening was exclusively applied to the top Au/PET layer with DI water to pre-form a dense EDL. **d**, The top and bottom Au/PET layers were pre-moistened.

Fig. S7 | The influence of the swelling property, water content, and dehydration situation of the Au/PET layer on the PDC-TING output was further investigated through systematic experiments. a, The I_{sc} of the PDC-TING remained relatively stable at $8.20 \mu\text{A}$. **b,** The Q_{sc} remained relatively stable at 523.15 nC . The PDC-TING output might be minimally influenced by the swelling property, water content, and dehydration status of the PET substrate. This observation might be attributed to two primary factors. Firstly, the EDL formation at the interface occurs once the Au/PET layer comes into full contact with DI water, largely independent of the water content within the PET substrate. Secondly, the PET film demonstrates low water absorption, typically ranging from 0.06% to 0.129%, resulting in the relative stability of the Au/PET layer in DI water with minimal swelling or water loss.

3. Page 10, line 189. the purpose of using ion-free oil, liquid paraffin and glycerol should be introduce within the text for improved clarity.

Response: Thank you for your valuable suggestions. As depicted in Fig. 1d, various substances including oil, liquid paraffin, and 99% glycerol were utilized within the PDC-TING as alternatives to DI water during experimentation. The aim was to further corroborate that the functional mechanism of the PDC-TING involved the directional migration of ions propelled by the ion concentration gradient. Notably, upon applying 200 μL of oil or liquid paraffin onto the pristine bottom Au/PET layer, the absence of ions in these solutions could impede the formation of an ion concentration gradient. The PDC-TING was converted to a conventional solid-liquid triboelectric nanogenerator (TENG) based on the coupling principle of solid-liquid contact electrification (CE) and electrostatic induction. Consequently, lower alternating current (AC) electronics signals were generated solely by electrostatic induction were generated, with I_{SC} of 3.17 nA, Q_{SC} of 0.37 nC, V_{OC} of 1.00 V for oil, and I_{SC} of 3.53 nA, Q_{SC} of 0.30 nC, V_{OC} of 0.85 V for liquid paraffin (Fig. S4a and b). In contrast, when 99% glycerol containing a small number of ions was dropped onto the pristine bottom Au/PET layer, the lower ion concentration gradient was constructed. The PDC-TING generated a weak DC output with I_{SC} of 59.17 nA, Q_{SC} of 3.30 nC, and V_{OC} of 0.042 V (Fig. S4c). Experiments demonstrated that ion migration propelled by the ion concentration gradient might play a pivotal role in the operating

mechanism of the PDC-TING, rather than the coupling of solid-liquid CE and electrostatic induction.

Corresponding revisions please see the revised manuscript (Pages 6-7, lines 122-129):

“Notably, upon applying 200 μL of oil or liquid paraffin onto the pristine bottom Au/PET layer, the absence of ions in these solutions could impede the formation of an ion concentration gradient. Lower AC displacement electrical signals were generated by the CE and electrostatic induction, with I_{SC} of 3.17 nA, Q_{SC} of 0.37 nC, V_{OC} of 1.00 V for oil, and I_{SC} of 3.53 nA, Q_{SC} of 0.30 nC, V_{OC} of 0.85 V for liquid paraffin (Fig. 1d and Fig. S4). In contrast, under the 99% glycerol containing a small number of ions, the weak DC output was generated (I_{SC} of 59.17 nA, Q_{SC} of 3.30 nC, and V_{OC} 0.042 V). Experiments demonstrated that ion migration propelled by the ion concentration gradient might play a pivotal role in the operating mechanism of the PDC-TING.”

Fig. 1 | The principle of the triboiontronics and the construction of the PDC-TING. **a**, Triboiontronics via temporally controlling EDL formation. (i) Firstly, when DI water contacted the bottom Au/PET layer, it interacted with the PET substrate through microscopic cracks in the sputtered Au layer, forming a stable EDL. (ii) Secondly, as the top Au/PET layer moved downward to contact with water, the initial CE led to form a new EDL, establishing two EDLs with significantly different symmetries. This created an ion concentration gradient, generating an ionic current I_{il} . (iii) Thirdly, the detachment of the top Au/PET layer halted ion migration. (iv) Through repeated contact and separation cycles, the ion migration persisted until equilibrium was achieved between the two EDLs. **b**, The model display of the PDC-TING. **c**, Comparison of output performance of different generators. **d**, The effect of different liquid types on the PDC-TING output. **e**, The effect of the extent and direction of the ion concentration gradient on the PDC-TING output. **f**, The effect of the Au-layer sputtering time of Au/PET layers on the PDC-TING output. **g**, The effect of the Au-layer sputtering time of Au/PET layers on the internal resistance of the PDC-TING.

Fig. S4 | The effect of different liquid types on the PDC-TING output was investigated. a, The comparison of the Q_{sc} . **b,** The comparison of the V_{oc} . **c,** The detailed display of the V_{oc} during the directional migration of ions in the PDC-TING with 99% glycerol.

Finally, we would like to thank the reviewer again for these valuable comments as well as the thoughtful and careful review towards improving our manuscript.

Reviewer #2 (Remarks to the Author):

Triboiontronic nanogenerators represents an interesting approach for energy harvesting applications, especially in the low frequency range and for energy storage. I interpret this approach as a combination of traditional TENGs and energy harvested from ion concentration/salinity gradients. This study investigates parameters and properties that affects output characteristics of triboiontronic nanogenerators. Charge collecting layer, operating frequency, and salt concentration is a few parameters in focus of this study that highly influence the performance.

Response: We are grateful to the reviewer for the thorough review and help to improve the paper. In response to the reviewer's precious suggestions and valuable comments, we have made point-to-point modifications to the Manuscript.

1. I find the term “asymmetric EDL” a bit unclear as it is very open for interpretation. My interpretation is that the asymmetry is mostly based on a not yet fully formed/charged EDL on the upper substrate, originated from a difference in concentration gradient. But from the word asymmetric, that could be material properties or device geometries for example. Therefore, I would consider interpreting and naming this approach as “temporally controlling EDL formation” or similar, rather than a “regulation of asymmetric EDL”.

Response: Thank you for the very helpful suggestions. Accordingly, based on a meticulous reassessment of the fundamental mechanism underlying our research and taking into account the valuable feedback provided by the reviewer, we have determined that “temporally controlling EDL formation” more precisely reflects the mechanism of triboiontronics. Consequently, we have revised the title of the article to “Triboiontronics via temporally controlling electrical double layer formation” and have made corresponding detailed adjustments to elucidate the formation mechanism of triboiontronics, as outlined below.

The EDL functioned as an exceptional interface for ionic-electronic coupling between the solid dielectric and liquid within the "two-step" model. By dynamically regulating the EDL, iontronics enabled precise control over ion migration and electron coupling transfer at the interface. Here, through temporally controlling EDL formation at solid-liquid interfaces, efficient triboiontronics was established (Fig. 1a). Firstly, a metal layer (Au) as a charge-

collecting layer was sputtered onto a dielectric substrate (polyethylene terephthalate, PET). Upon contact of DI water with the bottom Au/PET layer, it might engage in direct interaction with the PET substrate via microscopic cracks in the sputtered Au layer, leading to the solid-liquid CE and the establishment of a stable EDL [Fig. 1a(i)]. Secondly, as the top Au/PET layer moved downwards to contact with DI water, the initial CE led to the formation of a new EDL, thus establishing two EDLs with significantly different symmetries [Fig. 1a(ii)]. This created an ion concentration gradient, facilitating electron transfer in the external circuit and generating an ionic current I_{i1} . Thirdly, the detachment of the top Au/PET layer halted ion migration [Fig. 1a(iii)]. Through repeated contact and separation cycles, the constructed ion concentration gradient gradually weakened, yet the ion migration persisted until equilibrium was achieved between the two EDLs [Fig. 1a(iv)].

Please check the revised manuscript for corresponding revisions (Page 5, lines 92-105):

“The EDL functioned as an exceptional interface for ionic-electronic coupling between the solid dielectric and liquid within the "two-step" model. By dynamically regulating the EDL, iontronics enabled precise control over ion migration and electron coupling transfer at the interface. Here, through temporally controlling EDL formation at solid-liquid interfaces, efficient triboiontronics was established (Fig. 1a). Firstly, a metal layer (Au) as a charge-

collecting layer was sputtered onto a dielectric substrate (polyethylene terephthalate, PET). Upon contact of deionized (DI) water with the bottom Au/PET layer, it might engage in direct interaction with the PET substrate via microscopic cracks in the sputtered Au layer, leading to the solid-liquid CE and the establishment of a stable EDL [Fig. 1a(i)]. Secondly, as the top Au/PET layer moved downwards to contact with DI water, the initial CE led to the formation of a new EDL, thus establishing two EDLs with significantly different symmetries [Fig. 1a(ii)]. This created an ion concentration gradient, facilitating electron transfer in the external circuit and generating an ionic current I_{ii} . Thirdly, the detachment of the top Au/PET layer halted ion migration [Fig. 1a(iii)]. Through repeated contact and separation cycles, the constructed ion concentration gradient gradually weakened, yet the ion migration persisted until equilibrium was achieved between the two EDLs [Fig. 1a(iv)].”

Fig. 1 | The principle of the triboiontronics and the construction of the PDC-TING. **a**, Triboiontronics via temporally controlling EDL formation. (i) Firstly, when DI water contacted the bottom Au/PET layer, it interacted with the PET substrate through microscopic cracks in the sputtered Au layer, forming a stable EDL. (ii) Secondly, as the top Au/PET layer moved downward to contact with water, the initial CE led to form a new EDL, establishing two EDLs with significantly different symmetries. This created an ion concentration gradient, generating an ionic current I_{i1} . (iii) Thirdly, the detachment of the top Au/PET layer halted ion migration. (iv) Through repeated contact and separation cycles, the ion migration persisted until equilibrium was achieved between the two EDLs. **b**, The model display of the PDC-TING. **c**, Comparison of output performance of different generators. **d**, The effect of different liquid types on the PDC-TING output. **e**, The effect of the extent and direction of the ion concentration gradient on the PDC-TING output. **f**, The effect of the Au-layer sputtering time of Au/PET layers on the PDC-TING output. **g**, The effect of the Au-layer sputtering time of Au/PET layers on the internal resistance of the PDC-TING.

2. I find the comparison with TENG and previous TINGs is ambitious, yet a bit difficult to validate. From the perspective that the energy here can represent a combination of triboelectric energy and Gibbs free energy/chemical potential from the salinity gradient, I would really appreciate an analysis of this from the authors, and additional references from the chemical potential perspective. I note that the highest values are recorded at the highest concentration. The situation might be clearer with a more analytical description of the equation used. Can the numbers be divided into different sources of contribution?

Response: Thank you for your invaluable comments. The synergistic effect of temporally controlling EDL formation and redox reaction could effectively promote the coupling migration of ions, thereby enhancing the ion flux. In particular, under the synergistic effect, the pure energy harvesting device PDC-TING was converted to an integrated energy harvesting and storage device SDC-TING. To distinguish the contribution of triboelectric energy, Gibbs free energy from the ion concentration gradient, and chemical potential to the SDC-TING output, we first determined the impact of triboelectric energy. When 200 μL of DI water was dispensed onto the PTFE film adhered to the bottom Au/PET layer (Fig. S1c), the PDC-TING transitioned into a conventional solid-liquid TENG, operating on the coupling principle of the CE and electrostatic induction. It generated AC displacement electrical signals (I_{sc} of 3.5 nA and Q_{sc} of 0.32 nC in Fig. 1c and Fig. S2a), significantly smaller than those of the DC ionic current

signals (I_{SC} of 5.58 mA and Q_{SC} of 54.07 μC) produced by the SDC-TING (Fig. 4c and d). This suggested that the contribution of triboelectric energy to the SDC-TING output should be negligible.

In an environment with an ion concentration gradient, the variation in chemical potential or Gibbs free energy from the ion concentration gradient is analogous to the free energy change in electron transfer reactions. Marcus theory, primarily employed to elucidate electron transfer reactions, can be extrapolated to encompass other chemical processes. According to Marcus theory, the rate of electron transfer is governed by two primary factors: inner sphere reorganization and outer sphere reorganization. Inner sphere reorganization pertains to the movement of electrons between the electron donor and acceptor, influenced by the rearrangement of atomic nuclei during the electron transfer process. Outer sphere reorganization refers to the impact of solvent or surrounding molecules on electron transfer reactions, such as solvent polarity and the arrangement of solvent molecules. According to Marcus theory, it can be inferred that increasing ion concentration gradient may induce changes in chemical potential, consequently impacting the rate of chemical reactions and the thermodynamic stability of the system.

The relationship between electron transfer rate (k) and Gibbs free energy change (ΔG) can be expressed by the following equation:

$$k = k_0 e^{\left(\frac{-\Delta G}{4k_B T}\right)} \quad (\text{R1})$$

where k_0 is the pre-exponential factor, k_B is the Boltzmann constant, and T is the

temperature. ΔG is the Gibbs free energy change, which can be represented by the chemical potential difference Δu :

$$\Delta G = -zF\Delta u \quad (\text{R2})$$

where z is the number of charges in the reaction and F is the Faraday constant. In an ion concentration gradient environment, the relationship between the $\Delta\mu$ and the ion concentration gradient can be expressed by the Nernst equation:

$$\Delta u = RT \ln \left(\frac{a_2}{a_1} \right) \quad (\text{R3})$$

where R is the ideal gas constant, and a_1 and a_2 represent the activity of ions in the solution. The relationship between the Δu and the concentration of ions c can be derived from the activity of ions:

$$\Delta u = RT \ln \left(\frac{c_2}{c_1} \right) \quad (\text{R4})$$

In summary, an increase in ion concentration gradient might lead to an increase in chemical potential difference, thereby affecting the change in Gibbs free energy, and consequently impacting the electron transfer rate.

As shown in Fig. 4d, the optimal Q_{SC} of the SDC-TING under the synergistic effect of temporally controlling EDL formation and redox reaction was 54.07 μC . After the disappearance of the ion concentration gradient caused by temporally controlling EDL formation, its Q_{SC} was 10.10 μC solely under the redox reaction (Fig. S38b). This indicated that the Q_{SC} of 43.97 μC in the SDC-TING might be due to Gibbs free energy from the ion concentration gradient. Therefore, the contribution of Gibbs free energy from the ion concentration gradient and

chemical potential difference to the Q_{SC} of the SDC-TING during the initial contact was 81.32% and 18.68%, respectively.

Corresponding revisions please see the revised manuscript (Page 6, lines 118-121):

“When DI water was dispensed onto the PTFE film adhered to the bottom Au/PET surface (Fig. S1c), the ion migration was hindered, and the conventional solid-liquid TENG was constructed. Weak AC electronics signals were generated only by electrostatic induction with I_{SC} of 3.53 nA, Q_{SC} of 0.32 nC Q_{SC} , and V_{OC} of 0.48 V.”

Corresponding revisions please see the revised Supplementary Note 1 in the Supplementary Information (Pages 2-3, lines 15-43):

“In an environment with an ion concentration gradient, the variation in chemical potential or Gibbs free energy from the ion concentration gradient is analogous to the free energy change in electron transfer reactions. Marcus theory, primarily employed to elucidate electron transfer reactions, can be extrapolated to encompass other chemical processes. According to Marcus theory, the rate of electron transfer is governed by two primary factors: inner sphere reorganization and outer sphere reorganization. Inner sphere reorganization pertains to the movement of electrons between the electron donor and acceptor, influenced by the rearrangement of atomic nuclei during the electron transfer process. Outer

sphere reorganization refers to the impact of solvent or surrounding molecules on electron transfer reactions, such as solvent polarity and the arrangement of solvent molecules. According to Marcus theory, it can be inferred that increasing ion concentration gradient may induce changes in chemical potential, consequently impacting the rate of chemical reactions and the thermodynamic stability of the system.

The relationship between electron transfer rate (k) and Gibbs free energy change (ΔG) can be expressed by the following equation:

$$k = k_0 e^{\left(\frac{-\Delta G}{4k_B T}\right)} \quad (\text{S1})$$

where k_0 is the pre-exponential factor, k_B is the Boltzmann constant, and T is the temperature. ΔG is the Gibbs free energy change, which can be represented by the chemical potential difference Δu :

$$\Delta G = -zF\Delta u \quad (\text{S2})$$

where z is the number of charges in the reaction and F is the Faraday constant. In an ion concentration gradient environment, the relationship between the Δu and the ion concentration gradient can be expressed by the Nernst equation:

$$\Delta u = RT \ln\left(\frac{a_2}{a_1}\right) \quad (\text{S3})$$

where R is the ideal gas constant, and a_1 and a_2 represent the activity of ions in the solution. The relationship between the Δu and the concentration of ions c can be derived from the activity of ions:

$$\Delta u = RT \ln\left(\frac{c_2}{c_1}\right) \quad (\text{S4})$$

In summary, an increase in ion concentration gradient might lead to an increase in chemical potential difference, thereby affecting the change in Gibbs free energy, and consequently impacting the electron transfer rate.”

Corresponding revisions please see the revised manuscript (Page 21, lines 358-360):

“After comparison, the contribution of temporally controlling EDL formation and redox reaction on the Q_{sc} of the SDC-TING during the initial contact was 81.32% and 18.68%, respectively.”

Fig. S1 | Schematic and physical diagrams of different generators. **a**, When 200 μL deionized (DI) water was dropped on the pristine bottom gold/polyethylene terephthalate (Au/PET) surface, the physically adsorbed direct-current triboiontronic nanogenerator (PDC-TING) was constructed. **b**, When the polytetrafluoroethylene (PTFE) film was used to replace DI water and attached to the bottom Au/PET surface, the conventional solid-solid triboelectric nanogenerator (TENG) was constructed. **c**, When 200 μL DI water was dropped on the PTFE film attached to the bottom Au/PET surface, the conventional solid-liquid TENG was constructed.

Fig. 1 | The principle of the triboiontronics and the construction of the PDC-TING. **a**, Triboiontronics via temporally controlling EDL formation. (i) Firstly, when DI water contacted the bottom Au/PET layer, it interacted with the PET substrate through microscopic cracks in the sputtered Au layer, forming a stable EDL. (ii) Secondly, as the top Au/PET layer moved downward to contact with water, the initial CE led to form a new EDL, establishing two EDLs with significantly different symmetries. This created an ion concentration gradient, generating an ionic current I_{i1} . (iii) Thirdly, the detachment of the top Au/PET layer halted ion migration. (iv) Through repeated contact and separation cycles, the ion migration persisted until equilibrium was achieved between the two EDLs. **b**, The model display of the PDC-TING. **c**, Comparison of output performance of different generators. **d**, The effect of different liquid types on the PDC-TING output. **e**, The effect of the extent and direction of the ion concentration gradient on the PDC-TING output. **f**, The effect of the Au-layer sputtering time of Au/PET layers on the PDC-TING output. **g**, The effect of the Au-layer sputtering time of Au/PET layers on the internal resistance of the PDC-TING.

Fig. S2 | Comparison of output performance of different generators. **a**, The comparison of the transferred charge (Q_{sc}). **b**, The comparison of the open-circuit voltage (V_{oc}). **c**, The detailed display of the V_{oc} during the directional migration of ions in the PDC-TING.

Fig. 4 | More efficient triboiontronics via the synergistic effect of temporally controlling EDL formation and redox reaction. **a**, The synergistic principle of temporally controlling EDL formation and redox reaction in the SDC-TING. **b**, The verification experiment of their synergistic effect. **c**, **d**, and **e**, The I_{sc} density, Q_{sc} density, and P_R density of the SDC-TING. **f**, The comparison of the Q_{sc} density of the conventional solid-liquid TENGs, solid-solid TENGs, DC-TING, PDC-TING, and SDC-TING. **g**, The V_{OC} of the SDC-TINGs was improved by the 10-node connecting in series. **h**, Without the assistance of the rectifier bridge, the SDC-TINGs in series could directly supply energy to electrical devices. **i**, In the information flow field, the sensitivity of the PDC-TING/SDC-TING to external stimuli has facilitated the development of self-powered sensing systems.

Fig. S38 | After the disappearance of the ion concentration gradient caused by temporally controlling EDL formation, the PDC-TING output depended solely on the redox reaction. a, The I_{sc} was 1.01 mA. b, The Q_{sc} was 10.10 μC . c, The V_{oc} was 0.18 V. d, The P_R was 0.13 mW.

3. The previous work of the authors, referenced in 21. Li, X. et al. *Triboiontronics for efficient energy and information flow*. *Matter* 6, 3912-3926 (2023), is closely related. Here, peak power density is highlighted as the advancement that is outstanding in relation to previous work. In the current manuscript, the peak power density is lower, and the transferred charge density is higher. For what applications would either property be most important?

Response: Thanks for your comments and constructive suggestions. The DC-TING^[21] utilized the charge density of the diffuse layer within the EDL on the dielectric surface to create an ion concentration gradient, driving ion migration and adjusting the electronic displacement current to produce an ionic-electronic coupling current. While it offered impressive peak power (P_R) density characteristics, its high internal resistance (ten-mega Ohm level) limited the generated Q_{SC} . This configuration is best suited for applications requiring instant high-power output with lower overall energy consumption, such as flash lamps and pulsed lasers.

The PDC-TING and SDC-TING, built upon efficient ion migration propelled by the ion concentration gradient outlined in this study, exhibited several notable advantages. These included higher current and lower internal resistance (in the hundred Ohm level), leading to a conventional P_R density but a higher Q_{SC} density compared to their counterparts. As a result, they are better suited for applications demanding long-term, continuous supply of stable energy

output, such as small portable medical devices or monitoring devices (smartwatches, etc.).

Corresponding revisions please see the revised manuscript (Page 23, lines 283-288):

“DC-TING^[21] supplemented charge density within the diffuse layer on the dielectric surface to create an ion concentration gradient, driving ion migration and adjusting electronic displacement current for efficient ionic-electronic coupling. It had ultra-high power density characteristics, but its higher internal resistance restricts Q_{sc} . In contrast, PDC-TING and SDC-TING relied on efficient ion migration driven by concentration gradients, resulting in lower internal resistance (in the hundred-Ohm range) and ultra-high ion flux.”

4. I also lack a more thorough description of the addition of redox reactions, and method description of experiment 4a and b. What voltage was applied to drive the redox reactions? Apologies if this information is there but missed.

Response: Thank you for your feedback. The modification of the charge-collecting layer on the PET dielectric substrate, altering the metal type, facilitated a more efficient SDC-TING (Fig. 4a and Fig. S35). Upon contact between the top Au/PET layer and water, alongside the fully contacted bottom Cu/PET layer, an ion concentration gradient was established. The divergent electrochemical behaviors of Au and Cu in water prompted redox reactions on their surfaces. Cu, with higher electrochemical activity, underwent oxidation, releasing electrons into the water to generate Cu^{2+} ions, as represented by:

Simultaneously, the Au surface accepted these electrons and possibly reduced O_2 in the water to form OH^- . Its equation might be expressed as:

The cyclic voltammetry curve depicted in Fig. S36 served as verification that the redox reaction indeed occurs. Introducing this redox reaction enabled the production of more Cu^{2+} ions. Their directional migration, driven by the chemical potential difference, effectively boosts the ion flux.

The study systematically investigated the synergistic effect of controlling EDL formation and redox reaction, as illustrated in Fig. 4b. Initially, pre-

moistening the bottom Cu/PET layer individually with water established a dense EDL. Upon initial contact of the top Au/PET layer with water, the combined ion concentration gradient and chemical potential difference drove upward ion migration, notably increasing the I_{SC} to 16.82 μA . After about 50 seconds, the I_{SC} stabilized at 4.80 μA under the chemical potential difference alone. Next, pre-moistening the top Au/PET layer individually led to a reverse ion concentration gradient upon contact with water. This opposed the chemical potential difference, resulting in a decrease in I_{SC} to 1.33 μA , with the gradient disappearing after about 20 seconds. Finally, pre-moistening both layers constructed a weaker positive ion concentration gradient. This, combined with the chemical potential difference, contributed to an I_{SC} of 5.77 μA for PDC-TING, with synergy lasting 15 seconds. These findings highlighted the effective promotion of coupling migration of ions and enhancement of ion flux through the synergistic effect of temporally controlling EDL formation and redox reaction.

Corresponding revisions please see the revised manuscript (Pages 19-21, lines 324-348):

“The modification of the charge-collecting layer on the PET dielectric substrate, altering the metal type, facilitated a more efficient SDC-TING (Fig. 4a and Fig. S35). Upon contact between the top Au/PET layer and water, alongside the fully contacted bottom Cu/PET layer, an ion concentration gradient was established. The divergent electrochemical behaviors of Au and Cu in water

prompted redox reactions on their surfaces. Cu, with higher electrochemical activity, underwent oxidation, releasing electrons into the water to generate Cu^{2+} ions, as represented by:

Simultaneously, the Au surface accepted these electrons and possibly reduced O_2 in the water to form OH^- . Its equation might be expressed as:

The cyclic voltammetry curve depicted in Fig. S36 served as verification that the redox reaction indeed occurs. Introducing this redox reaction enabled the production of more Cu^{2+} ions. Their directional migration, driven by the chemical potential difference, effectively boosts the ion flux. The synergistic effect of temporally controlling EDL formation and redox reaction was systematically investigated (Fig. 4b). Initially, pre-moistening the bottom Cu/PET layer individually with water established a dense EDL. Upon initial contact of the top Au/PET layer with water, the combined ion concentration gradient and chemical potential difference drove upward ion migration, notably increasing the I_{SC} to $16.82 \mu\text{A}$. After about 50 seconds, the I_{SC} stabilized at $4.80 \mu\text{A}$ under the chemical potential difference alone. Next, pre-moistening the top Au/PET layer individually led to a reverse ion concentration gradient upon contact with water. This opposed the chemical potential difference, resulting in a decrease in I_{SC} to $1.33 \mu\text{A}$, with the gradient disappearing after about 20 seconds. Finally, pre-moistening both layers constructed a weaker positive ion concentration gradient.

This, combined with the chemical potential difference, contributed to an I_{SC} of 5.77 μA for PDC-TING, with synergy lasting 15 seconds. These findings highlighted the effective promotion of coupling migration of ions and enhancement of ion flux through the synergistic effect of temporally controlling EDL formation and redox reaction.”

Fig. 4 | More efficient triboiontronics via the synergistic effect of temporally controlling EDL formation and redox reaction. **a**, The synergistic principle of temporally controlling EDL formation and redox reaction in the SDC-TING. **b**, The verification experiment of their synergistic effect. **c**, **d**, and **e**, The I_{sc} density, Q_{sc} density, and P_R density of the SDC-TING. **f**, The comparison of the Q_{sc} density of the conventional solid-liquid TENGs, solid-solid TENGs, DC-TING, PDC-TING, and SDC-TING. **g**, The V_{OC} of the SDC-TINGs was improved by the 10-node connecting in series. **h**, Without the assistance of the rectifier bridge, the SDC-TINGs in series could directly supply energy to electrical devices. **i**, In the information flow field, the sensitivity of the PDC-TING/SDC-TING to external stimuli has facilitated the development of self-powered sensing systems.

Fig. S35 | The operation principle of the SDC-TENG. **a**, Firstly, the DI water fully contacted the bottom Cu/PET layer, resulting in the formation of a denser EDL. **b**, Secondly, as the top Au/PET layer moved downwards to contact with DI water, the initial CE led to a new EDL, establishing two EDLs with significantly different symmetries. The ion concentration gradient formed by temporally controlling EDL formation and the chemical potential difference generated by the redox reaction could produce a synergistic effect, thereby jointly promoting the rapid directional migration of ions and generating an efficient ionic current I_{15} . **c**, Thirdly, the detachment of the top Au/PET layer from the DI water halted ion migration **d**, After multiple contact and separation cycles, the ion concentration gradient formed by temporally controlling EDL formation disappeared, and the chemical potential difference generated by the redox reaction could continue to drive the directional migration of ions, resulting in a lower stable current output.

Fig. S36 | In 1 M LiCl solution, the cyclic voltammery curves between the two Au/PET layers and between the Au/PET layer and Cu/PET layer were compared.

5. When discussing the collecting layer thickness, would the authors like to comment on the drop in internal resistance from 1 to 10 min? Could not just thickness but also coverage/pinholes play a role? Here, microscopy images would be interesting, but not necessary. I would however consider moving some images from SI to main text.

Response: Thank you for your constructive suggestions. Increasing the sputtering time of Au on the PET substrate reduced the internal resistance of the PDC-TING from 10.6 k Ω to 7.4 k Ω (Fig. 1g). Scanning electron microscope (SEM) observations revealed microscopic cracks on the surfaces of PET film and Au/PET film sputtered for 1 minute (Fig. S11a and b), indicating insufficient Au deposition within 1 minute, leading to higher internal resistance (10.6 k Ω). With 10 minutes of sputtering, the cracks notably diminished (Fig. S11c), suggesting improved Au coverage and enhanced electrical conductivity, resulting in reduced internal resistance (7.6 k Ω). Further extending sputtering to more than 20 minutes nearly eliminated the cracks, maintaining internal resistance at around 7.4 k Ω (Fig. S11d and e). Therefore, adjusting the sputtering time of the metal enabled control over its coverage on the dielectric substrate, thereby affecting the charge-collecting layer's conductivity.

Corresponding revisions please see the revised manuscript (Page 10, lines 174-184):

“In addition, increasing the sputtering time of Au on the PET substrate reduced the internal resistance of the PDC-TING from 10.6 k Ω to 7.4 k Ω (Fig. 1g). Scanning electron microscope (SEM) observations revealed microscopic cracks on the surfaces of PET film and Au/PET film sputtered for 1 minute (Fig. S11a and b), indicating insufficient Au deposition within 1 minute, leading to higher internal resistance (10.6 k Ω). With 10 minutes of sputtering, the cracks notably diminished (Fig. S11c), suggesting improved Au coverage and enhanced electrical conductivity, resulting in reduced internal resistance (7.6 k Ω). Further extending sputtering to more than 20 minutes nearly eliminated the cracks, maintaining internal resistance at around 7.4 k Ω (Fig. S11d and e). With the increase in sputtering time, the decrease in transmittance of Au/PET films also proved the increase of Au coverage on the PET substrate (Fig. S12). Therefore, adjusting the sputtering time of the metal enabled control over its coverage on the dielectric substrate, thereby affecting the charge-collecting layer's conductivity.”

Fig. 1 | The principle of the triboiontronics and the construction of the PDC-TING. **a**, Triboiontronics via temporally controlling EDL formation. (i) Firstly, when DI water contacted the bottom Au/PET layer, it interacted with the PET substrate through microscopic cracks in the sputtered Au layer, forming a stable EDL. (ii) Secondly, as the top Au/PET layer moved downward to contact with water, the initial CE led to form a new EDL, establishing two EDLs with significantly different symmetries. This created an ion concentration gradient, generating an ionic current I_{il} . (iii) Thirdly, the detachment of the top Au/PET layer halted ion migration. (iv) Through repeated contact and separation cycles, the ion migration persisted until equilibrium was achieved between the two EDLs. **b**, The model display of the PDC-TING. **c**, Comparison of output performance of different generators. **d**, The effect of different liquid types on the PDC-TING output. **e**, The effect of the extent and direction of the ion concentration gradient on the PDC-TING output. **f**, The effect of the Au-layer sputtering time of Au/PET layers on the PDC-TING output. **g**, The effect of the Au-layer sputtering time of Au/PET layers on the internal resistance of the PDC-TING.

Fig. S11 | The surface morphology of the different films was observed using a scanning electron microscope (SEM). **a**, PET film. **b** Au/PET layer with the 1-minute sputtering time. **c**, Au/PET layer with the 10-minute sputtering time. **d**, Au/PET layer with the 20-minute sputtering time. **e**, Au/PET layer with the 30-minute sputtering time.

6. The text is well organized and easy to follow. The introduction and conclusion sections are especially well conducted, and well summarized. Plenty of data is provided, and the SI is very thorough and a great support for the study.

Response: Thank you very much for your positive feedback and valuable comments that will help us to improve the paper.

7. Again, I find this work and approach very interesting! The energy storage perspective (and information flow) could be highlighted a bit more. Some more analysis and descriptions/discussion is needed for me to properly validate the comparison presented in the graphical abstract and Fig 4f. Figure 4i is motivating, but I would appreciate more discussion/validation and comments on how this method would be realised.

Response: Thanks for your very valuable advice. The introduction of the redox reaction facilitated the generation of additional Cu^{2+} ions. Their directional migration, driven by the chemical potential difference, effectively boosts the ion flux. The synergistic effect of temporally controlling EDL formation and redox reaction was systematically investigated (Fig. 4b). It highlighted the effective promotion of coupling migration of ions and enhancement of ion flux through the synergistic effect of temporally controlling EDL formation and redox reaction. In particular, under the synergistic effect, the pure energy harvesting device PDC-TING was converted to an integrated energy harvesting and storage device SDC-TING. Utilizing the substantial Cu^{2+} ion flux generated by the redox reaction, the SDC-TING could enhance its output performance and operational stability simultaneously. With an internal resistance of 500Ω , it achieved the I_{SC} of 5.58 mA (Fig. 4c), Q_{SC} of 54.07 μC (Fig. 4d), P_{R} of 3.86 mW (Fig. 4e), and V_{OC} of 0.30 V (Fig. S37). The I_{SC} density, Q_{SC} density, and P_{R} density could reach up to 55.77 A/m^2 , 540.70 mC/m^2 , and 38.64 W/m^2 . The efficient SDC-TING output not

only increased by 452.48%, 435.35%, and 2869.23% respectively compared to that completely depending on the redox reaction (Fig. S38), but also increased by 114.62%, 31.08%, and 354.12% respectively compared to the PDC-TING output. Comparative analysis with the PDC-TING revealed significantly improved stability under the synergistic effect (Fig. S39). After comparison, the contribution of temporally controlling EDL formation and redox reaction on the Q_{SC} of the SDC-TING during the initial contact was 81.32% and 18.68%, respectively.

Compared with the Q_{SC} densities of hydrovoltaic technology by moving the EDL boundary^[7, 28, 30], conventional solid-liquid TENGs^[8, 31-33] and solid-solid TENGs^[34-49], or DC-TING^[21], that of PDC-TING or SDC-TING was increased by several orders of magnitude (Fig. 4f). Hydrovoltaic technology^[7, 28, 30] typically manipulated the EDL boundary on the charge-collecting layer of the dielectric substrate, often resulting in limited ion concentration gradients and increased migration distances, which restricted overall output. Conventional solid-liquid TENG^[8, 31-33] and solid-solid TENG^[34-49] utilized triboelectric charges on dielectric materials to induce electronic displacement currents. The insulation properties of dielectrics usually led to higher internal resistance^[50] (in the ten mega-Ohm range), enabling higher V_{OC} but limiting both I_{SC} and Q_{SC} . DC-TING^[21] supplemented charge density within the diffuse layer on the dielectric surface to create an ion concentration gradient, driving ion migration and adjusting electronic displacement current for efficient ionic-electronic coupling.

It had ultra-high power density characteristics, but its higher internal resistance restricts Q_{sc} . In contrast, PDC-TING and SDC-TING relied on efficient ion migration driven by concentration gradients, resulting in lower internal resistance (in the hundred-Ohm range) and ultra-high ion flux.

In practical applications, temporally controlling EDL formation and redox reaction presented a promising avenue for enhancing energy and information flow (Fig. 4i). Regarding information flow, the sensitivity of PDC-TING and SDC-TING to external stimuli has spurred the development of self-powered sensing systems. These systems offer a tunable ionic-electronic coupling interface, serving as precise probes for investigating the dynamic properties of the EDL and its impact on electrochemical reactions, charge transport, and energy storage processes. Additionally, by regulating EDL formation, PDC-TING and SDC-TING enable controlled migration and distribution of ions such as Na^+ , K^+ , Cl^- , and Ca^{2+} . In biological systems, the transport of these ions plays a crucial role in various life activities, including cellular communication, nerve impulse transmission, and muscle contraction. This capability mimics ionic flow in biological systems, opening avenues for seamless integration between electronic devices and biological processes. Potential applications include biosensors for real-time monitoring of biochemical processes, brain-computer interfaces for enhanced communication, and neuromorphic computing inspired by brain neural networks.

Corresponding revisions please see the revised manuscript (Pages 20-21, lines 346-360):

“These findings highlighted the effective promotion of coupling migration of ions and enhancement of ion flux through the synergistic effect of temporally controlling EDL formation and redox reaction. In particular, under the synergistic effect, the pure energy harvesting device PDC-TING was converted to an integrated energy harvesting and storage device SDC-TING. Utilizing the substantial Cu^{2+} ion flux generated by the redox reaction, the SDC-TING could enhance its output performance and operational stability simultaneously. With an internal resistance of $500\ \Omega$, it achieved the I_{SC} of 5.58 mA (Fig. 4c), Q_{SC} of 54.07 μC (Fig. 4d), P_{R} of 3.86 mW (Fig. 4e), and V_{OC} of 0.30 V (Fig. S37). The I_{SC} density, Q_{SC} density, and P_{R} density could reach up to 55.77 A/m^2 , 540.70 mC/m^2 , and 38.64 W/m^2 . The efficient SDC-TING output not only increased by 452.48%, 435.35%, and 2869.23% respectively compared to that completely depending on the redox reaction (Fig. S38), but also increased by 114.62%, 31.08%, and 354.12% respectively compared to the PDC-TING output. Comparative analysis with the PDC-TING revealed significantly improved stability under the synergistic effect (Fig. S39). After comparison, the contribution of temporally controlling EDL formation and redox reaction on the QSC of the SDC-TING during the initial contact was 81.32% and 18.68%, respectively.”

Corresponding revisions please see the revised manuscript (Page 23, lines

375-388):

“Compared with the Q_{sc} densities of hydrovoltaic technology by moving the EDL boundary^[7, 28, 30], conventional solid-liquid TENGs^[8, 31-33] and solid-solid TENGs^[34-49], or DC-TING^[21], that of PDC-TING or SDC-TING was increased by several orders of magnitude (Fig. 4f). Hydrovoltaic technology^[7, 28, 30] typically manipulated the EDL boundary on the charge-collecting layer of the dielectric substrate, often resulting in limited ion concentration gradients and increased migration distances, which restricted overall output. Conventional solid-liquid TENG^[8, 31-33] and solid-solid TENG^[34-49] utilized triboelectric charges on dielectric materials to induce electronic displacement currents. The insulation properties of dielectrics usually led to higher internal resistance^[50] (in the ten mega-Ohm range), enabling higher V_{oc} but limiting both I_{sc} and Q_{sc} . DC-TING^[21] supplemented charge density within the diffuse layer on the dielectric surface to create an ion concentration gradient, driving ion migration and adjusting electronic displacement current for efficient ionic-electronic coupling. It had ultra-high power density characteristics, but its higher internal resistance restricts Q_{sc} . In contrast, PDC-TING and SDC-TING relied on efficient ion migration driven by concentration gradients, resulting in lower internal resistance (in the hundred-Ohm range) and ultra-high ion flux.”

Corresponding revisions please see the revised manuscript (Page 24, lines 397-407):

“Regarding information flow, the sensitivity of PDC-TING and SDC-TING to external stimuli has spurred the development of self-powered sensing systems (Fig. 4i). These systems offer a tunable ionic-electronic coupling interface, serving as precise probes for investigating the dynamic properties of the EDL and its impact on electrochemical reactions, charge transport, and energy storage processes. Additionally, by regulating EDL formation, PDC-TING and SDC-TING enable controlled migration and distribution of ions such as Na^+ , K^+ , Cl^- , and Ca^{2+} . In biological systems, the transport of these ions plays a crucial role in various life activities, including cellular communication, nerve impulse transmission, and muscle contraction. This capability mimics ionic flow in biological systems, opening avenues for seamless integration between electronic devices and biological processes. Potential applications include biosensors for real-time monitoring of biochemical processes, brain-computer interfaces for enhanced communication, and neuromorphic computing inspired by brain neural networks.”

Fig. 4 | More efficient triboiontronics via the synergistic effect of temporally controlling EDL formation and redox reaction. **a**, The synergistic principle of temporally controlling EDL formation and redox reaction in the SDC-TING. **b**, The verification experiment of their synergistic effect. **c**, **d**, and **e**, The I_{sc} density, Q_{sc} density, and P_R density of the SDC-TING. **f**, The comparison of the Q_{sc} density of the conventional solid-liquid TENGs, solid-solid TENGs, DC-TING, PDC-TING, and SDC-TING. **g**, The V_{OC} of the SDC-TINGs was improved by the 10-node connecting in series. **h**, Without the assistance of the rectifier bridge, the SDC-TINGs in series could directly supply energy to electrical devices. **i**, In the information flow field, the sensitivity of the PDC-TING/SDC-TING to external stimuli has facilitated the development of self-powered sensing systems.

Fig. S37 | The V_{OC} of the SDC-TING could reach 0.30 V.

Fig. S38 | After the disappearance of the ion concentration gradient caused by temporally controlling EDL formation, the PDC-TING output depended solely on the redox reaction. **a**, The I_{sc} was 1.01 mA. **b**, The Q_{sc} was 10.10 μC . **c**, The V_{oc} was 0.18 V. **d**, The P_R was 0.13 mW.

Fig. S39 | The SDC-TING could operate continuously for more than two hours.

Finally, we would like to thank the reviewer again for these valuable comments and for the thoughtful and careful review towards improving our manuscript.

Reviewer #3 (Remarks to the Author):

This manuscript reports a direct-current triboiontronic nanogenerator based on the physically adsorbed ions. The data format in the manuscript is very monotonous, and there is a lack of data or characterization to support the concept of an asymmetric double electric layer. The concept proposed by the authors is not innovative, the basic principle is not novel, the output performance is not impressive, and there are many confusing and unexplained descriptions. Considering the high standards of this journal, I believe that this manuscript is not suitable for publication in this journal. Here are some questions for reference:

Response: Thanks for the precious comments. In response to the reviewer's valuable suggestions and comments, we have made point-to-point modifications to improve the manuscript.

1. Lines 114-116 states that the denser Stern layer on the PET surface hinders subsequent liquid-solid interface charge transfer. The authors should provide more evidence to support this claim. One way to do this would be to characterize the potential of the PET solid surface. Additionally, does a denser Stern layer on the PET surface mean that the solid surface charge is saturated?

Response: Thank you for your very constructive suggestions. When a DI water droplet slid through the pristine PET film surface, it generated a positive I_{SC} of 3.50 nA and Q_{SC} of 0.38 nC (Fig. 2c and d, and Fig. S16). However, the charge carried by subsequent droplets gradually diminished and became negligible after about 300 seconds. This decline might be attributed to the densification of the EDL on the moistened PET film surface over time, induced by the continuous impact of droplets, which screened further charge transfer between subsequent droplets and the moistened PET film^[19, 25, 26]. To confirm this screening effect, the surface potential of the PET film was measured at various intervals during continuous droplet dripping using a TREK347 high-speed surface potentiometer (Fig S17). Initially calibrated to 0 V, the surface potential of the pristine PET film dropped to about -600 V after approximately 60 seconds of continuous droplet deposition, indicating EDL formation and the accumulation of a net negative triboelectric charge on the PET film. Subsequent droplet deposition further decreased the surface potential, indicating densification of the EDL and the accumulation of more net negative triboelectric charge. After approximately 300

seconds of continuous droplet dripping, the surface potential stabilized, suggesting the formation of a dense and stable EDL on the PET surface. This EDL likely acted as a barrier, inhibiting charge transfer between the moistened PET film and subsequent droplets, potentially resulting in subsequent droplets carrying no triboelectric charge.

19 Nie, J. et al. Probing contact-electrification-induced electron and ion transfers at a liquid-solid interface. *Adv. Mater.* **32**, e1905696 (2020).

22 Zhang, J. et al. Triboelectric nanogenerator as a probe for measuring the charge transfer between liquid and solid surfaces. *ACS Nano.* **12**, 14830-14837 (2021).

23 Chi, J. et al. Harvesting water-evaporation-induced electricity based on liquid-solid triboelectric nanogenerator. *Adv. Sci.* **9**, e2201586 (2022).

Corresponding revisions please see the revised manuscript (Page 12, lines 207-212):

“In addition, the electrical signals generated by triboelectric charges gradually decreased as the droplets continued to slide through the PET film and became negligible after about 300 seconds (Fig. S16). The changing trend of the surface potential (Fig. S17) might prove that the EDL on the moistened PET film gradually became denser due to the continuous impact of the droplets. This EDL likely acted as a barrier, inhibiting charge transfer between the moistened PET film and subsequent droplets, potentially resulting in subsequent droplets carrying no triboelectric charge^[19, 25, 26].”

Corresponding revisions please see the revised Supplementary Information

(Page 20, lines 120-129):

“Fig. S17 | To confirm the screening effect of the EDL, the surface potential of the PET film was measured at various intervals during continuous droplet dripping using a TREK347 high-speed surface potentiometer. Initially calibrated to 0 V, the surface potential of the pristine PET film dropped to about -600 V after approximately 60 seconds of continuous droplet deposition, indicating EDL formation and the accumulation of a net negative triboelectric charge on the PET film. Subsequent droplet deposition further decreased the surface potential, indicating densification of the EDL and the accumulation of more net negative triboelectric charge. After approximately 300 seconds of continuous droplet dripping, the surface potential stabilized, suggesting the formation of a dense and stable EDL on the PET surface. This EDL likely acted as a barrier, inhibiting charge transfer between the moistened PET film and subsequent droplets, potentially resulting in subsequent droplets carrying no triboelectric charge.”

Fig. 2 | The effect of the Au layer sputtering time of the Au/PET layer on the substrate-DI water CE was investigated. **a**, The testing system of triboelectric charges carried by the droplet sliding through the tested film surface. **b**, The “two-step” EDL model for the dielectric surface. **c** and **d**, the I_{sc} and Q_{sc} generated by water droplets sliding through the different film surfaces. **e**, The surface potential of different films after droplet sliding. **f-h**, EDLs on different surfaces of the dielectric, metal, and dielectric substrate sputtered with the metal layer, respectively.

Fig. S16 | When a DI water droplet slid through the pristine PET film surface, it generated a positive I_{SC} of 3.50 nA and Q_{SC} of 0.38 nC. However, the charge carried by subsequent droplets gradually diminished and became negligible after about 300 seconds. This decline might be attributed to the densification of the EDL on the moistened PET film surface over time, induced by the continuous impact of droplets, which screened further charge transfer between subsequent droplets and the moistened PET film. **a.** The generated I_{SC} . **b.** The generated Q_{SC} .

Fig. S17 | To confirm the screening effect of the EDL, the surface potential of the PET film was measured at various intervals during continuous droplet dripping using a TREK347 high-speed surface potentiometer. Initially calibrated to 0 V, the surface potential of the pristine PET film dropped to about -600 V after approximately 60 seconds of continuous droplet deposition, indicating EDL formation and the accumulation of a net negative triboelectric charge on the PET film. Subsequent droplet deposition further decreased the surface potential, indicating densification of the EDL and the accumulation of more net negative triboelectric charge. After approximately 300 seconds of continuous droplet dripping, the surface potential stabilized, suggesting the formation of a dense and stable EDL on the PET surface. This EDL likely acted as a barrier, inhibiting charge transfer between the moistened PET film and subsequent droplets, potentially resulting in subsequent droplets carrying no triboelectric charge.

2. Lines 122-123 states that the low ion concentration in the diffusion layer limits the output. However, in liquid-solid contact electrification, high ion concentrations in the liquid usually suppress the output. The authors are requested to explain this contradiction.

Response: Thank you very much for your feedback. We rewrite the description in lines 122-123 in the old manuscript to provide a clearer and more detailed analysis and description.

Upon contact with a solid surface, a liquid with higher ion concentration could lead to the rapid formation of a dense EDL due to the direct adhesion of ions to the surface. This effectively screens charge transfer between the solid and liquid^[19, 25, 26]. For example, when a DI water droplet slid across the pristine PET film surface, it produced a positive I_{SC} of 3.50 nA and Q_{SC} of 0.38 nC (Fig. 2c and d, and Fig. S16). Subsequent droplets carried a gradually diminishing charge, becoming negligible after approximately 300 seconds. In contrast, when a 1M LiCl solution replaced the DI water, droplets sliding across the pristine PET film carried minimal charge (0.025 nC) and rapidly decayed to zero within 25 seconds (Fig. S18).

For the description in lines 122-123 in the old manuscript, we would like to express that hydrovoltaic technology by moving the EDL boundary and conventional solid-liquid TENGs are limited by their reliance on a small amount of triboelectric charge in the diffuse layer, restricting their output. Hydrovoltaic

technology^[7, 28, 30] typically manipulated the EDL boundary on the charge-collecting layer of the dielectric substrate, leading to reduced ion concentration gradients and longer migration distances, thus limiting output. Similarly, conventional solid-liquid TENGs^[8, 31-33] rely on the movement of the diffuse layer on the dielectric surface to induce the electronic displacement current. However, their higher internal resistance^[50] (in the ten mega-Ohm range) results in higher V_{OC} but limits both I_{SC} and Q_{SC} . In contrast, PDC-TING operated on efficient ion migration driven by the ion concentration gradient, resulting in lower internal resistance (in the hundred-Ohm range) and ultra-high ion flux. Compared with the Q_{SC} densities of hydrovoltaic technology by moving the EDL boundary^[7, 28, 30] and conventional solid-liquid TENGs^[8, 31-33], that of PDC-TING or SDC-TING was increased by several orders of magnitude (Fig. 4f).

Corresponding revisions please see the revised manuscript (Page 12, lines 212-216):

“When a 1M LiCl solution replaced the DI water, droplets sliding through the pristine PET film carried minimal charge (0.025 nC) and rapidly decayed to zero within 25 seconds (Fig. S18). This might be due to a large number of ions being directly adhered to the solid surface to form a dense EDL, effectively screening charge transfer between the solid and liquid.”

Corresponding revisions please see the revised manuscript (Page 23, lines

375-383):

“Compared with the Q_{SC} densities of hydrovoltaic technology by moving the EDL boundary^[7, 28, 30], conventional solid-liquid TENGs^[8, 31-33] and solid-solid TENGs^[34-49], or DC-TING^[21], that of PDC-TING or SDC-TING was increased by several orders of magnitude (Fig. 4f). Hydrovoltaic technology^[7, 28, 30] typically manipulated the EDL boundary on the charge-collecting layer of the dielectric substrate, often resulting in limited ion concentration gradients and increased migration distances, which restricted overall output. Conventional solid-liquid TENGs^[8, 31-33] and solid-solid TENGs^[34-49] utilized triboelectric charges on dielectric materials to induce electronic displacement currents. The insulation properties of dielectrics usually led to higher internal resistance^[50] (in the ten mega-Ohm range), enabling higher V_{OC} but limiting both I_{SC} and Q_{SC} .”

Corresponding revisions please see the revised manuscript (Page 23, lines 386-388):

“In contrast, PDC-TING and SDC-TING relied on efficient ion migration driven by concentration gradients, resulting in lower internal resistance (in the hundred-Ohm range) and ultra-high ion flux.”

Fig. 2 | The effect of the Au layer sputtering time of the Au/PET layer on the substrate-DI water CE was investigated. **a**, The testing system of triboelectric charges carried by the droplet sliding through the tested film surface. **b**, The “two-step” EDL model for the dielectric surface. **c** and **d**, the I_{sc} and Q_{sc} generated by water droplets sliding through the different film surfaces. **e**, The surface potential of different films after droplet sliding. **f-h**, EDLs on different surfaces of the dielectric, metal, and dielectric substrate sputtered with the metal layer, respectively.

Fig. S16 | When a DI water droplet slid through the pristine PET film surface, it generated a positive I_{SC} of 3.50 nA and Q_{SC} of 0.38 nC. However, the charge carried by subsequent droplets gradually diminished and became negligible after about 300 seconds. This decline might be attributed to the densification of the EDL on the moistened PET film surface over time, induced by the continuous impact of droplets, which screened further charge transfer between subsequent droplets and the moistened PET film. a, The generated I_{SC} . b, The generated Q_{SC} .

Fig. S18 | As droplets of 1 M LiCl solution continued to slide through the PET film surface, the generated electrical signals were tested. **a**, The generated I_{sc} . **b**, The generated Q_{sc} .

Fig. 4 | More efficient triboiontronics via the synergistic effect of temporally controlling EDL formation and redox reaction. **a**, The synergistic principle of temporally controlling EDL formation and redox reaction in the SDC-TING. **b**, The verification experiment of their synergistic effect. **c**, **d**, and **e**, The I_{sc} density, Q_{sc} density, and P_R density of the SDC-TING. **f**, The comparison of the Q_{sc} density of the conventional solid-liquid TENGs, solid-solid TENGs, DC-TING, PDC-TING, and SDC-TING. **g**, The V_{OC} of the SDC-TINGs was improved by the 10-node connecting in series. **h**, Without the assistance of the rectifier bridge, the SDC-TINGs in series could directly supply energy to electrical devices. **i**, In the information flow field, the sensitivity of the PDC-TING/SDC-TING to external stimuli has facilitated the development of self-powered sensing systems.

3. Figure 1 shows that the authors' data indicate that water carries almost no charge in the later stages of the test. The reason for this needs to be further explained.

Response: Thank you for your very valuable feedback. When a DI water droplet slid through the pristine PET film surface, it generated a positive I_{SC} of 3.50 nA and Q_{SC} of 0.38 nC (Fig. 2c and d, and Fig. S16). However, the charge carried by subsequent droplets gradually diminished and became negligible after about 300 seconds. This decline might be attributed to the densification of the EDL on the moistened PET film surface over time, induced by the continuous impact of droplets, which screened further charge transfer between subsequent droplets and the moistened PET film^[19, 25, 26]. To confirm this screening effect, the surface potential of the PET film was measured at various intervals during continuous droplet dripping using a TREK347 high-speed surface potentiometer (Fig S17). Initially calibrated to 0 V, the surface potential of the pristine PET film dropped to about -600 V after approximately 60 seconds of continuous droplet deposition, indicating EDL formation and the accumulation of a net negative triboelectric charge on the PET film. Subsequent droplet deposition further decreased the surface potential, indicating densification of the EDL and the accumulation of more net negative triboelectric charge. After approximately 300 seconds of continuous droplet dripping, the surface potential stabilized, suggesting the formation of a dense and stable EDL on the PET surface. This EDL likely acted

as a barrier, inhibiting charge transfer between the moistened PET film and subsequent droplets, potentially resulting in subsequent droplets carrying no triboelectric charge.

19 Nie, J. et al. Probing contact-electrification-induced electron and ion transfers at a liquid-solid interface. *Adv. Mater.* **32**, e1905696 (2020).

22 Zhang, J. et al. Triboelectric nanogenerator as a probe for measuring the charge transfer between liquid and solid surfaces. *ACS Nano*. **12**, 14830-14837 (2021).

23 Chi, J. et al. Harvesting water-evaporation-induced electricity based on liquid-solid triboelectric nanogenerator. *Adv. Sci.* **9**, e2201586 (2022).

Corresponding revisions please see the revised manuscript (Page 12, lines 207-212):

“In addition, the electrical signals generated by triboelectric charges gradually decreased as the droplets continued to slide through the PET film and became negligible after about 300 seconds (Fig. S16). The changing trend of the surface potential (Fig. S17) might prove that the EDL on the moistened PET film gradually became denser due to the continuous impact of the droplets. This EDL likely acted as a barrier, inhibiting charge transfer between the moistened PET film and subsequent droplets, potentially resulting in subsequent droplets carrying no triboelectric charge^[19, 25, 26].”

Corresponding revisions please see the revised Supplementary Information (Page 20, lines 120-129):

“**Fig. S17 | To confirm the screening effect of the EDL, the surface**

potential of the PET film was measured at various intervals during continuous droplet dripping using a TREK347 high-speed surface potentiometer. Initially calibrated to 0 V, the surface potential of the pristine PET film dropped to about -600 V after approximately 60 seconds of continuous droplet deposition, indicating EDL formation and the accumulation of a net negative triboelectric charge on the PET film. Subsequent droplet deposition further decreased the surface potential, indicating densification of the EDL and the accumulation of more net negative triboelectric charge. After approximately 300 seconds of continuous droplet dripping, the surface potential stabilized, suggesting the formation of a dense and stable EDL on the PET surface. This EDL likely acted as a barrier, inhibiting charge transfer between the moistened PET film and subsequent droplets, potentially resulting in subsequent droplets carrying no triboelectric charge.”

Fig. 2 | The effect of the Au layer sputtering time of the Au/PET layer on the substrate-DI water CE was investigated. **a**, The testing system of triboelectric charges carried by the droplet sliding through the tested film surface. **b**, The “two-step” EDL model for the dielectric surface. **c** and **d**, the I_{sc} and Q_{sc} generated by water droplets sliding through the different film surfaces. **e**, The surface potential of different films after droplet sliding. **f-h**, EDLs on different surfaces of the dielectric, metal, and dielectric substrate sputtered with the metal layer, respectively.

Fig. S16 | When a DI water droplet slid through the pristine PET film surface, it generated a positive I_{SC} of 3.50 nA and Q_{SC} of 0.38 nC. However, the charge carried by subsequent droplets gradually diminished and became negligible after about 300 seconds. This decline might be attributed to the densification of the EDL on the moistened PET film surface over time, induced by the continuous impact of droplets, which screened further charge transfer between subsequent droplets and the moistened PET film. a, The generated I_{SC} . b, The generated Q_{SC} .

Fig. S17 | To confirm the screening effect of the EDL, the surface potential of the PET film was measured at various intervals during continuous droplet dripping using a TREK347 high-speed surface potentiometer. Initially calibrated to 0 V, the surface potential of the pristine PET film dropped to about -600 V after approximately 60 seconds of continuous droplet deposition, indicating EDL formation and the accumulation of a net negative triboelectric charge on the PET film. Subsequent droplet deposition further decreased the surface potential, indicating densification of the EDL and the accumulation of more net negative triboelectric charge. After approximately 300 seconds of continuous droplet dripping, the surface potential stabilized, suggesting the formation of a dense and stable EDL on the PET surface. This EDL likely acted as a barrier, inhibiting charge transfer between the moistened PET film and subsequent droplets, potentially resulting in subsequent droplets carrying no triboelectric charge.

4. The thickness of gold or copper should be characterized for different deposition times, and their surface morphology should also be characterized. Is the metal film uniform and completely covers the PET when the deposition time is short?

Response: Thank you for your very valuable suggestions. Increasing the sputtering time of Au on the PET substrate reduced the internal resistance of the PDC-TING from 10.6 k Ω to 7.4 k Ω (Fig. 1g). To systematically investigate Au layer coverage, we observed surface morphology using SEM for Au/PET films with varied sputtering times (Fig. S11). SEM observations revealed microscopic cracks on the surfaces of PET film and Au/PET film sputtered for 1 minute (Fig. S11a and b), indicating insufficient Au deposition within 1 minute, leading to higher internal resistance (10.6 k Ω). With 10 minutes of sputtering, the cracks notably diminished (Fig. S11c), suggesting improved Au coverage and enhanced electrical conductivity, resulting in reduced internal resistance (7.6 k Ω). Further extending sputtering to more than 20 minutes nearly eliminated the cracks, maintaining internal resistance at around 7.4 k Ω (Fig. S11d and e). Therefore, adjusting the sputtering time of the metal enabled control over its coverage on the dielectric substrate, thereby affecting the charge-collecting layer's conductivity.

Corresponding revisions please see the revised manuscript (Page 10, lines 174-184):

“In addition, increasing the sputtering time of Au on the PET substrate

reduced the internal resistance of the PDC-TING from 10.6 k Ω to 7.4 k Ω (Fig. 1g). Scanning electron microscope (SEM) observations revealed microscopic cracks on the surfaces of PET film and Au/PET film sputtered for 1 minute (Fig. S11a and b), indicating insufficient Au deposition within 1 minute, leading to higher internal resistance (10.6 k Ω). With 10 minutes of sputtering, the cracks notably diminished (Fig. S11c), suggesting improved Au coverage and enhanced electrical conductivity, resulting in reduced internal resistance (7.6 k Ω). Further extending sputtering to more than 20 minutes nearly eliminated the cracks, maintaining internal resistance at around 7.4 k Ω (Fig. S11d and e). With the increase in sputtering time, the decrease in transmittance of Au/PET films also proved the increase of Au coverage on the PET substrate (Fig. S12). Therefore, adjusting the sputtering time of the metal enabled control over its coverage on the dielectric substrate, thereby affecting the charge-collecting layer's conductivity.”

Fig. 1 | The principle of the triboiontronics and the construction of the PDC-TING. **a**, Triboiontronics via temporally controlling EDL formation. (i) Firstly, when DI water contacted the bottom Au/PET layer, it interacted with the PET substrate through microscopic cracks in the sputtered Au layer, forming a stable EDL. (ii) Secondly, as the top Au/PET layer moved downward to contact with water, the initial CE led to form a new EDL, establishing two EDLs with significantly different symmetries. This created an ion concentration gradient, generating an ionic current I_{il} . (iii) Thirdly, the detachment of the top Au/PET layer halted ion migration. (iv) Through repeated contact and separation cycles, the ion migration persisted until equilibrium was achieved between the two EDLs. **b**, The model display of the PDC-TING. **c**, Comparison of output performance of different generators. **d**, The effect of different liquid types on the PDC-TING output. **e**, The effect of the extent and direction of the ion concentration gradient on the PDC-TING output. **f**, The effect of the Au-layer sputtering time of Au/PET layers on the PDC-TING output. **g**, The effect of the Au-layer sputtering time of Au/PET layers on the internal resistance of the PDC-TING.

Fig. S11 | The surface morphology of the different films was observed using a scanning electron microscope (SEM). **a**, PET film. **b** Au/PET layer with the 1-minute sputtering time. **c**, Au/PET layer with the 10-minute sputtering time. **d**, Au/PET layer with the 20-minute sputtering time. **e**, Au/PET layer with the 30-minute sputtering time.

5. The authors did not provide a description of Fig. 1 g-i. The comparison of these three EDL is very important, and the authors are requested to supplement it.

Response: Thanks for the reviewer's precious comments. To systematically investigate Au layer coverage, we observed surface morphology using SEM for Au/PET films with varied sputtering times (Fig. S11). Furthermore, we established a system to further investigate the impact of Au layer coverage on the CE characteristics of Au/PET films in greater detail (Fig. 2a and Fig. S13). The surface potential of the pristine tested films was calibrated to 0 V. When the water droplet slid through the pristine PET, it generated a positive I_{SC} of 3.50 nA and Q_{SC} of 0.38 nC (Fig. 2c and d), and the surface potential was about -59 V (Fig. 2e). Conversely, sliding on pristine pure Au resulted in weaker reverse signals, measuring I_{SC} of 0.16 nA and Q_{SC} of 0.022 nC, with the surface potential around 3 V. Increasing sputtering time from 1 minute to 30 minutes for different pristine Au/PET films decreased positive electrical signals from I_{SC} of 1.42 nA and Q_{SC} of 0.15 nC to I_{SC} of 0.26 nA and Q_{SC} of 0.030 nC. Correspondingly, the surface potential gradually changed from -30 V to -6 V. Similarly, for Cu/PET films (Fig. S14), signals decreased from I_{SC} of 1.37 nA and Q_{SC} of 0.14 nC to I_{SC} of 0.23 nA and Q_{SC} of 0.029 mC (Fig. S15). These signals, reflecting the EDL charge density, revealed that adjusting metal sputtering time controls coverage on the dielectric substrate. It affected charge-collecting layer conductivity and CE characteristics of the dielectric substrate, enabling the construction of EDLs with varied densities

at DI water-PET substrates.

Based on the above experiments, EDL models were developed for various solid-liquid contact interfaces. Firstly, the dense EDL for the dielectric surface followed the "two-step" model (Fig. 2f). Secondly, weaker reverse electrical signals from falling droplets indicated reversed charge distribution and lower density in the EDL for the pure metal surface (Fig. 2g). Due to high hydration-free energy, cations couldn't directly adsorb onto the solid surface^[27], forming an IHP with the water molecule dipole layer while hydrated cations constituted the OHP. Thirdly, when DI water contacted the sputtered metal charge-collecting layer of the dielectric substrate, the EDL could form through microscopic cracks (Fig. 2h). Its charge distribution resembled that of the dielectric surface EDL, with intermediate charge density compared to the dielectric and pure metal surfaces.

24 Bockris, J. O. M., Devanathan, M. A. V. & Muller, K. On the structure of charged interfaces. *Electrochemistry* 832-863 (1965).

Corresponding revisions please see the revised manuscript (Pages 11-12, lines 188-207):

“To explore the effect of Au layer coverage on the CE characteristics of Au/PET films in more detail, a system was constructed (Fig. 2a). A 6 cm long tested film was mounted on an acrylic substrate at a 45° inclination angle, and a grounding syringe consistently released a single drop of approximately 25 μL DI

water onto the film. Triboelectric charges carried by the droplet were measured when it slid through the tested film into the Faraday cylinder, and then the surface potential of the tested film was measured (Fig. S13). According to the “two-step” EDL model (Fig. 2b), the Stern layer adhered tightly to the tested film, so these triboelectric charges belong to the diffuse layer retained within the droplet. Correspondingly, the evaluation of the triboelectric charge and the surface potential might indicate the EDL density on the tested film. The surface potential of the pristine tested films was calibrated to 0 V. When the water droplet slid through the pristine PET, it generated a positive I_{SC} of 3.50 nA and Q_{SC} of 0.38 nC (Fig. 2c and d), and the surface potential was about -59 V (Fig. 2e). Conversely, sliding on pristine pure Au resulted in weaker reverse signals, measuring I_{SC} of 0.16 nA and Q_{SC} of 0.022 nC, with the surface potential around 3 V. Increasing sputtering time from 1 minute to 30 minutes for different pristine Au/PET films decreased positive electrical signals from I_{SC} of 1.42 nA and Q_{SC} of 0.15 nC to I_{SC} of 0.26 nA and Q_{SC} of 0.030 nC. Correspondingly, the surface potential gradually changed from -30 V to -6 V. Similarly, for Cu/PET films (Fig. S14), signals decreased from I_{SC} of 1.37 nA and Q_{SC} of 0.14 nC to I_{SC} of 0.23 nA and Q_{SC} of 0.029 nC (Fig. S15). These signals, reflecting the EDL charge density, revealed that adjusting metal sputtering time controls coverage on the dielectric substrate. It affected charge-collecting layer conductivity and CE characteristics of the dielectric substrate, enabling the construction of EDLs with varied densities at DI water-PET substrates.”

Corresponding revisions please see the revised manuscript (Page 14, lines 225-233):

“Based on the above experiments, EDL models were developed for various solid-liquid contact interfaces. Firstly, the dense EDL for the dielectric surface followed the "two-step" model (Fig. 2f). Secondly, weaker reverse electrical signals from falling droplets indicated reversed charge distribution and lower density in the EDL for the pure metal surface (Fig. 2g). Due to high hydration-free energy, cations couldn't directly adsorb onto the solid surface^[27], forming an IHP with the water molecule dipole layer while hydrated cations constituted the OHP. Thirdly, when DI water contacted the sputtered metal charge-collecting layer of the dielectric substrate, the EDL could form through microscopic cracks (Fig. 2h). Its charge distribution resembled that of the dielectric surface EDL, with intermediate charge density compared to the dielectric and pure metal surfaces.”

Fig. S11 | The surface morphology of the different films was observed using a scanning electron microscope (SEM). **a**, PET film. **b** Au/PET layer with the 1-minute sputtering time. **c**, Au/PET layer with the 10-minute sputtering time. **d**, Au/PET layer with the 20-minute sputtering time. **e**, Au/PET layer with the 30-minute sputtering time.

Fig. 2 | The effect of the Au layer sputtering time of the Au/PET layer on the substrate-DI water CE was investigated. **a**, The testing system of triboelectric charges carried by the droplet sliding through the tested film surface. **b**, The “two-step” EDL model for the dielectric surface. **c** and **d**, the I_{sc} and Q_{sc} generated by water droplets sliding through the different film surfaces. **e**, The surface potential of different films after droplet sliding. **f-h**, EDLs on different surfaces of the dielectric, metal, and dielectric substrate sputtered with the metal layer, respectively.

Fig. S13 | The experimental system for measuring surface potential by a high-speed surface potentiometer.

Fig. S14 | Comparison of light transmittance of the PET film, pure Cu, and Cu/PET layers with different sputtering times of the Cu layer.

Fig. S15 | The electric signals generated as DI Water droplets slid through the PET film, pure Cu, and Cu/PET films with different sputtering times of the Cu layer, respectively. **a**, The comparison of the I_{sc} . **b**, The comparison of the Q_{sc} .

6. The organization of the author's Figure 1 is problematic. Overall, the manuscript is more concerned with increasing the output performance. However, from the perspective of Figure 1, the introduction of the charge collection layer reduces the charge carried by the liquid. This is easy to cause misunderstanding for readers because they cannot obtain the advantages of this new strategy. I strongly suggest that the author should put the relevant data of Figure 2 into Figure 1 to further highlight the ability of this strategy to increase output performance. In addition, the significance of the data in Figure 1 is confusing, and the contact between the liquid and PET does not guide the subsequent data.

Response: Thank you very much for taking the time to review our work and for providing such valuable feedback. Based on your comments, we have made the adjustments accordingly to the graphics, ensuring that they are now presented in a clearer and more comprehensible manner. Specifically, we have reorganized Fig. 1 and Fig. 2 and provided more detailed and accurate descriptions of their contents. We believe that these improvements will greatly benefit our readers by helping them better grasp the main results and perspectives of our study. We sincerely appreciate your professional guidance and constructive criticism, and we are confident that these enhancements will significantly improve the quality and readability of our paper.

The testing of the electrical signal generated by the continuous sliding of droplets from the PET surface served two purposes. Firstly, it aimed to confirm

that the EDL on the moistened PET film surface became denser over time due to the continuous impact of droplets, thus screening further charge transfer between subsequent droplets and the moistened PET film. Secondly, it sought to verify that the triboelectric charge carried in the diffuse layer during the solid-liquid CE process is minimal. Hydrovoltaic technology by moving the EDL boundary^[7, 28, 30] and conventional solid-liquid TENGs^[8, 31-33] rely on the movement of a small amount of triboelectric charge in the diffuse layer, which limits their output. This experiment was crucial for guiding the investigation of the charge transfer mechanism between solid-liquid CE and the charge distribution in the formed EDL. However, as it was not directly related to PDC-TING, it was transferred to Fig. S16.

Corresponding revisions please see the revised manuscript (Page 12, lines 207-216):

“In addition, the electrical signals generated by triboelectric charges gradually decreased as the droplets continued to slide through the PET film and became negligible after about 300 seconds (Fig. S16). The changing trend of the surface potential (Fig. S17) might prove that the EDL on the moistened PET film gradually became denser due to the continuous impact of the droplets. This EDL likely acted as a barrier, inhibiting charge transfer between the moistened PET film and subsequent droplets, potentially resulting in subsequent droplets carrying no triboelectric charge^[19, 25, 26]. When a 1M LiCl solution replaced the

DI water, droplets sliding through the pristine PET film carried minimal charge (0.025 nC) and rapidly decayed to zero within 25 seconds (Fig. S18). This might be due to a large number of ions being directly adhered to the solid surface to form a dense EDL, effectively screening charge transfer between the solid and liquid.”

Corresponding revisions please see the revised manuscript (Page 23, lines 375-383):

“Compared with the Q_{sc} densities of hydrovoltaic technology by moving the EDL boundary^[7, 28, 30], conventional solid-liquid TENGs^[8, 31-33] and solid-solid TENGs^[34-49], or DC-TING^[21], that of PDC-TING or SDC-TING was increased by several orders of magnitude (Fig. 4f). Hydrovoltaic technology^[7, 28, 30] typically manipulated the EDL boundary on the charge-collecting layer of the dielectric substrate, often resulting in limited ion concentration gradients and increased migration distances, which restricted overall output. Conventional solid-liquid TENG^[8, 31-33] and solid-solid TENG^[34-49] utilized triboelectric charges on dielectric materials to induce electronic displacement currents. The insulation properties of dielectrics usually led to higher internal resistance^[50] (in the ten mega-Ohm range), enabling higher V_{oc} but limiting both I_{sc} and Q_{sc} .”

Fig. 1 | The principle of the triboiontronics and the construction of the PDC-TING. **a**, Triboiontronics via temporally controlling EDL formation. (i) Firstly, when DI water contacted the bottom Au/PET layer, it interacted with the PET substrate through microscopic cracks in the sputtered Au layer, forming a stable EDL. (ii) Secondly, as the top Au/PET layer moved downward to contact with water, the initial CE led to form a new EDL, establishing two EDLs with significantly different symmetries. This created an ion concentration gradient, generating an ionic current I_{il} . (iii) Thirdly, the detachment of the top Au/PET layer halted ion migration. (iv) Through repeated contact and separation cycles, the ion migration persisted until equilibrium was achieved between the two EDLs. **b**, The model display of the PDC-TING. **c**, Comparison of output performance of different generators. **d**, The effect of different liquid types on the PDC-TING output. **e**, The effect of the extent and direction of the ion concentration gradient on the PDC-TING output. **f**, The effect of the Au-layer sputtering time of Au/PET layers on the PDC-TING output. **g**, The effect of the Au-layer sputtering time of Au/PET layers on the internal resistance of the PDC-TING.

Fig. 2 | The effect of the Au layer sputtering time of the Au/PET layer on the substrate-DI water CE was investigated. **a**, The testing system of triboelectric charges carried by the droplet sliding through the tested film surface. **b**, The “two-step” EDL model for the dielectric surface. **c** and **d**, the I_{sc} and Q_{sc} generated by water droplets sliding through the different film surfaces. **e**, The surface potential of different films after droplet sliding. **f-h**, EDLs on different surfaces of the dielectric, metal, and dielectric substrate sputtered with the metal layer, respectively.

Fig. S16 | When a DI water droplet slid through the pristine PET film surface, it generated a positive I_{SC} of 3.50 nA and Q_{SC} of 0.38 nC. However, the charge carried by subsequent droplets gradually diminished and became negligible after about 300 seconds. This decline might be attributed to the densification of the EDL on the moistened PET film surface over time, induced by the continuous impact of droplets, which screened further charge transfer between subsequent droplets and the moistened PET film. a, The generated I_{SC} . b, The generated Q_{SC} .

7. It is not sufficient to only discuss the change in liquid charge in Figure 1. Data on the change in solid surface charge should also be included.

Response: Thanks for the precious comments. To explore the effect of Au layer coverage on the CE characteristics of Au/PET films in more detail, we measured triboelectric charges carried by sliding droplets and subsequently tested film surface potential (Fig. 2a and Fig. S13). The surface potential of the pristine tested films was calibrated to 0 V. When the water droplet slid through the pristine PET, it generated a positive I_{SC} of 3.50 nA and Q_{SC} of 0.38 nC (Fig. 2c and d), and the surface potential was about -59 V (Fig. 2e). Conversely, sliding on pristine pure Au resulted in weaker reverse signals, measuring I_{SC} of 0.16 nA and Q_{SC} of 0.022 nC, with the surface potential around 3 V. Increasing sputtering time from 1 minute to 30 minutes for different pristine Au/PET films decreased positive electrical signals from I_{SC} of 1.42 nA and Q_{SC} of 0.15 nC to I_{SC} of 0.26 nA and Q_{SC} of 0.030 nC. Correspondingly, the surface potential gradually changed from -30 V to -6 V. These signals, reflecting the EDL charge density, revealed that adjusting metal sputtering time controls coverage on the dielectric substrate. It affected charge-collecting layer conductivity and CE characteristics of the dielectric substrate, enabling the construction of EDLs with varied densities at DI water-PET substrates.

Corresponding revisions please see the revised manuscript (Pages 11-12,

lines 188-207):

“To explore the effect of Au layer coverage on the CE characteristics of Au/PET films in more detail, a system was constructed (Fig. 2a). A 6 cm long tested film was mounted on an acrylic substrate at a 45° inclination angle, and a grounding syringe consistently released a single drop of approximately 25 μ L DI water onto the film. Triboelectric charges carried by the droplet were measured when it slid through the tested film into the Faraday cylinder, and then the surface potential of the tested film was measured (Fig. S13). According to the “two-step” EDL model (Fig. 2b), the Stern layer adhered tightly to the tested film, so these triboelectric charges belong to the diffuse layer retained within the droplet. Correspondingly, the evaluation of the triboelectric charge and the surface potential might indicate the EDL density on the tested film. The surface potential of the pristine tested films was calibrated to 0 V. When the water droplet slid through the pristine PET, it generated a positive I_{SC} of 3.50 nA and Q_{SC} of 0.38 nC (Fig. 2c and d), and the surface potential was about -59 V (Fig. 2e). Conversely, sliding on pristine pure Au resulted in weaker reverse signals, measuring I_{SC} of 0.16 nA and Q_{SC} of 0.022 nC, with the surface potential around 3 V. Increasing sputtering time from 1 minute to 30 minutes for different pristine Au/PET films decreased positive electrical signals from I_{SC} of 1.42 nA and Q_{SC} of 0.15 nC to I_{SC} of 0.26 nA and Q_{SC} of 0.030 nC. Correspondingly, the surface potential gradually changed from -30 V to -6 V. Similarly, for Cu/PET films (Fig. S14), signals decreased from I_{SC} of 1.37 nA and Q_{SC} of 0.14 nC to I_{SC} of 0.23 nA and Q_{SC} of

0.029 mC (Fig. S15). These signals, reflecting the EDL charge density, revealed that adjusting metal sputtering time controls coverage on the dielectric substrate. It affected charge-collecting layer conductivity and CE characteristics of the dielectric substrate, enabling the construction of EDLs with varied densities at DI water-PET substrates.”

Fig. 2 | The effect of the Au layer sputtering time of the Au/PET layer on the substrate-DI water CE was investigated. **a**, The testing system of triboelectric charges carried by the droplet sliding through the tested film surface. **b**, The “two-step” EDL model for the dielectric surface. **c** and **d**, the I_{sc} and Q_{sc} generated by water droplets sliding through the different film surfaces. **e**, The surface potential of different films after droplet sliding. **f-h**, EDLs on different surfaces of the dielectric, metal, and dielectric substrate sputtered with the metal layer, respectively.

Fig. S13 | The experimental system for measuring surface potential by a high-speed surface potentiometer.

8. In Line 161, the authors believe that the cations in the lower EDL will participate in the construction of the upper EDL. This seems impossible, as the size of the EDL is only a few to tens of nanometers. The signal may be generated by the dynamic change of the upper contact area with water, which causes the charge imbalance between the upper and lower devices.

Response: Thanks for the reviewer's precious comments. In response to the reviewer's suggestion, we have further refined the operational mechanism of the PDC-TING in this paper. By temporally controlling EDL formation, we construct asymmetric EDLs. This process induces an ion concentration gradient, driving ions in the liquid to migrate directionally, thereby generating an effective ionic current. Further elaboration on this mechanism is provided below:

The EDL functioned as an exceptional interface for ionic-electronic coupling between the solid dielectric and liquid within the "two-step" model. By dynamically regulating the EDL, iontronics enabled precise control over ion migration and electron coupling transfer at the interface. Here, through temporally controlling EDL formation at solid-liquid interfaces, efficient triboiontronics was established (Fig. 1a). Firstly, a metal layer (Au) as a charge-collecting layer was sputtered onto a dielectric substrate (polyethylene terephthalate, PET). Upon contact of DI water with the bottom Au/PET layer, it might engage in direct interaction with the PET substrate via microscopic cracks in the sputtered Au layer, leading to the solid-liquid CE and the establishment of

a stable EDL [Fig. 1a(i)]. Secondly, as the top Au/PET layer moved downwards to contact with DI water, the initial CE led to the formation of a new EDL, thus establishing two EDLs with significantly different symmetries [Fig. 1a(ii)]. This created an ion concentration gradient, facilitating electron transfer in the external circuit and generating an ionic current I_{i1} . Thirdly, the detachment of the top Au/PET layer halted ion migration [Fig. 1a(iii)]. Through repeated contact and separation cycles, the constructed ion concentration gradient gradually weakened, yet the ion migration persisted until equilibrium was achieved between the two EDLs [Fig. 1a(iv)].

Corresponding revisions please see the revised manuscript (Page 5, lines 92-105):

“The EDL functioned as an exceptional interface for ionic-electronic coupling between the solid dielectric and liquid within the "two-step" model. By dynamically regulating the EDL, iontronics enabled precise control over ion migration and electron coupling transfer at the interface. Here, through temporally controlling EDL formation at solid-liquid interfaces, efficient triboiontronics was established (Fig. 1a). Firstly, a metal layer (Au) as a charge-collecting layer was sputtered onto a dielectric substrate (polyethylene terephthalate, PET). Upon contact of deionized (DI) water with the bottom Au/PET layer, it might engage in direct interaction with the PET substrate via microscopic cracks in the sputtered Au layer, leading to the solid-liquid CE and

the establishment of a stable EDL [Fig. 1a(i)]. Secondly, as the top Au/PET layer moved downwards to contact with DI water, the initial CE led to the formation of a new EDL, thus establishing two EDLs with significantly different symmetries [Fig. 1a(ii)]. This created an ion concentration gradient, facilitating electron transfer in the external circuit and generating an ionic current I_{i1} . Thirdly, the detachment of the top Au/PET layer halted ion migration [Fig. 1a(iii)]. Through repeated contact and separation cycles, the constructed ion concentration gradient gradually weakened, yet the ion migration persisted until equilibrium was achieved between the two EDLs [Fig. 1a(iv)].”

Fig. 1 | The principle of the triboiontronics and the construction of the PDC-TING. **a**, Triboiontronics via temporally controlling EDL formation. (i) Firstly, when DI water contacted the bottom Au/PET layer, it interacted with the PET substrate through microscopic cracks in the sputtered Au layer, forming a stable EDL. (ii) Secondly, as the top Au/PET layer moved downward to contact with water, the initial CE led to form a new EDL, establishing two EDLs with significantly different symmetries. This created an ion concentration gradient, generating an ionic current I_{il} . (iii) Thirdly, the detachment of the top Au/PET layer halted ion migration. (iv) Through repeated contact and separation cycles, the ion migration persisted until equilibrium was achieved between the two EDLs. **b**, The model display of the PDC-TING. **c**, Comparison of output performance of different generators. **d**, The effect of different liquid types on the PDC-TING output. **e**, The effect of the extent and direction of the ion concentration gradient on the PDC-TING output. **f**, The effect of the Au-layer sputtering time of Au/PET layers on the PDC-TING output. **g**, The effect of the Au-layer sputtering time of Au/PET layers on the internal resistance of the PDC-TING.

9. The “pre-moistening of PET film substrates” statement is quite confusing. Why wet the PET film instead of the metal? How is this process achieved? A relevant schematic diagram and description are necessary.

Response: Thanks for the reviewer’s precious comments. The effect of the extent and direction of asymmetry in the EDL formation on the PDC-TING output was investigated by pre-moistening different Au/PET layers (Fig. 1e and Fig. S5). The specific pre-moistening methods were shown in Fig. S6. When applying 200 μL water in the PDC-TING with both pristine Au/PET layers sputtered with 1-minute Au, the constructed ion concentration gradient is weaker, resulting in lower positive outputs with I_{SC} of 0.97 μA , Q_{SC} of 59.00 nC, and V_{OC} of 0.073 V. Pre-moistening was exclusively applied to the bottom Au/PET layer with water to pre-form a dense EDL. Subsequently, it could effectively increase the extent of the ion concentration gradient in the PDC-TING, improving the positive output to I_{SC} of 8.20 μA , Q_{SC} of 523.15 nC, and V_{OC} of 0.23 V. On the contrary, the pre-moistening of the top Au/PET layer could pre-built a dense EDL, causing a higher reverse ion concentration gradient in the PDC-TING with negative electrical signals (I_{SC} of -2.75 μA , Q_{SC} of -100.69 nC, and V_{OC} of -0.11 V). When both Au/PET layers were pre-moistened, dense EDLs formed on their respective surfaces, resulting in a diminished ion concentration gradient within the PDC-TING and weaker positive electrical signals (I_{SC} of 0.45 μA , Q_{SC} of 26.78 nC, and V_{OC} of 0.034 V). Experimental results confirmed that the extent and direction of

asymmetry in the EDL formation played a crucial role in determining the efficacy and direction of ion migration, thereby influencing the PDC-TING output. Pre-moistening the bottom Au/PET layer could facilitate establishing a higher ion concentration gradient, significantly enhancing the ion flux within the PDC-TING.

Corresponding revisions please see the revised manuscript (Page 9, lines 143-159):

“The effect of the extent and direction of asymmetry in the EDL formation on the PDC-TING output was investigated by pre-moistening different Au/PET layers (Fig. 1e and Fig. S5). The specific pre-moistening methods were shown in Fig. S6. When applying 200 μL water in the PDC-TING with both pristine Au/PET layers sputtered with 1-minute Au, the constructed ion concentration gradient is weaker, resulting in lower positive outputs with I_{SC} of 0.97 μA , Q_{SC} of 59.00 nC, and V_{OC} of 0.073 V. Pre-moistening was exclusively applied to the bottom Au/PET layer with water to pre-form a dense EDL. Subsequently, it could effectively increase the extent of the ion concentration gradient in the PDC-TING, improving the positive output to I_{SC} of 8.20 μA , Q_{SC} of 523.15 nC, and V_{OC} of 0.23 V. On the contrary, the pre-moistening of the top Au/PET layer could pre-build a dense EDL, causing a higher reverse ion concentration gradient in the PDC-TING with negative electrical signals (I_{SC} of -2.75 μA , Q_{SC} of -100.69 nC, and V_{OC} of -0.11 V). When both Au/PET layers were pre-moistened, dense EDLs

formed on their respective surfaces, resulting in a diminished ion concentration gradient within the PDC-TING and weaker positive electrical signals (I_{SC} of 0.45 μA , Q_{SC} of 26.78 nC, and V_{OC} of 0.034 V). Experimental results confirmed that the extent and direction of asymmetry in the EDL formation played a crucial role in determining the efficacy and direction of ion migration, thereby influencing the PDC-TING output. Pre-moistening the bottom Au/PET layer could facilitate establishing a higher ion concentration gradient, significantly enhancing the ion flux within the PDC-TING.”

Fig. 1 | The principle of the triboiontronics and the construction of the PDC-TING. **a**, Triboiontronics via temporally controlling EDL formation. (i) Firstly, when DI water contacted the bottom Au/PET layer, it interacted with the PET substrate through microscopic cracks in the sputtered Au layer, forming a stable EDL. (ii) Secondly, as the top Au/PET layer moved downward to contact with water, the initial CE led to form a new EDL, establishing two EDLs with significantly different symmetries. This created an ion concentration gradient, generating an ionic current I_{il} . (iii) Thirdly, the detachment of the top Au/PET layer halted ion migration. (iv) Through repeated contact and separation cycles, the ion migration persisted until equilibrium was achieved between the two EDLs. **b**, The model display of the PDC-TING. **c**, Comparison of output performance of different generators. **d**, The effect of different liquid types on the PDC-TING output. **e**, The effect of the extent and direction of the ion concentration gradient on the PDC-TING output. **f**, The effect of the Au-layer sputtering time of Au/PET layers on the PDC-TING output. **g**, The effect of the Au-layer sputtering time of Au/PET layers on the internal resistance of the PDC-TING.

Fig. S5 | The effect of the extent and direction of asymmetry in the EDL formation on the PDC-TING output was investigated by pre-moistening different Au/PET layers. **a**, The comparison of the Q_{sc} . **b**, The comparison of the V_{oc} .

Fig. S6 | The specific pre-moistening methods of pre-moistening different Au/PET layers. a, The top and bottom Au/PET layers were pristine. **b,** Pre-moistening was exclusively applied to the bottom Au/PET layer with DI water to pre-form a dense EDL. **c,** Pre-moistening was exclusively applied to the top Au/PET layer with DI water to pre-form a dense EDL. **d,** The top and bottom Au/PET layers were pre-moistened.

10. Figure 3a and Fig S11 show a significant increase in transferred charge, while the current remains constant. This seems to contradict our expectations, as transferred charge and current should be positively correlated.

Response: Thanks for the reviewer's precious comments. In the triboiontronics proposed in this work, temporally controlling EDL formation could construct an ion concentration gradient, thereby generating effective electrical signals. Specifically, when DI water contacted the bottom Au/PET layer, it interacted with the PET substrate through microscopic cracks in the sputtered Au layer, forming a stable EDL [Fig. 1a(i)]. Then, as the top Au/PET layer moved downwards to contact with DI water, the initial CE led to the formation of a new EDL, thus establishing two EDLs with significantly different symmetries [Fig. 1a(ii)]. This created an ion concentration gradient, facilitating electron transfer in the external circuit and generating an ionic current I_{i1} . Hence, as long as the ion concentration gradient formed when the upper Au/PET layer contacted DI water remained consistent, the potential difference driving the directional migration of ions remained unchanged. Furthermore, the stable internal resistance of the entire system ensured a stable amplitude of the I_{SC} . During the initial contact of the top Au/PET and DI water, the ion concentration gradient was related to the extent of asymmetry of the EDL formation in the two interfaces, and it might be independent of the operating frequency. Therefore, solely reducing the operating frequency of the PDC-TING from 4 Hz to 1 Hz might hardly influence the initial

amplitude of the V_{OC} of 0.19 V (Fig. S19b) and I_{SC} of 12.18 μA (Fig. 3a). However, decreasing the operating frequency prolonged the contact time between the top Au/PET layer and DI water, thereby increasing the migration time of ions and augmenting the generated Q_{SC} from 0.35 μC to 0.60 μC (Fig. S19a).

Corresponding revisions please see the revised manuscript (Page 15, lines 252-258):

“After determining the operation principle of the PDC-TING, the influence of operating frequency on its output was studied (Fig. 3a and Fig. S19). As it decreased from 4 Hz to 1 Hz, the I_{SC} of 12.18 μA and V_{OC} of 0.19 V of the PDC-TING remained at a stable amplitude. This might be due to during the initial contact of the top Au/PET and water, the ion concentration gradient was related to the asymmetry extent of the EDL formation, independent of the operating frequency. However, its decrease prolonged the contact time between the top Au/PET layer and water, thereby increasing the migration time of ions and augmenting the generated Q_{SC} from 0.35 μC to 0.60 μC .”

Fig. 1 | The principle of the triboiontronics and the construction of the PDC-TING. **a**, Triboiontronics via temporally controlling EDL formation. (i) Firstly, when DI water contacted the bottom Au/PET layer, it interacted with the PET substrate through microscopic cracks in the sputtered Au layer, forming a stable EDL. (ii) Secondly, as the top Au/PET layer moved downward to contact with water, the initial CE led to form a new EDL, establishing two EDLs with significantly different symmetries. This created an ion concentration gradient, generating an ionic current I_{il} . (iii) Thirdly, the detachment of the top Au/PET layer halted ion migration. (iv) Through repeated contact and separation cycles, the ion migration persisted until equilibrium was achieved between the two EDLs. **b**, The model display of the PDC-TING. **c**, Comparison of output performance of different generators. **d**, The effect of different liquid types on the PDC-TING output. **e**, The effect of the extent and direction of the ion concentration gradient on the PDC-TING output. **f**, The effect of the Au-layer sputtering time of Au/PET layers on the PDC-TING output. **g**, The effect of the Au-layer sputtering time of Au/PET layers on the internal resistance of the PDC-TING.

Fig. 3 | The specific affecting factors on the PDC-TING output were explored. a, The I_{sc} of the PDC-TING at different operating frequencies. **b and c,** The positive and negative electrostatic fields were used to adjust the constructed ion concentration gradient. **d,** The influence of hydrophilicity of bottom Au/PET layer on the constructed ion concentration gradient. **e,** The effect of ion concentrations in the liquid on the constructed ion concentration gradient. **f, g, and h,** The I_{sc} density, Q_{sc} density, and P_R density of the PDC-TING could reach 26.00 A/m^2 , 412.54 mC/m^2 , and 8.45 W/m^2 , respectively. **i,** The PDC-TING output could be restored by the electrochemical recovery. **j,** Through temporally controlling EDL formation between the liquid and the pure metal, an ion concentration gradient could also be constructed to drive the directional migration of ions in the metal-based PDC-TING. **k,** The electrical signal direction of metal-based PDC-TING was opposite to the PDC-TING based on the metal/dielectric substrate. **l,** Due to the sparse EDL on the pure metal surface, the optimal negative output of the metal-based PDC-TING was weaker than the optimal positive output of the PDC-TING based on the metal/dielectric substrate.

Fig. S19 | The effect of the operating frequency on the PDC-TING output was investigated. **a**, The comparison of the Q_{sc} . **b**, The comparison of the V_{oc} .

11. Many data plots show the same pattern, with the second peak being smaller than the first. Please explain the reason for this.

Response: Thanks for the precious comments. During the PDC-TING operation, as the top Au/PET layer moved downwards to contact with DI water, the initial CE led to the formation of a new EDL, thus establishing two EDLs with significantly different symmetries [Fig. 1a(ii)]. This created an ion concentration gradient, facilitating electron transfer in the external circuit and generating an ionic current I_{i1} . Thirdly, the detachment of the top Au/PET layer halted ion migration [Fig. 1a(iii)]. At this time, the top Au/PET layer surface formed a relatively sparse EDL. As a result, when the top Au/PET moved downwards to contact DI water again, the ion concentration gradient constructed again was weaker than that in the first contact. So, the I_{SC} amplitude during the second contact was lower than that in the first contact. Through repeated contact and separation cycles, the constructed ion concentration gradient gradually weakened, yet the migration of ions persisted until equilibrium was achieved between the two EDLs [Fig. 1a(iv)].

Corresponding revisions please see the revised manuscript (Page 5, lines 99-105):

“Secondly, as the top Au/PET layer moved downwards to contact with DI water, the initial CE led to the formation of a new EDL, thus establishing two

EDLs with significantly different symmetries [Fig. 1a(ii)]. This created an ion concentration gradient, facilitating electron transfer in the external circuit and generating an ionic current I_{i1} . Thirdly, the detachment of the top Au/PET layer halted ion migration [Fig. 1a(iii)]. Through repeated contact and separation cycles, the constructed ion concentration gradient gradually weakened, yet the ion migration persisted until equilibrium was achieved between the two EDLs [Fig. 1a(iv)].”

Fig. 1 | The principle of the triboiontronics and the construction of the PDC-TING. **a**, Triboiontronics via temporally controlling EDL formation. (i) Firstly, when DI water contacted the bottom Au/PET layer, it interacted with the PET substrate through microscopic cracks in the sputtered Au layer, forming a stable EDL. (ii) Secondly, as the top Au/PET layer moved downward to contact with water, the initial CE led to form a new EDL, establishing two EDLs with significantly different symmetries. This created an ion concentration gradient, generating an ionic current I_{il} . (iii) Thirdly, the detachment of the top Au/PET layer halted ion migration. (iv) Through repeated contact and separation cycles, the ion migration persisted until equilibrium was achieved between the two EDLs. **b**, The model display of the PDC-TING. **c**, Comparison of output performance of different generators. **d**, The effect of different liquid types on the PDC-TING output. **e**, The effect of the extent and direction of the ion concentration gradient on the PDC-TING output. **f**, The effect of the Au-layer sputtering time of Au/PET layers on the PDC-TING output. **g**, The effect of the Au-layer sputtering time of Au/PET layers on the internal resistance of the PDC-TING.

12. The change in the contact angle of the material after different plasma treatments needs to be supplemented.

Response: Thanks for the precious comments. Plasma treatment of the bottom Au/PET layer was conducted to examine the impact of surface hydrophilicity on the ion concentration gradient (Fig. 3d and Fig. S25). Initially, untreated Au/PET layers generated I_{SC} of 12.18 μA and Q_{SC} of 0.60 μC , with a contact angle of approximately 100° (Fig. S26), indicating hydrophobic properties. Increasing the power of Plasma treatment from 25 W to 75 W reduced the contact angle from about 49° to 19° by introducing oxygen-containing functional groups, enhancing hydrophilicity. It effectively increased the bottom EDL density, boosting the ion concentration gradient and elevating output from I_{SC} of 54.79 μA and 2.48 μC to I_{SC} of 150.08 μA and Q_{SC} of 5.97 μC . Beyond 75 W, further power increase did not significantly alter hydrophilicity, maintaining output stability.

Corresponding revisions please see the revised manuscript (Page 16, lines 271-279):

“Secondly, Plasma treatment of the bottom Au/PET layer was conducted to examine the impact of surface hydrophilicity on the ion concentration gradient (Fig. 3d and Fig. S25). Initially, untreated Au/PET layers generated I_{SC} of 12.18 μA and Q_{SC} of 0.60 μC , with a contact angle of approximately 100° (Fig. S26), indicating hydrophobic properties. Increasing the power of Plasma treatment

from 25 W to 75 W reduced the contact angle from about 49° to 19° by introducing oxygen-containing functional groups, enhancing hydrophilicity. It effectively increased the bottom EDL density, boosting the ion concentration gradient and elevating output from I_{SC} of 54.79 μA and 2.48 μC to I_{SC} of 150.08 μA and Q_{SC} of 5.97 μC . Beyond 75 W, further power increase did not significantly alter hydrophilicity, maintaining output stability.”

Fig. 3 | The specific affecting factors on the PDC-TING output were explored. a, The I_{sc} of the PDC-TING at different operating frequencies. **b and c**, The positive and negative electrostatic fields were used to adjust the constructed ion concentration gradient. **d**, The influence of hydrophilicity of bottom Au/PET layer on the constructed ion concentration gradient. **e**, The effect of ion concentrations in the liquid on the constructed ion concentration gradient. **f, g, and h**, The I_{sc} density, Q_{sc} density, and P_R density of the PDC-TING could reach 26.00 A/m², 412.54 mC/m², and 8.45 W/m², respectively. **i**, The PDC-TING output could be restored by the electrochemical recovery. **j**, Through temporally controlling EDL formation between the liquid and the pure metal, an ion concentration gradient could also be constructed to drive the directional migration of ions in the metal-based PDC-TING. **k**, The electrical signal direction of metal-based PDC-TING was opposite to the PDC-TING based on the metal/dielectric substrate. **l**, Due to the sparse EDL on the pure metal surface, the optimal negative output of the metal-based PDC-TING was weaker than the optimal positive output of the PDC-TING based on the metal/dielectric substrate.

Fig. S25 | Plasma treatment of the bottom Au/PET layer was conducted to examine the impact of surface hydrophilicity on the ion concentration gradient. **a**, The comparison of the Q_{sc} . **b**, The comparison of the V_{oc} .

Fig. S26 | Comparison of contact angles of Au/PET layers with different Plasma treatment power. **a**, The contact angle of the initial Au/PET layer was about 100° . **b**, The contact angle of the Au/PET layer with 25-W Plasma treatment was about 49° . **c**, The contact angle of the Au/PET layer with 50-W Plasma treatment was about 29° . **d**, The contact angle of the Au/PET layer with 75-W Plasma treatment was about 19° . **e**, The contact angle of the Au/PET layer with 100-W Plasma treatment was about 18° .

13. The strategy of enhancing the output by redox reaction is not desirable, as it will continuously consume copper and greatly limit the durability of the device.

Response: Thanks for the reviewer's precious comments. The synergistic effect of temporally controlling EDL formation and redox reaction could effectively promote the coupling migration of multiple ions, thereby enhancing the ion flux, as illustrated in Fig. 4b. In particular, under the synergistic effect, the pure energy harvesting device PDC-TING was converted to an integrated energy harvesting and storage device SDC-TING. Utilizing the substantial Cu^{2+} ion flux generated by the redox reaction, the SDC-TING could enhance its output performance and operational stability simultaneously. With an internal resistance of $500\ \Omega$, it achieved the I_{SC} of $5.58\ \text{mA}$ (Fig. 4c), Q_{SC} of $54.07\ \mu\text{C}$ (Fig. 4d), P_{R} of $3.86\ \text{mW}$ (Fig. 4e), and V_{OC} of $0.30\ \text{V}$ (Fig. S37). The I_{SC} density, Q_{SC} density, and P_{R} density could reach up to $55.77\ \text{A/m}^2$, $540.70\ \text{mC/m}^2$, and $38.64\ \text{W/m}^2$. The efficient SDC-TING output not only increased by 452.48% , 435.35% , and 2869.23% respectively compared to that completely depending on the redox reaction (Fig. S38), but also increased by 114.62% , 31.08% , and 354.12% respectively compared to the PDC-TING output. Comparative analysis with the PDC-TING revealed significantly improved stability under the synergistic effect (Fig. S39).

To address the issue of copper dissipation, selecting copper with a high specific surface area could enhance the operational stability of the SDC-TING.

Furthermore, copper, being economically feasible compared to precious metals like Au, could be readily replaced once the Cu/PET layer was consumed, ensuring the efficient and stable operation of the SDC-TING.

Corresponding revisions please see the revised manuscript (Page 21, lines 348-358):

“In particular, under the synergistic effect, the pure energy harvesting device PDC-TING was converted to an integrated energy harvesting and storage device SDC-TING. Utilizing the substantial Cu^{2+} ion flux generated by the redox reaction, the SDC-TING could enhance its output performance and operational stability simultaneously. With an internal resistance of 500Ω , it achieved the I_{SC} of 5.58 mA (Fig. 4c), Q_{SC} of $54.07 \mu\text{C}$ (Fig. 4d), P_{R} of 3.86 mW (Fig. 4e), and V_{OC} of 0.30 V (Fig. S37). The I_{SC} density, Q_{SC} density, and P_{R} density could reach up to 55.77 A/m^2 , 540.70 mC/m^2 , and 38.64 W/m^2 . The efficient SDC-TING output not only increased by 452.48% , 435.35% , and 2869.23% respectively compared to that completely depending on the redox reaction (Fig. S38), but also increased by 114.62% , 31.08% , and 354.12% respectively compared to the PDC-TING output. Comparative analysis with the PDC-TING revealed significantly improved stability under the synergistic effect (Fig. S39).”

Corresponding revisions please see the revised manuscript (Page 21, lines 360-364):

“To address the issue of copper dissipation, selecting copper with a high specific surface area could enhance the operational stability of the SDC-TING. Furthermore, copper, being economically feasible compared to precious metals like Au, could be readily replaced once the Cu/PET layer was consumed, ensuring the efficient and stable operation of the SDC-TING.”

Fig. 4 | More efficient triboiontronics via the synergistic effect of temporally controlling EDL formation and redox reaction. **a**, The synergistic principle of temporally controlling EDL formation and redox reaction in the SDC-TING. **b**, The verification experiment of their synergistic effect. **c**, **d**, and **e**, The I_{sc} density, Q_{sc} density, and P_R density of the SDC-TING. **f**, The comparison of the Q_{sc} density of the conventional solid-liquid TENGs, solid-solid TENGs, DC-TING, PDC-TING, and SDC-TING. **g**, The V_{OC} of the SDC-TINGs was improved by the 10-node connecting in series. **h**, Without the assistance of the rectifier bridge, the SDC-TINGs in series could directly supply energy to electrical devices. **i**, In the information flow field, the sensitivity of the PDC-TING/SDC-TING to external stimuli has facilitated the development of self-powered sensing systems.

Fig. S37 | The V_{oc} of the SDC-TING could reach 0.30 V.

Fig. S38 | After the disappearance of the ion concentration gradient caused by temporally controlling EDL formation, the PDC-TING output depended solely on the redox reaction. **a**, The I_{sc} was 1.01 mA. **b**, The Q_{sc} was 10.10 μC . **c**, The V_{oc} was 0.18 V. **d**, The P_R was 0.13 mW.

Fig. S39 | The SDC-TING could operate continuously for more than two hours.

14. Different pH liquids should be measured, and the durability of the device output should also be measured.

Response: Thanks for the precious comments. To assess the impact of varying acidity and alkalinity of different solutions on PDC-TING output, experiments were conducted using three solutions: HCl, LiCl, and NaOH, across different pH levels. As shown in Fig. S28, as the ion concentration increased from 1 μM to 1 M, the output performance remained consistent for neutral (LiCl) and alkaline (NaOH) solutions, with I_{SC} increasing from about 40 μA to approximately 1000 μA . However, under acidic conditions (HCl), the PDC-TING exhibited weaker I_{SC} compared to the other two solutions at the same ion concentration. Particularly at a higher concentration of 1 M, a lower reverse electrical signal was generated (about 450 μA). This behavior might be attributed to the higher adsorption of H^+ ions from the HCl solution onto the dielectric surface, leading to a reversal of the charge distribution pattern within the EDL^[19, 28, 30].

In addition, as illustrated in Fig. 4b, the synergistic effect transformed the standalone energy harvesting device PDC-TING into an integrated energy harvesting and storage device, SDC-TING. Utilizing the substantial Cu^{2+} ion flux generated by the redox reaction, the SDC-TING could enhance its output performance and operational stability simultaneously. Comparative analysis with the PDC-TING revealed significantly improved stability under the synergistic effect (Fig. S39).

- 19 Nie, J. et al. Probing contact-electrification-induced electron and ion transfers at a liquid-solid interface. *Adv. Mater.* **32**, e1905696 (2020).
- 25 Yin, J. et al. Generating electricity by moving a droplet of ionic liquid along graphene. *Nat. Nanotechnol.* **9**, 378-383 (2014).
- 29 Yin, J. et al. Waving potential in graphene. *Nat. Commun.* **5**, 3582 (2014).

Corresponding revisions please see the revised manuscript (Pages 16-17, lines 287-293):

“In addition, as the ion concentration increased from 1 μM to 1 M (Fig. S28), the output performance remained consistent for neutral (LiCl) and alkaline (NaOH) solutions, with I_{SC} increasing from about 40 μA to approximately 1000 μA . However, under acidic conditions (HCl), the PDC-TING exhibited weaker I_{SC} compared to the other two solutions. Particularly at a higher concentration of 1 M, a lower reverse electrical signal was generated (about 450 μA). This behavior might be attributed to the higher adsorption of H^+ ions from the HCl solution onto the dielectric surface, leading to a reversal of the charge distribution pattern within the EDL^[19, 28, 30].”

Corresponding revisions please see the revised manuscript (Page 21, lines 350-352):

“Utilizing the substantial Cu^{2+} ion flux generated by the redox reaction, the SDC-TING could enhance its output performance and operational stability simultaneously.”

Corresponding revisions please see the revised manuscript (Page 21, lines

357-358):

“Comparative analysis with the PDC-TING revealed significantly improved stability under the synergistic effect (Fig. S39).”

Fig. S28 | The effect of varying acidity and alkalinity of different solutions on the PDC-TING output was investigated.

Fig. 4 | More efficient triboiontronics via the synergistic effect of temporally controlling EDL formation and redox reaction. **a**, The synergistic principle of temporally controlling EDL formation and redox reaction in the SDC-TING. **b**, The verification experiment of their synergistic effect. **c**, **d**, and **e**, The I_{sc} density, Q_{sc} density, and P_R density of the SDC-TING. **f**, The comparison of the Q_{sc} density of the conventional solid-liquid TENGs, solid-solid TENGs, DC-TING, PDC-TING, and SDC-TING. **g**, The V_{oc} of the SDC-TINGs was improved by the 10-node connecting in series. **h**, Without the assistance of the rectifier bridge, the SDC-TINGs in series could directly supply energy to electrical devices. **i**, In the information flow field, the sensitivity of the PDC-TING/SDC-TING to external stimuli has facilitated the development of self-powered sensing systems.

Fig. S39 | The SDC-TING could operate continuously for more than two hours.

15. The authors believe that the origin of the device output in this manuscript is from liquid-solid contact electrification. However, contact electrification is dominated by displacement current, and in this study, the liquid is directly contacted with the metal conductor, in which case the output may be dominated by conduction current.

Response: Thanks for the precious comments. In the triboiontronics proposed in this work, temporally controlling EDL formation could construct an ion concentration gradient, thereby generating effective ionic electrical signals (Fig. 1a). Specifically, when DI water contacted the bottom Au/PET layer, it interacted with the PET substrate through microscopic cracks in the sputtered Au layer, forming a stable EDL [Fig. 1a(i)]. Then, as the top Au/PET layer moved downwards to contact with DI water, the initial CE led to the formation of a new EDL, thus establishing two EDLs with significantly different symmetries [Fig. 1a(ii)]. This created an ion concentration gradient, facilitating electron transfer in the external circuit and generating an ionic current I_{i1} . The PDC-TING relied on efficient ion migration driven by concentration gradients, resulting in lower internal resistance (in the hundred-Ohm range) and ultra-high ion flux. In contrast, as an electronic device, conventional solid-liquid TENG^[8, 31-33] utilized triboelectric charges on dielectric materials to induce electronic displacement currents. The insulation properties of dielectrics led to higher internal resistance^[50] (in the ten mega-Ohm range), enabling higher V_{OC} but limiting both I_{SC} and Q_{SC} .

Compared with the Q_{SC} of conventional solid-liquid TENGs^[8, 31-33], that of PDC-TING was increased by several orders of magnitude (Fig. 4f).

Corresponding revisions please see the revised manuscript (Page 5, lines 96-103):

“Firstly, a metal layer (Au) as a charge-collecting layer was sputtered onto a dielectric substrate (polyethylene terephthalate, PET). Upon contact of deionized (DI) water with the bottom Au/PET layer, it might engage in direct interaction with the PET substrate via microscopic cracks in the sputtered Au layer, leading to the solid-liquid CE and the establishment of a stable EDL [Fig. 1a(i)]. Secondly, as the top Au/PET layer moved downwards to contact with DI water, the initial CE led to the formation of a new EDL, thus establishing two EDLs with significantly different symmetries [Fig. 1a(ii)]. This created an ion concentration gradient, facilitating electron transfer in the external circuit and generating an ionic current I_{i1} .”

Corresponding revisions please see the revised manuscript (Page 23, lines 375-377):

“Compared with the Q_{SC} densities of hydrovoltaic technology by moving the EDL boundary^[7, 28, 30], conventional solid-liquid TENGs^[8, 31-33] and solid-solid TENGs^[34-49], or DC-TING^[21], that of PDC-TING or SDC-TING was increased by several orders of magnitude (Fig. 4f).”

Corresponding revisions please see the revised manuscript (Page 23, lines 380-383):

“Conventional solid-liquid TENG^[8, 31-33] and solid-solid TENG^[34-49] utilized triboelectric charges on dielectric materials to induce electronic displacement currents. The insulation properties of dielectrics usually led to higher internal resistance^[50] (in the ten mega-Ohm range), enabling higher V_{OC} but limiting both I_{SC} and Q_{SC} .”

Corresponding revisions please see the revised manuscript (Page 23, lines 386-388):

“In contrast, PDC-TING and SDC-TING relied on efficient ion migration driven by concentration gradients, resulting in lower internal resistance (in the hundred-Ohm range) and ultra-high ion flux.”

Fig. 1 | The principle of the triboiontronics and the construction of the PDC-TING. **a**, Triboiontronics via temporally controlling EDL formation. (i) Firstly, when DI water contacted the bottom Au/PET layer, it interacted with the PET substrate through microscopic cracks in the sputtered Au layer, forming a stable EDL. (ii) Secondly, as the top Au/PET layer moved downward to contact with water, the initial CE led to form a new EDL, establishing two EDLs with significantly different symmetries. This created an ion concentration gradient, generating an ionic current I_{i1} . (iii) Thirdly, the detachment of the top Au/PET layer halted ion migration. (iv) Through repeated contact and separation cycles, the ion migration persisted until equilibrium was achieved between the two EDLs. **b**, The model display of the PDC-TING. **c**, Comparison of output performance of different generators. **d**, The effect of different liquid types on the PDC-TING output. **e**, The effect of the extent and direction of the ion concentration gradient on the PDC-TING output. **f**, The effect of the Au-layer sputtering time of Au/PET layers on the PDC-TING output. **g**, The effect of the Au-layer sputtering time of Au/PET layers on the internal resistance of the PDC-TING.

Fig. 4 | More efficient triboiontronics via the synergistic effect of temporally controlling EDL formation and redox reaction. **a**, The synergistic principle of temporally controlling EDL formation and redox reaction in the SDC-TING. **b**, The verification experiment of their synergistic effect. **c**, **d**, and **e**, The I_{sc} density, Q_{sc} density, and P_R density of the SDC-TING. **f**, The comparison of the Q_{sc} density of the conventional solid-liquid TENGs, solid-solid TENGs, DC-TING, PDC-TING, and SDC-TING. **g**, The V_{OC} of the SDC-TINGs was improved by the 10-node connecting in series. **h**, Without the assistance of the rectifier bridge, the SDC-TINGs in series could directly supply energy to electrical devices. **i**, In the information flow field, the sensitivity of the PDC-TING/SDC-TING to external stimuli has facilitated the development of self-powered sensing systems.

16. The presence of PET is puzzling. What would happen if PET is removed? Does PET generate electrostatic induction? How will the thickness of PET affect the output? This raises a new question, the relationship between the existence of Figure 1 and Figures 2-4 is confusing.

Response: Thanks for the precious comments. EDL models were developed for various solid-liquid contact interfaces. Firstly, the dense EDL for the dielectric surface followed the "two-step" model (Fig. 2f). Secondly, weaker reverse electrical signals from falling droplets indicated reversed charge distribution and lower density in the EDL for the pure metal surface (Fig. 2g). Due to high hydration-free energy, cations couldn't directly adsorb onto the solid surface^[27], forming an IHP with the water molecule dipole layer while hydrated cations constituted the OHP. Thirdly, when DI water contacted the sputtered metal charge-collecting layer of the dielectric substrate, the EDL could form through microscopic cracks (Fig. 2h). Its charge distribution resembled that of the dielectric surface EDL, with intermediate charge density compared to the dielectric and pure metal surfaces.

While pure dielectric could form the dense EDL, they cannot directly transfer charges by insulation. Conversely, although pure Au exhibited excellent conductivity, the minimal triboelectric charge in the EDL limited the ion concentration gradient, leading to a weak electrical signal. Only the dielectric substrate that sputtered the metal layer could ensure that PDC-TING produces

effective ion flux, based on the distinct advantages of the formation of cracks in the metal sputtering layer. These cracks serve as conduits for liquid infiltration, thereby leveraging the solid-liquid CE properties of the dielectric layer. This enhances charge storage and transfer efficiency. Additionally, the cracks facilitate efficient current collection, reducing resistance and improving charge transport within the material. This combined effect enhances the performance of the hybrid material, offering opportunities for applications requiring high power density and rapid charge/discharge rates. It offers versatility and innovative solutions across various domains.

Therefore, the optimal negative output of the metal-based PDC-TING with I_{SC} of $-163.85 \mu\text{A}$, Q_{SC} of $-7.45 \mu\text{C}$, and P_R of $3.03 \mu\text{W}$ (Fig. 3l and Fig. S34) was weaker than the optimal positive output of the PDC-TENG with Au/PET (I_{SC} of 2.60 mA , Q_{SC} of $41.25 \mu\text{C}$ and P_R of 0.85 mW in Fig. 3f-h). This might be due to the charge distribution and density in the EDL for the pure metal surface (Fig. 2g) being opposite and weaker than those for the metal/dielectric surface (Fig. 2h).

The effect of the PET substrate thickness in the Au/PET layer on the PDC-TING output was investigated. When keeping the sputtering time of the Au layer for 10 min, increasing the thickness of the PET substrate from $125 \mu\text{m}$ to $500 \mu\text{m}$ had minimal effect on the PDC-TING output maintained at about I_{SC} of $8.20 \mu\text{A}$ and Q_{SC} of 523.15 nC (Fig. S8). This phenomenon might be attributed to the nature of the EDL as an interfacial phenomenon occurring at the solid-liquid contact interface. The charge density within the EDL remained unaffected by the

thickness of the solid material when the contact area remained consistent. This was different from conventional TENG. When the dielectric material (PET film) increased from 60 μm to 225 μm , the electrical signal generated by TENG decreased from about I_{SC} of 0.80 μA and Q_{SC} of 6.0 nC to I_{SC} of 0.47 μA and Q_{SC} of 4.3 nC (Fig. S9). Because it used the triboelectric charge on the dielectric surface to directly drive the electron transfer, too thick dielectric material might reduce the output performance. Therefore, the PDC-TING had a wider applicability to materials than conventional TENG.

We have made the adjustments accordingly to the graphics, ensuring that they are now presented in a clearer and more comprehensible manner. Specifically, we have reorganized Fig. 1 and Fig. 2 and provided more detailed and accurate descriptions of their contents. The purpose of the new Fig.2 (old Fig.1) is to explore the EDL model of the contact between the dielectric substrate of the sputtered metal layer and deionized water. The effects of the coverage of the metal layer on the charge density in the EDL and the internal resistance of the device were investigated, so as to explore its effect on the output performance of PDC-TING.

Corresponding revisions please see the revised manuscript (Page 14, lines 225-243):

“Based on the above experiments, EDL models were developed for various solid-liquid contact interfaces. Firstly, the dense EDL for the dielectric surface

followed the "two-step" model (Fig. 2f). Secondly, weaker reverse electrical signals from falling droplets indicated reversed charge distribution and lower density in the EDL for the pure metal surface (Fig. 2g). Due to high hydration-free energy, cations couldn't directly adsorb onto the solid surface^[27], forming an IHP with the water molecule dipole layer while hydrated cations constituted the OHP. Thirdly, when DI water contacted the sputtered metal charge-collecting layer of the dielectric substrate, the EDL could form through microscopic cracks (Fig. 2h). Its charge distribution resembled that of the dielectric surface EDL, with intermediate charge density compared to the dielectric and pure metal surfaces. While pure dielectric could form the dense EDL, they cannot directly transfer charges by insulation. Conversely, although pure Au exhibited excellent conductivity, the minimal triboelectric charge in the EDL limited the ion concentration gradient, leading to a weak electrical signal. Only the dielectric substrate that sputtered the metal layer could ensure that PDC-TING produces effective ion flux, based on the distinct advantages of the formation of cracks in the metal sputtering layer. These cracks serve as conduits for liquid infiltration, thereby leveraging the solid-liquid CE properties of the dielectric layer. This enhances charge storage and transfer efficiency. Additionally, the cracks facilitate efficient current collection, reducing resistance and improving charge transport within the material. This combined effect enhances the performance of the hybrid material, offering opportunities for applications requiring high power density and rapid charge/discharge rates. It offers versatility and innovative solutions across

various domains.”

Corresponding revisions please see the revised manuscript (Page 19, lines 307-308):

“The optimal output of PDC-TING could reach I_{SC} of 2.60 mA (Fig. 3f), Q_{SC} of 41.25 μ C (Fig. 3g), P_R (peak power) of 0.85 mW (Fig. 3h), and V_{OC} of 0.25 V (Fig. S29).”

Corresponding revisions please see the revised manuscript (Page 19, lines 318-322):

“The optimal negative output of the metal-based PDC-TING could reach I_{SC} of -163.85 μ A, Q_{SC} of -7.45 μ C, V_{OC} of -0.21 V, and P_R of 3.03 μ W (Fig. 3l and Fig. S34), which was weaker than the optimal positive output of the PDC-TENG with Au/PET. This might be due to the charge distribution and density in the EDL for the pure metal surface (Fig. 2g) being opposite and weaker than those for the metal/dielectric surface (Fig. 2h).”

Corresponding revisions please see the revised manuscript (Page 10, lines 164-168):

“As an interface phenomenon, as long as the water was in full contact with the bottom Au/PET layer, the EDL density could be guaranteed to ensure the PDC-TING output, independent of the dielectric substrate thickness (Fig. S8). It

was different from the traditional TENG required a thinner dielectric material to ensure output (Fig. S9), thereby the PDC-TING had wider applicability to materials.”

Fig. 2 | The effect of the Au layer sputtering time of the Au/PET layer on the substrate-DI water CE was investigated. **a**, The testing system of triboelectric charges carried by the droplet sliding through the tested film surface. **b**, The “two-step” EDL model for the dielectric surface. **c** and **d**, the I_{sc} and Q_{sc} generated by water droplets sliding through the different film surfaces. **e**, The surface potential of different films after droplet sliding. **f-h**, EDLs on different surfaces of the dielectric, metal, and dielectric substrate sputtered with the metal layer, respectively.

Fig. 3 | The specific affecting factors on the PDC-TING output were explored. a, The I_{sc} of the PDC-TING at different operating frequencies. **b and c,** The positive and negative electrostatic fields were used to adjust the constructed ion concentration gradient. **d,** The influence of hydrophilicity of bottom Au/PET layer on the constructed ion concentration gradient. **e,** The effect of ion concentrations in the liquid on the constructed ion concentration gradient. **f, g,** and **h,** The I_{sc} density, Q_{sc} density, and P_R density of the PDC-TING could reach 26.00 A/m^2 , 412.54 mC/m^2 , and 8.45 W/m^2 , respectively. **i,** The PDC-TING output could be restored by the electrochemical recovery. **j,** Through temporally controlling EDL formation between the liquid and the pure metal, an ion concentration gradient could also be constructed to drive the directional migration of ions in the metal-based PDC-TING. **k,** The electrical signal direction of metal-based PDC-TING was opposite to the PDC-TING based on the metal/dielectric substrate. **l,** Due to the sparse EDL on the pure metal surface, the optimal negative output of the metal-based PDC-TING was weaker than the optimal positive output of the PDC-TING based on the metal/dielectric substrate.

Fig. S34 | Combining the three promotion strategies of the ion concentration gradient, the optimal negative output performance of metal-based PDC-TING was tested. a, The Q_{sc} could reach -7.45 μC . **b,** The V_{oc} could reach -0.21 V. **c,** The peak power (P_R) could reach 3.03 μW .

Fig. S8 | The effect of the thickness of the PET substrate in the Au/PET layer on the PDC-TING output was investigated. a, When keeping the sputtering time of the Au layer for 10 min, increasing the thickness of the PET substrate from 125 μm to 500 μm had minimal effect on the PDC-TING output, and the I_{sc} remained relatively stable with 8.20 μA . **b,** The Q_{sc} remained relatively stable with 523.15 nC.

Fig. S9 | The effect of the thickness of the dielectric material on the conventional solid-solid TENG output was investigated. a, When the dielectric material (PET film) increased from 60 μm to 225 μm , the I_{sc} decreased from about 0.80 μA to 0.47 μA . **b,** The Q_{sc} decreased from about 6.0 nC to 4.3 nC.

17. In this manuscript, are the charge transfer carriers electrons and ions? Can the contributions of these two be distinguished?

Response: Thanks for the reviewer's precious comments. In the triboiontronics proposed in this work, temporally controlling EDL formation could construct an ion concentration gradient, thereby generating effective ionic electrical signals (Fig. 1a). Specifically, when DI water contacted the bottom Au/PET layer, it interacted with the PET substrate through microscopic cracks in the sputtered Au layer, forming a stable EDL [Fig. 1a(i)]. Then, as the top Au/PET layer moved downwards to contact with DI water, the initial CE led to the formation of a new EDL, thus establishing two EDLs with significantly different symmetries [Fig. 1a(ii)]. This created an ion concentration gradient, facilitating electron transfer in the external circuit and generating an ionic current I_{i1} . So the charge carriers in the PDC-TING were mainly ions, not electrons.

To distinguish the contribution of ion migration and electron transfer to the SDC-TING output, we first determined the impact of electron transfer. When 200 μL of DI water was dispensed onto the PTFE film adhered to the lower Au/PET surface (Fig. S1c), the PDC-TING transitioned into a conventional solid-liquid TENG, operating on the principles of solid-liquid CE and electrostatic induction. It generated AC displacement electrical signals (3.5 nA and Q_{SC} of 0.32 nC) (Fig. 1c and Fig. S2a), significantly smaller than ionic current signals generated by PDC-TING (I_{SC} of 2.60 mA and Q_{SC} of 41.25 μC Fig. 3f and g) and by the SDC-

TING (I_{SC} of 5.58 mA and Q_{SC} of 54.07 μC in Fig. 4c and d). This suggested that the contribution of electron transfer to the SDC-TING output should be negligible.

Corresponding revisions please see the revised manuscript (Page 5, lines 96-103)

“Firstly, a metal layer (Au) as a charge-collecting layer was sputtered onto a dielectric substrate (polyethylene terephthalate, PET). Upon contact of deionized (DI) water with the bottom Au/PET layer, it might engage in direct interaction with the PET substrate via microscopic cracks in the sputtered Au layer, leading to the solid-liquid CE and the establishment of a stable EDL [Fig. 1a(i)]. Secondly, as the top Au/PET layer moved downwards to contact with DI water, the initial CE led to the formation of a new EDL, thus establishing two EDLs with significantly different symmetries [Fig. 1a(ii)]. This created an ion concentration gradient, facilitating electron transfer in the external circuit and generating an ionic current I_{i1} .”

Corresponding revisions please see the revised manuscript (Page 6, lines 118-121):

“When DI water was dispensed onto the PTFE film adhered to the bottom Au/PET surface (Fig. S1c), the ion migration was hindered, and the conventional solid-liquid TENG was constructed. Weak AC electronics signals were generated only by electrostatic induction with I_{SC} of 3.53 nA, Q_{SC} of 0.32 nC Q_{SC} , and V_{OC}

of 0.48 V.”

Corresponding revisions please see the revised manuscript (Page 19, lines 307-308):

“The optimal output of PDC-TING could reach I_{SC} of 2.60 mA (Fig. 3f), Q_{SC} of 41.25 μ C (Fig. 3g), P_R (peak power) of 0.85 mW (Fig. 3h), and V_{OC} of 0.25 V (Fig. S29).”

Corresponding revisions please see the revised manuscript (Page 21, lines 352-353):

“With an internal resistance of 500 Ω , it achieved the I_{SC} of 5.58 mA (Fig. 4c), Q_{SC} of 54.07 μ C (Fig. 4d), P_R of 3.86 mW (Fig. 4e), and V_{OC} of 0.30 V (Fig. S37).”

Fig. 1 | The principle of the triboiontronics and the construction of the PDC-TING. **a**, Triboiontronics via temporally controlling EDL formation. (i) Firstly, when DI water contacted the bottom Au/PET layer, it interacted with the PET substrate through microscopic cracks in the sputtered Au layer, forming a stable EDL. (ii) Secondly, as the top Au/PET layer moved downward to contact with water, the initial CE led to form a new EDL, establishing two EDLs with significantly different symmetries. This created an ion concentration gradient, generating an ionic current I_{il} . (iii) Thirdly, the detachment of the top Au/PET layer halted ion migration. (iv) Through repeated contact and separation cycles, the ion migration persisted until equilibrium was achieved between the two EDLs. **b**, The model display of the PDC-TING. **c**, Comparison of output performance of different generators. **d**, The effect of different liquid types on the PDC-TING output. **e**, The effect of the extent and direction of the ion concentration gradient on the PDC-TING output. **f**, The effect of the Au-layer sputtering time of Au/PET layers on the PDC-TING output. **g**, The effect of the Au-layer sputtering time of Au/PET layers on the internal resistance of the PDC-TING.

Fig. S2 | Comparison of output performance of different generators. **a**, The comparison of the transferred charge (Q_{sc}). **b**, The comparison of the open-circuit voltage (V_{oc}). **c**, The detailed display of the V_{oc} during the directional migration of ions in the PDC-TING.

Fig. 3 | The specific affecting factors on the PDC-TING output were explored. a, The I_{sc} of the PDC-TING at different operating frequencies. **b and c,** The positive and negative electrostatic fields were used to adjust the constructed ion concentration gradient. **d,** The influence of hydrophilicity of bottom Au/PET layer on the constructed ion concentration gradient. **e,** The effect of ion concentrations in the liquid on the constructed ion concentration gradient. **f, g,** and **h,** The I_{sc} density, Q_{sc} density, and P_R density of the PDC-TING could reach 26.00 A/m^2 , 412.54 mC/m^2 , and 8.45 W/m^2 , respectively. **i,** The PDC-TING output could be restored by the electrochemical recovery. **j,** Through temporally controlling EDL formation between the liquid and the pure metal, an ion concentration gradient could also be constructed to drive the directional migration of ions in the metal-based PDC-TING. **k,** The electrical signal direction of metal-based PDC-TING was opposite to the PDC-TING based on the metal/dielectric substrate. **l,** Due to the sparse EDL on the pure metal surface, the metal-based PDC-TING output was weaker than the PDC-TING based on the metal/dielectric substrate.

Fig. 4 | More efficient triboiontronics via the synergistic effect of temporally controlling EDL formation and redox reaction. **a**, The synergistic principle of temporally controlling EDL formation and redox reaction in the SDC-TING. **b**, The verification experiment of their synergistic effect. **c**, **d**, and **e**, The I_{sc} density, Q_{sc} density, and P_R density of the SDC-TING. **f**, The comparison of the Q_{sc} density of the conventional solid-liquid TENGs, solid-solid TENGs, DC-TING, PDC-TING, and SDC-TING. **g**, The V_{OC} of the SDC-TINGs was improved by the 10-node connecting in series. **h**, Without the assistance of the rectifier bridge, the SDC-TINGs in series could directly supply energy to electrical devices. **i**, In the information flow field, the sensitivity of the PDC-TING/SDC-TING to external stimuli has facilitated the development of self-powered sensing systems.

18. There are some details problems. It is suggested that the author carefully check the manuscript and make necessary modifications. Line 227 should include the figure caption “Fig. 2g” for clarity.

Response: We extend our sincere gratitude to the reviewers for their valuable feedback on this critical aspect. Fig. 1 and Fig. 2 have been meticulously revised, providing enhanced and precise explanations of their components. Additionally, we have thoroughly scrutinized the manuscript, implementing necessary revisions and refinements to ensure the excellence and coherence of the paper. We deeply appreciate your insightful guidance and support.

Corresponding revisions please see the revised manuscript (Page 10, lines 174-175):

“In addition, increasing the sputtering time of Au on the PET substrate reduced the internal resistance of the PDC-TING from 10.6 k Ω to 7.4 k Ω (Fig. 1g).”

Fig. 1 | The principle of the triboiontronics and the construction of the PDC-TING. **a**, Triboiontronics via temporally controlling EDL formation. (i) Firstly, when DI water contacted the bottom Au/PET layer, it interacted with the PET substrate through microscopic cracks in the sputtered Au layer, forming a stable EDL. (ii) Secondly, as the top Au/PET layer moved downward to contact with water, the initial CE led to form a new EDL, establishing two EDLs with significantly different symmetries. This created an ion concentration gradient, generating an ionic current I_{il} . (iii) Thirdly, the detachment of the top Au/PET layer halted ion migration. (iv) Through repeated contact and separation cycles, the ion migration persisted until equilibrium was achieved between the two EDLs. **b**, The model display of the PDC-TING. **c**, Comparison of output performance of different generators. **d**, The effect of different liquid types on the PDC-TING output. **e**, The effect of the extent and direction of the ion concentration gradient on the PDC-TING output. **f**, The effect of the Au-layer sputtering time of Au/PET layers on the PDC-TING output. **g**, The effect of the Au-layer sputtering time of Au/PET layers on the internal resistance of the PDC-TING.

Finally, we would like to thank the reviewer again for these valuable comments and for the thoughtful and careful review towards improving our manuscript.

REVIEWER COMMENTS

Reviewer #1 (Remarks to the Author):

Authors properly addressed the comments.

Reviewer #2 (Remarks to the Author):

The manuscript is now clearer, but the revision is major, and the full article got heavier to digest. I suggest that the authors to reduce the amount of data written in text, and instead focus of referring to figures and/or transfer to table data when applicable.

I appreciate the inclusion of discussion in the response (Reviewer 2, Q2) and additions in SI. The addition to SI needs to be referred to in the main text. Smaller comments: Marcus theory seems to me unnecessary, and could be removed. "ion concentration gradient might lead to an increase in chemical potential difference," is wrong, it does indeed, so remove "might". The readers of the main text now get a clearer image of the different sources of energy.

The manuscript explains and demonstrate a combination of well-known sources of energy, and combine them into one harvesting technique. I am however still not convinced by the realization of this combination.

The following comments from the response (Reviewer 2, Q3) should definitely be included in the main text, as it gives a clearer perspective:

"The PDC-TING and SDC-TING, built upon efficient ion migration propelled by the ion concentration gradient outlined in this study, exhibited several notable advantages. These included higher current and lower internal resistance (in the hundred Ohm level), leading to a conventional PR density but a higher QSC density compared to their counterparts. As a result, they are better suited for applications demanding long-term, continuous supply of stable energy output, such as small portable medical devices or monitoring devices (smartwatches, etc.)."

But for this to hold validity, I would need a convincing argument where this can be implemented in a useful way. This following section does not make this more realistic.

"Additionally, by regulating EDL formation, PDC-TING and SDC-TING enable controlled migration and distribution of ions such as Na⁺, K⁺, Cl⁻, and Ca²⁺. In biological systems, the transport of these ions plays a crucial role in various life activities, including cellular communication, nerve impulse transmission, and muscle contraction. This capability mimics ionic flow in biological systems, opening avenues for seamless integration between electronic devices and biological processes. Potential applications include biosensors for real-time monitoring of biochemical processes, brain-computer interfaces for enhanced communication, and neuromorphic computing inspired by brain neural networks."

This energy harvesting technique require controlled and repeated contact and loss of contact, under very specific conditions. Conditions that I see as very artificial and not biological.

In summary, I am still motivated to listen to argument, but I am afraid I am not convinced about the impact of this work. Once declared, I suggest also a revision for 4i where the perspective to where this work is impactful, to again avoid the implication that this is an all-in-one solution, but give the reader a realistic perspective.

Reviewer #3 (Remarks to the Author):

The authors' responses to the questions were thorough and addressed my concerns. I recommend accepting this manuscript for publication in this journal.

Replies to Comments

Triboiontronics via temporally controlling electrical double layer formation

Ref. No.: NCOMMS-24-01792A

Dear editor and reviewers

We have deeply appreciated all reviewers' helpful comments. Provided below is our detailed response to each comment/suggestion. The specific changes made to the manuscript to address each point are highlighted in yellow.

Responses to Reviewers' Comments:

Reviewer #1 (Remarks to the Author):

Authors properly addressed the comments.

Response: We would like to thank the reviewer again for the valuable comments as well as the thoughtful and careful review towards improving our manuscript.

Reviewer #2 (Remarks to the Author):

1. The manuscript is now clearer, but the revision is major, and the full article got heavier to digest. I suggest that the authors to reduce the amount of data written in text, and instead focus of referring to figures and/or transfer to table data when applicable.

Response: Thank you for your valuable suggestions. To improve readability and refine the manuscript, we have removed certain supplementary figures from the Supplementary Information that lacked direct relevance to the main content of the work and removed redundant text narration from the manuscript. In addition, we have enhanced the remaining supplementary figures in the Supplementary Information with more supporting data, presenting them more intuitively as visual information. This approach aimed to make the manuscript clearer and more comprehensible. All the modified parts have been incorporated into the manuscript.

Please check the revised manuscript for corresponding revisions (Page 6, lines 107-123):

“The verification experiment of the regulation mechanism was carried out (Fig. 1c). When 200 μ L water was dropped on the pristine bottom Au/PET layer sputtered with 1-minute Au (Fig. S1a), the PDC-TING that was driven by the ion

concentration gradient could generate a short-circuit current (I_{SC}) of 0.97 μ A in DC form at an operating frequency of 1 Hz. When the water was replaced by polytetrafluoroethylene (PTFE) film attached to the bottom Au/PET layer (Fig. S1b), the PDC-TING was converted to a conventional solid-solid TENG based on the coupling principle of the CE and electrostatic induction (Fig. S2), generating alternating current (AC) electronics signals with I_{SC} of 0.80 μ A. When DI water was dispensed onto the PTFE film adhered to the bottom Au/PET surface (Fig. S1c), the ion migration was hindered, and the conventional solid-liquid TENG was constructed. Weaker AC electronics signals with I_{SC} of 3.53 nA were generated only by electrostatic induction. The corresponding transferred charge (Q_{SC}) and open-circuit voltage (V_{OC}) are shown in Fig. S3. Notably, upon applying 200 μ L of oil or liquid paraffin onto the pristine bottom Au/PET layer, the absence of ions in these solutions could impede the formation of an ion concentration gradient. The AC displacement electrical signals were generated by the CE and electrostatic induction, with I_{SC} of 3.17 nA for oil and I_{SC} of 3.53 nA for liquid paraffin (Fig. 1d). In contrast, under the 99% glycerol containing a small number of ions, the DC output was generated (I_{SC} of 59.17 nA). The corresponding Q_{SC} and V_{OC} are exhibited in Fig. S4.”

Please check the revised manuscript for corresponding revisions (Page 8, lines 140-149):

“When applying 200 μ L water in the PDC-TING with both pristine Au/PET

layers sputtered with 1-minute Au, the constructed ion concentration gradient is weaker, resulting in a lower positive I_{SC} of 0.97 μA . Pre-moistening was exclusively applied to the bottom Au/PET layer with water to pre-form the EDL. It could effectively increase the extent of the ion concentration gradient in the PDC-TING, improving the positive I_{SC} to 8.20 μA . On the contrary, the pre-moistening of the top Au/PET layer could pre-built the EDL, causing a higher reverse ion concentration gradient in the PDC-TING with negative electrical signals (I_{SC} of -2.75 μA). When both Au/PET layers were pre-moistened, the EDLs formed on their respective surfaces, resulting in a diminished ion concentration gradient within the PDC-TING and weaker positive electrical signals with I_{SC} of 0.45 μA . The corresponding Q_{SC} and V_{OC} are displayed in Fig. S6.”

Please check the revised manuscript for corresponding revisions (Page 8, lines 153-157):

“Furthermore, as indicated in Fig. S7, the PDC-TING output remained relatively stable at I_{SC} of about 8.20 μA regardless of whether the bottom Au/PET layer was pre-moistened, immersed in DI water for 2 hours, or subsequently underwent evaporation and drying at room temperature (approximately 25°C) followed by re-moistening. So, the PDC-TING output might be minimally influenced by the swelling property, water content, and dehydration status of the PET substrate.”

Please check the revised manuscript for corresponding revisions (Page 9, lines 162-167):

“Subsequently, the effect of the Au sputtering time of the Au/PET layers on the PDC-TING output was investigated (Fig. 1f). Compared to the PDC-TING output (I_{SC} of 8.20 μA) using the Au/PET layer sputtered with 1-minute Au, when the Au sputtering time was increased to 10 minutes, the I_{SC} was enhanced to 12.18 μA . However, as the Au sputtering time further gradually increased to 30 minutes, the I_{SC} of the PDC-TING was decreased continuously to 4.79 μA . The corresponding Q_{SC} and V_{OC} are shown in Fig. S10.”

Please check the revised manuscript for corresponding revisions (Page 13, lines 246-248):

“The positive electrostatic field on the PA film could promote the PDC-TING output (Fig. 3b), increasing I_{SC} from 12.42 μA to 14.70 μA (corresponding Q_{SC} and V_{OC} are shown in Fig. S16).”

Please check the revised manuscript for corresponding revisions (Page 14, lines 249-251):

“Conversely, the negative electrostatic field on the PTFE film could reduce the PDC-TING output (Fig. 3c), decreasing I_{SC} from 9.06 μA to 6.05 μA (corresponding Q_{SC} and V_{OC} are shown in Fig. S18).”

Please check the revised manuscript for corresponding revisions (Page 14, lines 254-260):

“Initially, untreated Au/PET layers generated I_{SC} of 12.18 μA , with a contact angle of approximately 100° (Fig. S20), indicating hydrophobic properties. Increasing the power of Plasma treatment from 25 W to 75 W reduced the contact angle from about 49° to 19° by introducing oxygen-containing functional groups, enhancing hydrophilicity. It effectively increased the bottom EDL density, boosting the ion concentration gradient and elevating I_{SC} from 54.79 μA to 150.08 μA . Beyond 75 W, further increases in power did not significantly affect hydrophilicity, thereby maintaining output stability. The corresponding Q_{SC} and V_{OC} are shown in Fig. S21.”

Please check the revised manuscript for corresponding revisions (Page 14, lines 262-265):

“When the concentration of LiCl solution was increased from 1 nM to 1 M, improvements were observed in the PDC-TING output (I_{SC} from 13.10 μA to 1035.74 μA), and the internal resistance was effectively reduced from about 7.6 M Ω to 500 Ω . The corresponding Q_{SC} and V_{OC} are shown in Fig. S22.”

Please check the revised manuscript for corresponding revisions (Page 17, lines 287-289):

“The optimal output of PDC-TING could reach I_{SC} of 2.60 mA (Fig. 3f), Q_{SC}

of 41.25 μC (Fig. 3g), P_R (peak power) of 0.85 mW (Fig. 3h), and V_{OC} of 0.25 V (Fig. S24). The Q_{SC} density and P_R density could reach 412.54 mC/m² and 8.45 W/m², respectively.”

2. I appreciate the inclusion of discussion in the response (Reviewer 2, Q2) and additions in SI. The addition to SI needs to be referred to in the main text. Smaller comments: Marcus theory seems to me unnecessary, and could be removed. “ion concentration gradient might lead to an increase in chemical potential difference,” is wrong, it does indeed, so remove “might”. The readers of the main text now get a clearer image of the different sources of energy.

Response: Thank the reviewer for the recognition of our response. We have shifted the relevant theoretical analysis from the Supplementary Information to the manuscript. According to the reviewer's advice and careful analysis, the Marcus theory is indeed unnecessary, so it has been deleted. In addition, the sentence *“In summary, an increase in ion concentration gradient might lead to an increase in chemical potential difference, thereby affecting the change in Gibbs free energy, and consequently impacting the electron transfer rate.”* in the manuscript was modified to *“In summary, an increase in ion concentration gradient will lead to an increase in chemical potential difference, thereby affecting the change in Gibbs free energy, and consequently impacting the electron transfer rate.”* accordingly.

Corresponding revisions please see the revised Supplementary Note 1 in the Supplementary Information (Pages 19-20, lines 332-355):

“In an environment with an ion concentration gradient, the variation in

chemical potential or Gibbs free energy from the ion concentration gradient is analogous to the free energy change in electron transfer reactions. The relationship between electron transfer rate (k) and Gibbs free energy change (ΔG) can be expressed by the following equation:

$$k = k_0 e^{\left(\frac{\Delta G}{4k_B T}\right)} \quad (1)$$

where k_0 is the pre-exponential factor, k_B is the Boltzmann constant, and T is the temperature. ΔG is the Gibbs free energy change, which can be represented by the chemical potential difference Δu :

$$\Delta G = -zF\Delta u \quad (2)$$

where z is the number of charges in the reaction and F is the Faraday constant. In an ion concentration gradient environment, the relationship between the $\Delta \mu$ and the ion concentration gradient can be expressed by the Nernst equation:

$$\Delta u = RT \ln \left(\frac{a_2}{a_1} \right) \quad (3)$$

where R is the ideal gas constant, and a_1 and a_2 represent the activity of ions in the solution. The relationship between the Δu and the concentration of ions c can be derived from the activity of ions:

$$\Delta u = RT \ln \left(\frac{c_2}{c_1} \right) \quad (4)$$

In summary, an increase in ion concentration gradient will lead to an increase in chemical potential difference, thereby affecting the change in Gibbs free energy, and consequently impacting the electron transfer rate. With an internal resistance of 500Ω , the SDC-TING with 1M LiCl could achieve the I_{SC} of 5.58 mA (Fig.

4c), Q_{SC} of 54.07 μC (Fig. 4d), P_R of 3.86 mW (Fig. 4e), and V_{OC} of 0.30 V (Fig. S32). The Q_{SC} density and P_R density of the SDC-TING could reach up to 540.70 mC/m^2 and 38.64 W/m^2 , which was increased by 435.35% and 2869.23% respectively compared to that completely depending on the redox reactions (Fig. S33). After comparison, the contribution of temporally controlling EDL formation and redox reaction on the Q_{SC} of the SDC-TING during the initial contact was 81.32% and 18.68%, respectively.”

Fig. 4 | More efficient triboiontronics via the synergistic effect of temporally controlling EDL formation and redox reaction. **a**, The synergistic principle of temporally controlling EDL formation and redox reaction in the SDC-TING. **b**, The verification experiment of their synergistic effect. **c**, **d**, and **e**, The I_{sc} density, Q_{sc} density, and P_R density of the SDC-TING. **f**, The comparison of the Q_{sc} density of the conventional solid-liquid TENGs, solid-solid TENGs, DC-TING, PDC-TING, and SDC-TING. **g**, The comparison of the charging performance of different types of generators for 1mF capacitors. **h**, The SDC-TING, as an outstanding integrated device for energy harvesting and storage, unveiled profound application prospects in the realms of energy and information flow.

Fig. S32 | The V_{oc} of the SDC-TING could reach about 0.30 V.

Fig. S33 | After the disappearance of the ion concentration gradient caused by temporally controlling EDL formation, the PDC-TING output depended solely on the redox reaction. a, The I_{SC} was 1.01 mA. b, The Q_{SC} was 10.10 μC . c, The V_{OC} was 0.18 V. d, The P_R was 0.13 mW.

3. The manuscript explains and demonstrate a combination of well-known sources of energy, and combine them into one harvesting technique. I am however still not convinced by the realization of this combination.

The following comments from the response (Reviewer 2, Q3) should definitely be included in the main text, as it gives a clearer perspective:

“The PDC-TING and SDC-TING, built upon efficient ion migration propelled by the ion concentration gradient outlined in this study, exhibited several notable advantages. These included higher current and lower internal resistance (in the hundred Ohm level), leading to a conventional PR density but a higher QSC density compared to their counterparts. As a result, they are better suited for applications demanding long-term, continuous supply of stable energy output, such as small portable medical devices or monitoring devices (smartwatches, etc.).”

But for this to hold validity, I would need a convincing argument where this can be implemented in a useful way. This following section does not make this more realistic.

Response: Thanks for your comments and constructive suggestions. The manuscript has been updated to include the output advantages of PDC-TING and

SDC-TING, as well as their typical applications as energy devices, in comparison to hydrovoltaic technology using EDL boundary movement, conventional solid-liquid TENGs, solid-solid TENGs, and DC-TING. The details are as follows:

In contrast, the PDC-TING and SDC-TING, built upon efficient ion migration propelled by the ion concentration gradient, exhibited higher current and lower internal resistance (in the hundred Ohm level), leading to a conventional P_R density but a higher Q_{SC} density compared to their counterparts. They could be better suited for applications demanding long-term, continuous supply of stable energy output, such as calculators or portable environmental monitoring devices. As shown in Fig. 4g, under the condition of ensuring the same total working area and at the same operating frequency of 1 Hz, the conventional solid-solid TENG could hardly supply energy for the 1 mF capacitor, and the DC-TING could only charge it to 0.15 V within 100 s. Excitingly, the pristine SDC-TINGs in the 10-node series could charge a 1 mF capacitor to 1.5V within 60 seconds without the rectifier bridge to power a calculator (Fig. S36 and Movie S1). After 7 recovery times, it could still ensure the stable operation of a hygrothermograph (Fig. S37, and Movie S2).

Moreover, the SDC-TING, as an exceptional integrated device for energy harvesting and storage, showed significant application prospects in the realm of energy. It could operate in two interrelated stages: the integrated energy harvesting and storage stage, followed by the separate energy storage stage (Fig. S38). During the integrated stage, the SDC-TING synergistically generated more

power, delivering an accelerated energy supply to electrical apparatuses such as capacitors (Fig. S38a and b). The subsequent separate energy storage stage substantially enhanced the stability of the SDC-TING (Fig. S38c), significantly surpassing that of the PDC-TING. It could serve as an outstanding integrated power management system without complex conversion circuits to connect primary energy storage, energy harvester, and additional intermediate energy storage units, potentially opening new avenues for research in the efficient harvesting and storage of energy [Fig. 4h(i)].

Corresponding revisions please see the revised manuscript (Pages 22-23, lines 378-389):

“In contrast, the PDC-TING and SDC-TING, built upon efficient ion migration propelled by the ion concentration gradient, exhibited higher current and lower internal resistance (in the hundred Ohm level), leading to a conventional P_R density but a higher Q_{SC} density compared to their counterparts. They could be better suited for applications demanding long-term, continuous supply of stable energy output, such as calculators or portable environmental monitoring devices. As shown in Fig. 4g, under the condition of ensuring the same total working area and at the same operating frequency of 1 Hz, the conventional solid-solid TENG could hardly supply energy for the 1 mF capacitor, and the DC-TING could only charge it to 0.15 V within 100 s. Excitingly, the pristine SDC-TINGs in the 10-node series could charge a 1 mF capacitor to 1.5V

within 60 seconds without the rectifier bridge to power a calculator (Fig. S36 and Movie S1). After 7 recovery times, it could still ensure the stable operation of a hygrothermograph (Fig. S37, and Movie S2).”

Corresponding revisions please see the revised manuscript (Page 23, lines 389-399):

“Furthermore, the SDC-TING, as an outstanding integrated device for energy harvesting and storage, unveiled profound application prospects in the realms of energy and information flow (Fig. 4h). Specifically, in the field of energy, the SDC-TING could operate in two interrelated stages: the integrated energy harvesting and storage stage, followed by the separate energy storage stage (Fig. S38). During the integrated stage, the SDC-TING synergistically generated more power, delivering an accelerated energy supply to electrical apparatuses such as capacitors (Fig. S38a and b). The subsequent separate energy storage stage substantially enhanced the stability of the SDC-TING (Fig. S38c), significantly surpassing that of the PDC-TING. It could serve as an outstanding integrated power management system without complex conversion circuits to connect primary energy storage, energy harvester, and additional intermediate energy storage units, potentially opening new avenues for research in the efficient harvesting and storage of energy [Fig. 4h(i)].”

Fig. 4 | More efficient triboiontronics via the synergistic effect of temporally controlling EDL formation and redox reaction. **a**, The synergistic principle of temporally controlling EDL formation and redox reaction in the SDC-TING. **b**, The verification experiment of their synergistic effect. **c**, **d**, and **e**, The I_{sc} density, Q_{sc} density, and P_R density of the SDC-TING. **f**, The comparison of the Q_{sc} density of the conventional solid-liquid TENGs, solid-solid TENGs, DC-TING, PDC-TING, and SDC-TING. **g**, The comparison of the charging performance of different types of generators for 1mF capacitors. **h**, the SDC-TING, as an outstanding integrated device for energy harvesting and storage, unveiled profound application prospects in the realms of energy and information flow.

Fig. S36 | The pristine SDC-TINGs in series could charge the 1 mF capacitor to 1.5V within 60 seconds and ensure that the calculator was operating normally.

Fig. S37 | The SDC-TINGs in series after 7 recovery times could still charge it to 1.4V and ensure that the hygromicrograph was operating normally.

Fig. S38 | In the field of energy, the SDC-TING could operate in two interrelated stages: the integrated energy harvesting and storage stage, followed by the separate energy storage stage. a, The I_{sc} generated by the SDC-TING in two interrelated stages was compared. b, The charging performance of the SDC-TING for 1 mF capacitor in two interrelated stages was compared. c, The subsequent separate energy storage stage substantially enhanced the stability of the SDC-TING.

4. “Additionally, by regulating EDL formation, PDC-TING and SDC-TING enable controlled migration and distribution of ions such as Na^+ , K^+ , Cl^- , and Ca^{2+} . In biological systems, the transport of these ions plays a crucial role in various life activities, including cellular communication, nerve impulse transmission, and muscle contraction. This capability mimics ionic flow in biological systems, opening avenues for seamless integration between electronic devices and biological processes. Potential applications include biosensors for real-time monitoring of biochemical processes, brain-computer interfaces for enhanced communication, and neuromorphic computing inspired by brain neural networks.”

This energy harvesting technique require controlled and repeated contact and loss of contact, under very specific conditions. Conditions that I see as very artificial and not biological.

In summary, I am still motivated to listen to argument, but I am afraid I am not convinced about the impact of this work. Once declared, I suggest also a revision for 4i where the perspective to where this work is impactful, to again avoid the implication that this is an all-in-one solution, but give the reader a realistic perspective..

Response: Thank you for your feedback. In line with the reviewer's valuable

suggestions and to provide a realistic perspective, we have positioned the future application of SDC-TING in the realms of energy and information flow (Fig. 4h). Specifically, in the field of energy, the SDC-TING could operate in two interrelated stages: the integrated energy harvesting and storage stage, followed by the separate energy storage stage (Fig. S38). During the integrated stage, the SDC-TING synergistically generated more power, delivering an accelerated energy supply to electrical apparatuses such as capacitors (Fig. S38a and b). The subsequent separate energy storage stage substantially enhanced the stability of the SDC-TING (Fig. S38c), significantly surpassing that of the PDC-TING. It could serve as an outstanding integrated power management system without complex conversion circuits to connect primary energy storage, energy harvester, and additional intermediate energy storage units, potentially opening new avenues for research in the efficient harvesting and storage of energy [Fig. 4h(i)]. In the field of information flow, the SDC-TING demonstrated the capability to regulate various ion fluxes, such as Na^+ and Ca^{2+} , by establishing a mechano-driven ion concentration gradient [Fig. 4h(ii)], thereby generating information flow. Ions carry essential information and perform specific functions in biological systems. For instance, Ca^{2+} signals activate neurotransmitter release, modulate neuronal function, and facilitate cardiac muscle relaxation efficiently^[51, 52]. Na^+ ions regulate blood pressure, volume, osmotic equilibrium, and enable excitation propagation by controlling action potential rate and duration^[53, 54]. The property of SDC-TING allows it to mimic human tactile neural circuits, facilitating the

development of bionic neurologic circuits. These circuits can perform threshold-sensing control functions, paving the way for future human-computer interaction and neuromorphic computing. This highlights the significant potential of SDC-TING in energy and information flow.

51. Burgoyne, R. D. Neuronal calcium sensor proteins: generating diversity in neuronal Ca^{2+} signalling. *Nat. Rev. Neurosci.* 8, 182-193 (2007).
52. James, P. et al. Nature and site of phospholamban regulation of the Ca^{2+} pump of sarcoplasmic reticulum. *Nature* 342, 90-92 (1989).
53. Liu, Q. et al. Engineered ionic gates for ion conduction based on sodium and potassium activated nanochannels. *J. Am. Chem. Soc.* 137, 11976-11983 (2015).
54. Wu, K. et al. Biomimetic voltage-gated ultrasensitive potassium-activated nanofluidic based on a solid-state nanochannel. *Langmuir* 33, 8463-8467 (2017).

Corresponding revisions please see the revised manuscript (Pages 23-24, lines 389-409):

“Furthermore, the SDC-TING, as an outstanding integrated device for energy harvesting and storage, unveiled profound application prospects in the realms of energy and information flow (Fig. 4h). Specifically, in the field of energy, the SDC-TING could operate in two interrelated stages: the integrated energy harvesting and storage stage, followed by the separate energy storage stage (Fig. S38). During the integrated stage, the SDC-TING synergistically generated more power, delivering an accelerated energy supply to electrical apparatuses such as capacitors (Fig. S38a and b). The subsequent separate energy

storage stage substantially enhanced the stability of the SDC-TING (Fig. S38c), significantly surpassing that of the PDC-TING. It could serve as an outstanding integrated power management system without complex conversion circuits to connect primary energy storage, energy harvester, and additional intermediate energy storage units, potentially opening new avenues for research in the efficient harvesting and storage of energy [Fig. 4h(i)]. In the field of information flow, the SDC-TING demonstrated the capability to regulate various ion fluxes, such as Na^+ and Ca^{2+} , by establishing a mechano-driven ion concentration gradient [Fig. 4h(ii)], thereby generating information flow. Ions carry essential information and perform specific functions in biological systems. For instance, Ca^{2+} signals activate neurotransmitter release, modulate neuronal function, and facilitate cardiac muscle relaxation efficiently^[51, 52]. Na^+ ions regulate blood pressure, volume, osmotic equilibrium, and enable excitation propagation by controlling action potential rate and duration^[53, 54]. The property of SDC-TING allows it to mimic human tactile neural circuits, facilitating the development of bionic neurologic circuits. These circuits can perform threshold-sensing control functions, paving the way for future human-computer interaction and neuromorphic computing. This highlights the significant potential of SDC-TING in energy and information flow.”

Fig. 4 | More efficient triboiontronics via the synergistic effect of temporally controlling EDL formation and redox reaction. **a**, The synergistic principle of temporally controlling EDL formation and redox reaction in the SDC-TING. **b**, The verification experiment of their synergistic effect. **c**, **d**, and **e**, The I_{sc} density, Q_{sc} density, and P_R density of the SDC-TING. **f**, The comparison of the Q_{sc} density of the conventional solid-liquid TENGs, solid-solid TENGs, DC-TING, PDC-TING, and SDC-TING. **g**, The comparison of the charging performance of different types of generators for 1mF capacitors. **h**, the SDC-TING, as an outstanding integrated device for energy harvesting and storage, unveiled profound application prospects in the realms of energy and information flow.

Fig. S38 | In the field of energy, the SDC-TING could operate in two interrelated stages: the integrated energy harvesting and storage stage, followed by the separate energy storage stage. a, The I_{SC} generated by the SDC-TING in two interrelated stages was compared. **b,** The charging performance of the SDC-TING for 1 mF capacitor in two interrelated stages was compared. **c,** The subsequent separate energy storage stage substantially enhanced the stability of the SDC-TING.

Finally, we would like to thank the reviewer again for these valuable comments and for the thoughtful and careful review towards improving our manuscript.

Reviewer #3 (Remarks to the Author):

The authors' responses to the questions were thorough and addressed my concerns.

I recommend accepting this manuscript for publication in this journal.

Response: Once again, we would like to thank the reviewer for the valuable comments and for the thoughtful and careful review towards improving our manuscript.